# Lost in the Non-convex Loss Landscape: How to Fine-tune the Large Time Series Model?

**Xu Zhang, Peng Wang**[*]**, Wei Wang**
Shanghai Key Laboratory of Data Science
College of Computer Science and Artificial Intelligence
Fudan University, Shanghai 200082, China
`xuzhang22@m.fudan.edu.cn`
`{pengwang5,weiwang1}@fudan.edu.cn`

## Abstract

Recently, large time series models (LTSMs) have gained increasing attention due to their similarities to large language models, including flexible context length, scalability, and task generality, outperforming advanced task-specific models. However, prior studies indicate that pre-trained LTSMs may exhibit a poorly conditioned non-convex loss landscape, leading to limited trainability. As a result, direct fine-tuning tends to cause overfitting and suboptimal performance, sometimes even worse than training from scratch, substantially diminishing the benefits of pre-training. To overcome this limitation, we propose Smoothed Full Fine-tuning (SFF), a novel fine-tuning technology. Specifically, we construct an auxiliary LTSM via random initialization to obtain a smoother loss landscape, and then linearly interpolate its weights with those of the pre-trained model to smooth the original landscape. This process improves trainability while preserving pre-trained knowledge, thereby enabling more effective downstream fine-tuning. From an optimization perspective, SFF perturbs sharp minima without significantly harming flat regions, facilitating escape from poor local basins toward smoother and more generalizable solutions. Extensive experiments on benchmark datasets demonstrate consistent improvements across eight representative LTSMs, including Timer, TimesFM, MOMENT, UniTS, MOIRAI, Chronos, TTMs, and Sundial, on diverse downstream tasks. The code is available at the link: `https://github.com/Meteor-Stars/SFF`.

## 1 Introduction

Time series (TS) are ubiquitous in the real world, such as financial trading (Zhang et al., 2025a; 2024; Sen & Mehtab, 2022), industry sensor monitoring (Zhang et al., 2025b), and traffic systems (Gao et al., 2024a). TS analysis can help inform better decisions by revealing underlying trends and behaviors. Recently, large models developed via generative pre-training transformers (GPT) have demonstrated advanced capabilities, including flexible context length, strong cross-domain generalization, versatility across diverse tasks, and scalability with increasing model size and pre-training data. Inspired by this success, large time series models (LTSMs), e.g., Timer (Liu et al., 2024), TimesFM (Das et al., 2024), and MOMENT (Goswami et al., 2024), have been proposed to transfer these advantages to time series analysis, achieving improved performance in forecasting (Box et al., 2015), interpolation (Friedman, 1962), and anomaly detection (Ren et al., 2019). Pre-trained on large-scale time series datasets, LTSMs effectively capture universal temporal patterns, such as trends, amplitudes, frequencies, and phases (Goswami et al., 2024), thereby substantially benefiting downstream tasks.

However, recent theoretical studies suggest that large-scale training may drive models toward sharp minima (Keskar et al., 2016) in a highly non-convex loss landscape (Li et al., 2018), leading to optimization challenges during fine-tuning. Empirically, Figure 1 visualizes the loss landscape of a pre-trained LTSM on downstream datasets, revealing pronounced local protrusions (highlighted by the black contour), consistent with theoretical findings. Such steep and *non-convex (non-smooth)* landscapes indicate poor trainability and may trap models in suboptimal local minima (e.g., orange

---
[*]Corresponding author

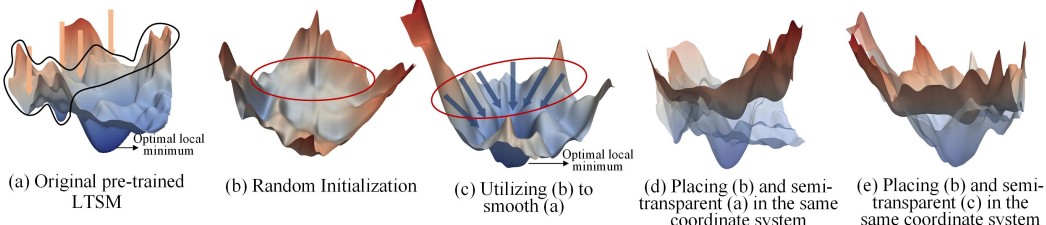

Figure 1: Loss landscape comparisons based on the LTSM Timer and exchange rate dataset. Smoother is better. The unsmooth loss landscape shows a non-convex structure and indicates poorer trainability (Li et al., 2018), e.g., Figure 1(a). More cases are shown in Appendix Figure 7 and Figure 8.

arrows in Figure 1(a)), thereby exacerbating overfitting and degrading generalization (Li et al., 2018). Furthermore, we observe that direct full fine-tuning consistently achieves the lowest training loss but even higher test loss than training from scratch, as shown in Appendix Figure 9, further confirming severe overfitting and supporting our analysis.

Intuitively, the non-convex loss landscape may arise from overfitting during pre-training, such as learning data-specific patterns rather than generalizable features or converging to sharp minima (Keskar et al., 2016). Consequently, existing fine-tuning strategies, including *Full Fine-tuning (FF)*, *Linear Probing (LP)*, and *Linear Probing then Full Fine-tuning (LPFF)* (Kumar et al.), often fail to achieve satisfactory performance, as they cannot effectively smooth the non-convex landscape. This limitation hinders the pre-trained LTSM from converging to better local minima during fine-tuning (e.g., the dark blue region in Figure 1(a)).

To mitigate the adverse impact of such a "non-convex loss landscape" on downstream fine-tuning, we empirically observe that randomly initialized LTSMs generally exhibit much smoother landscapes, as shown in Figure 1(b). This observation naturally raises the question: *Can we leverage the smooth landscape in Figure 1(b) to smooth and reshape the loss landscape of the pre-trained model in Figure 1(a), thereby enhancing its trainability while preserving its pre-trained knowledge?*

Based on these insights, we propose Smoothed Full Fine-tuning (SFF) to better exploit the pre-trained knowledge of LTSMs for improved fine-tuning performance. The method consists of two key steps. **First**, we construct an auxiliary LTSM via random initialization. Unlike the pre-trained model, this auxiliary one exhibits a smoother and more convex loss landscape (Figure 1(b)), leading to better trainability. However, it lacks pre-trained knowledge, as shown in Figure 1(d), where its minimum loss is significantly higher than that of the pre-trained model, explaining the latter's strong zero-shot performance. **Second**, we smooth the pre-trained model's loss landscape by linearly interpolating its weights with those of the auxiliary LTSM. The resulting smoothed LTSM effectively preserves the pre-trained knowledge while inheriting the improved trainability of the auxiliary model. As shown in Figure 1(e), its minimum loss remains substantially lower than that of the randomly initialized model, confirming effective knowledge preservation. Empirical evidence is provided in Section 5.4.

Overall, smoothing alleviates severe protrusions in the pre-trained loss landscape (Figure 1(c)), yielding a more convex structure and facilitating optimization. As a result, gradient descent is more likely to converge to better local optima (indicated by the dark blue arrows), thereby enhancing fine-tuning stability and effectiveness. The proposed technology requires only linear interpolation of model parameters prior to fine-tuning, without introducing additional memory or computational overhead. From an optimization perspective, SFF perturbs sharp minima while largely preserving flat regions, enabling escape from suboptimal basins toward smoother and more generalizable solutions. Further theoretical analysis is provided in Section 3.1.

In summary, our contributions are as follows:

- We uncover a key observation that pretrained LTSMs may suffer from overfitting during pre-training, leading to a poorly conditioned non-convex loss landscape and reduced trainability, which substantially limits their fine-tuning performance on downstream tasks.

- We are the first to propose the weight-interpolation-based fine-tuning technology, termed *Smoothed Full Fine-tuning (SFF)*. SFF smooths the loss landscape via linear interpolation between a pretrained model (with **rich** knowledge but **poor** trainability) and a randomly

initialized one (with **limited** knowledge but **good** trainability), yielding a smoothed model that preserves pretrained knowledge while achieving improved trainability. We further provide theoretical insights into its effectiveness from an optimization perspective (Section 3.1). Notably, SFF introduces no additional memory or computational overhead.

- Extensive experiments on time series forecasting (TSF) and anomaly detection tasks demonstrate that SFF consistently outperforms popular fine-tuning strategies, including *Full Fine-tuning (FF)*, *Linear Probing (LP)*, *Linear Probing then Full Fine-tuning (LPFF)*, and other popular optimization techniques. SFF improves the performance of eight representative LTSMs across diverse architectures (encoder-only, decoder-only, encoder–decoder, and MLP-only) and model scales (3.8GB to 3MB).

## 2 RELATED WORKS

Popular fine-tuning approaches include full fine-tuning, linear probing, and linear probing followed by full fine-tuning (LP-FF) (Kumar et al.). Due to limited space, we place related work on fine-tuning, optimization strategies, and time series foundation models in Appendix A.2.

**Difference from existing weight interpolation methods.** To the best of our knowledge, weight interpolation has not been explored for fine-tuning from the perspective of loss landscape theory. Although weight averaging and interpolation (Vlaar & Frankle, 2022) have been studied in model merging (Wortsman et al., 2022) and continual learning (Kozal et al., 2024), they do not address the core challenge in LTSMs. Our method differs fundamentally in the following aspects:

**(1) Different goals.** Existing methods (Wortsman et al., 2022; Kozal et al., 2024) mainly focus on model ensembling, i.e., interpolating among multiple **well-trained models** to improve generalization or alleviate catastrophic forgetting. **In contrast, we exploit interpolation to *smooth the loss landscape of a single pretrained model* using a randomly initialized model, thereby improving its trainability for fine-tuning.**

**(2) Different pipelines.** Prior works typically apply the interpolated model directly to downstream tasks without further training. In contrast, we perform **additional fine-tuning after interpolation**, enabling the smoothed loss landscape to yield better optimization and performance.

**(3) New theoretical analysis.** We formalize the contrast between the smooth loss landscape of randomly initialized models and the steep, irregular landscape of pretrained ones, and provide theoretical analysis explaining why interpolation can effectively exploit this difference to enhance fine-tuning.

## 3 SMOOTHING THE LOSS LANDSCAPE OF THE PRE-TRAINED LTSM FOR FINE-TUNING

We propose the *smoothed fine-tuning* technology to enhance the fine-tuning performance of various pretrained LTSMs, as illustrated in Figure 2.

### 3.1 MOTIVATION AND THEORETICAL ANALYSIS

In this section, we explain why Smoothed Full Fine-tuning (SFF) is effective from the perspective of deep learning optimization theory. Specifically, the loss landscape can be divided into flat and sharp regions Li et al. (2018). Flat regions imply better generalization, as the model is more tolerant to weight perturbations within these regions Keskar et al. (2016); Hochreiter & Schmidhuber (1997); Foret et al. (2020). This motivates SFF to perform linear interpolation between randomly initialized weights and pretrained weights to obtain a smoother loss landscape and improved fine-tuning performance. Such interpolation does not significantly alter the weights located in flat regions due to their high tolerance to perturbations. **For weights in sharp regions, SFF behaves similarly to introducing momentum (analogous to the momentum mechanism in the Adam Kingma & Ba (2014) optimizer) via random weight interpolation**, enabling them to escape sharp regions and move toward smoother basins, thereby facilitating more stable and effective fine-tuning.

Next, we provide a more rigorous theoretical derivation to prove the effectiveness of SFF.

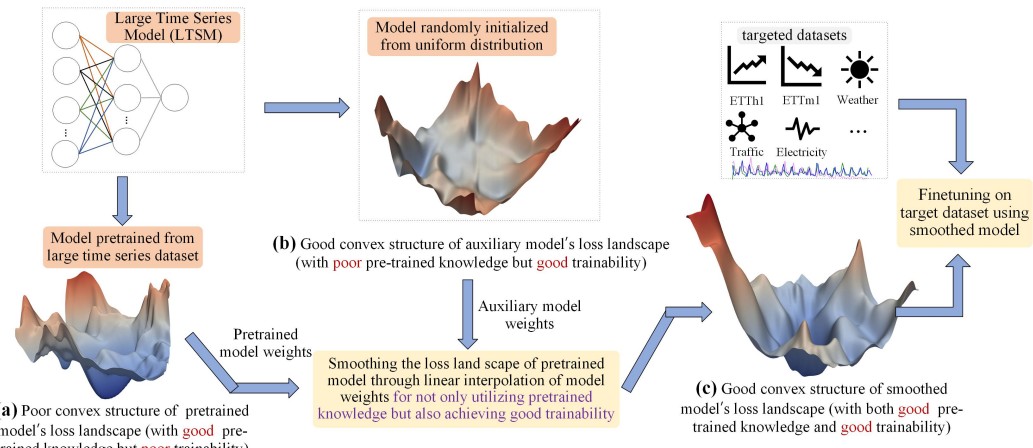

Figure 2: *Smoothed Full Fine-tuning* (SFF). By linearly interpolating the pretrained and randomly initialized LTSMs, we obtain a smoothed model that preserves pretrained knowledge while exhibiting a smoother and more trainable loss landscape, thereby enabling more stable and effective fine-tuning.

### 3.1.1 INFLUENCE OF INTERPOLATION ON SHARP AND FLAT LOSS LANDSCAPE

Given an MSE loss function $\mathcal{L}(\Theta)$, where $\Theta$ denotes the model parameters, the minimizer $\Theta^*$ can correspond to either a sharp or a flat minimum. Inspired by (Keskar et al., 2016), we characterize the sharpness and flatness of the loss landscape using the Hessian matrix. Specifically, by analyzing the maximum eigenvalue $\lambda_{\max}(\cdot)$ of the Hessian $H = \nabla^2\mathcal{L}(\Theta^*)$ ($H \in \mathbb{R}^{d \times d}$, where $d$ denotes the parameter dimension), we define sharp and flat minima as follows:

**Theorem 1** (Sharp minima)**.** *The Hessian $\nabla^2\mathcal{L}(\Theta^*)$ exhibits large eigenvalues (i.e., $\lambda_{max}(\nabla^2\mathcal{L}(\Theta^*)) \gg \tau$ where $\tau > 0$ is a threshold), indicating a steep loss landscape in which small parameter perturbations can cause substantial increases in loss.*

**Theorem 2** (Flat minima)**.** *The Hessian $\nabla^2\mathcal{L}(\Theta^*)$ has small eigenvalues (i.e., $\lambda_{max}(\nabla^2\mathcal{L}(\Theta^*)) \leq \tau$ where $\tau > 0$), indicating a flat loss landscape in which the loss is robust to parameter perturbations.*

Proof details for the above theorems are provided in Appendix Section A.1. Next, we analyze the impact of interpolation on the loss landscape in both sharp and flat regions.

**Smoothing (perturbing) sharp minima.** The SFF interpolation defines the smoothed parameters as $\Theta_3 = \alpha\Theta_1^* + (1 - \alpha)\Theta_2$, where $\Theta_1^*$ denotes the pretrained LTSM, which is more likely to lie in a sharp minimum (Figure 1(a)) due to large-scale pretraining (Keskar et al., 2016). In contrast, $\Theta_2$ (randomly initialized) lies in a flat region, as illustrated in Figure 1(b). **In the next section, we provide theoretical analysis showing that commonly used initialization schemes, including Kaiming and Xavier, inherently produce flat loss landscapes.** After pre-training the LTSM, the sharp minimum $\Theta_1^*$ is characterized by a large Hessian eigenvalue, i.e., $\lambda_{\max}(\nabla^2\mathcal{L}(\Theta_1^*)) \gg \tau$. In contrast, the randomly initialized point $\Theta_2$ lies in a flat region, yielding $\lambda_{\max}(\nabla^2\mathcal{L}(\Theta_2)) \leq \tau$.

Under a local quadratic approximation of the loss function along the interpolation path, the Hessian at $\Theta_3$ can be approximated as a convex combination of the Hessians at $\Theta_1^*$ and $\Theta_2$:

$$\nabla^2\mathcal{L}(\Theta_3) \approx \alpha\nabla^2\mathcal{L}(\Theta_1^*) + (1 - \alpha)\nabla^2\mathcal{L}(\Theta_2) \tag{1}$$

Consequently, the maximum eigenvalue satisfies:

$$\lambda_{\max}(\nabla^2\mathcal{L}(\Theta_3)) \lesssim \alpha\lambda_{\max}(\nabla^2\mathcal{L}(\Theta_1^*)) + (1 - \alpha)\lambda_{\max}(\nabla^2\mathcal{L}(\Theta_2)) \tag{2}$$

Since $\lambda_{\max}(\nabla^2\mathcal{L}(\Theta_2)) \ll \lambda_{\max}(\nabla^2\mathcal{L}(\Theta_1^*))$ (flat vs. sharp), it follows that $\lambda_{\max}(\nabla^2\mathcal{L}(\Theta_3)) < \lambda_{\max}(\nabla^2\mathcal{L}(\Theta_1^*))$ for $\alpha \in (0, 1)$. This indicates that interpolation effectively reduces local sharpness, enabling parameters in sharp regions to escape unfavorable basins and move toward smoother ones, thereby improving fine-tuning. These provide theoretical support for the smoothing effect of SFF.

**Preservation of Flat Regions.** After pre-training the LTSM, for flat minimum points $\bar{\Theta}_1^*$, interpolation preserves their flatness. Specifically, both $\bar{\Theta}_1^*$ and $\Theta_2$ reside in flat regions of the loss landscape, which can be formulated as:

$$\lambda_{\max}\left(\nabla^2\mathcal{L}(\bar{\Theta}_1^*)\right) \leq \tau, \quad \lambda_{\max}\left(\nabla^2\mathcal{L}(\Theta_2)\right) \leq \tau \tag{3}$$

Similarly, the Hessian at the interpolated point $\Theta_3 = \alpha\bar{\Theta}_1^* + (1-\alpha)\Theta_2$ satisfies $\nabla^2\mathcal{L}(\Theta_3) \approx \alpha\,\nabla^2\mathcal{L}(\bar{\Theta}_1^*) + (1-\alpha)\,\nabla^2\mathcal{L}(\Theta_2)$. Consequently, it follows that:

$$\lambda_{\max}\left(\nabla^2\mathcal{L}(\Theta_3)\right) \lesssim \alpha\lambda_{\max}\left(\nabla^2\mathcal{L}(\bar{\Theta}_1^*)\right) + (1-\alpha)\lambda_{\max}\left(\nabla^2\mathcal{L}(\Theta_2)\right) \leq \alpha\tau + (1-\alpha)\tau = \tau \quad (4)$$

Hence, $\Theta_3$ remains within a flat region. As noted in (Foret et al., 2020), this implies that interpolation does not compromise the pre-existing flat minima $\bar{\Theta}_1^*$.

Overall, from a rigorous mathematical perspective, interpolation (or perturbs) smooths sharp minima while preserving the integrity of originally flat regions, improving fine-tuning performance.

### 3.1.2 PARAMETER INITIALIZATION AND THE SMOOTHNESS OF THE CORRESPONDING LOSS LANDSCAPE

Visualizations indicate that initialization methods such as Kaiming (He et al., 2015) and Xavier (Glorot & Bengio, 2010) yield a smooth loss landscape. Motivated by this, we adopt them as auxiliary initialization schemes for LTSM to facilitate smoothed full finetuning. In this section, we further analyze this phenomenon from a theoretical perspective.

Prior work (Fort & Scherlis, 2019) quantifies loss landscape smoothness under Kaiming and Xavier initializations via the ratio of the Hessian trace $\mathrm{Tr}(H)$ to its Frobenius norm $\|H\|_F$, demonstrating that $\frac{\mathrm{Tr}(H)}{\|H\|_F} \gg 1$, both theoretically and experimentally. That is, mainstream initialization indeed produces a smooth, flat, and more easily optimizable loss landscape. We provide a further theoretical analysis of this conclusion. Specifically, exploiting the symmetry of $H$, we expand the formula as:

$$\frac{\mathrm{Tr}(H)}{\|H\|_F} = \frac{\mathrm{Tr}(H)}{\sqrt{\sum_{i=1}^{d} h_i^2}} = \frac{\mathrm{Tr}(H)}{\sqrt{\mathrm{Tr}(H^T H)}} = \frac{\sum \lambda_i}{\sqrt{\sum \lambda_i^2}} \gg 1 \tag{5}$$

This implies that the sum of eigenvalues far exceeds the square root of the sum of squared eigenvalues, suggesting that most eigenvalues are positive and relatively evenly distributed, rather than dominated by extreme outliers. In other words, most cases satisfy $\lambda(\nabla^2\mathcal{L}(\Theta^*)) \leq \tau$ with $\tau > 0$.

According to the definitions in **Theorem 1** and **Theorem 2**, this indicates that the loss surface exhibits a smooth, valley-like geometry dominated by positive curvature. Consequently, most random descent directions remain stable and low-curvature, facilitating more stable and efficient optimization.

In contrast, if $\frac{\mathrm{Tr}(H)}{\|H\|_F} \lesssim 1$, it indicates a roughly balanced mix of positive and negative eigenvalues $\lambda$, resulting in a steep and irregular loss landscape. Consequently, convergence slows, and the optimizer is more prone to suboptimal local minima. Therefore, Kaiming or Xavier initialization provides a stable and smooth loss landscape, and we adopt them for randomly initializing the auxiliary LTSM.

### 3.2 SMOOTHING THE LOSS LANDSCAPE THEN FINE-TUNING

We define the training set as $\mathcal{D} = \{(\mathcal{X}_1, Y_1), \ldots, (\mathcal{X}_N, Y_N)\}$, where each time series $\mathcal{X}_N = [x_1, x_2, \ldots, x_t] \in \mathbb{R}^{t \times v}$ has length $t$ across $v$ time variables. The pre-trained LSTM is divided into a backbone $G(\mathcal{X}, \Phi_1)$ and a linear head $\mathbf{W}_{\text{head1}} \in \mathbb{R}^{d \times h}$, where $\Phi_1$ denotes the backbone parameters, $d$ and $h$ are the output dimensions of the backbone and head.

Similarly, the parameters of the **auxiliary** LSTM are defined as $\Theta_2 = [\Phi_2, \mathbf{W}_{\text{head2}}]$. Given a coefficient $\alpha$, the new model weights $\Theta_3 = [\Phi_3, \mathbf{W}_{\text{head3}}]$ are obtained via linear interpolation between the **auxiliary** LSTM ($\Theta_2$) and the **pre-trained** LSTM ($\Theta_1$). Accordingly, the formulations for full fine-tuning, linear probing, and the loss function are given in Eq. 6, Eq. 7, and Eq. 8, respectively.

$$f(X, \Theta_3) = G(\mathcal{X}, \alpha\Phi_1 + (1-\alpha)\Phi_2)^T(\alpha\mathbf{W}_{\text{head1}} + (1-\alpha)\mathbf{W}_{\text{head2}}) \tag{6}$$

$$f(\mathcal{X}, \Theta_3) = G(\mathcal{X}, \alpha\Phi_1 + (1-\alpha)\Phi_2)_{\textbf{frozen}}^T(\alpha\mathbf{W}_{\text{head1}} + (1-\alpha)\mathbf{W}_{\text{head2}}) \tag{7}$$

$$\arg\min_{\alpha\Theta_1 + (1-\alpha)\Theta_2} \sum_{(\mathcal{X}_i, Y_i) \in \mathcal{D}} \mathcal{L}\left(f(\mathcal{X}_i, \alpha\Theta_1 + (1-\alpha)\Theta_2), Y_i\right) \tag{8}$$

where $\alpha$ is the interpolation coefficient controlling the proportion of pretrained knowledge retained, with larger $\alpha$ preserving more of it. $\mathcal{L}$ denotes the Mean Squared Error (MSE) loss. $Y_i$ depends on

the downstream task, e.g., for imputation, $Y_i \in \mathcal{X}_i$ represents the masked values; for forecasting, $Y_i$ denotes the consecutive future values of $\mathcal{X}_i$.

Smoothing the loss landscape can be implemented in just a few lines of PyTorch and the example code is provided in Appendix Algorithm 1.

## 4 EXPERIMENTAL SETTINGS

**LTSM Baselines.** We use the eight popular LTSMs as baselines with diverse architectures, including **encoder-only** (Moirai (Woo et al., 2024a) and MOMENT (Goswami et al., 2024)), **decoder-only** (Sundial (Liu et al., 2025), Timer (Liu et al., 2024), and TimesFM (Das et al., 2024)), **encoder-decoder** (Chronos (Ansari et al., 2024) and UniTs (Gao et al., 2024b)), and **light-weight MLP model** (TTMs (Ekambaram et al., 2024)). They also include models of different sizes, ranging from larger TimesFM (3.8GB) to smaller TTMs (3MB). We verify that our method can enhance their performance. We download these pre-trained models from the official links for experiments, e.g., with model sizes being 851MB, 2.6GB, and 3.8GB for Timer, MOMENT, and TimesFM, respectively.

**Fine-tuning baselines.** In addition to comparing with typical fine-tuning baselines, e.g., full fine-tuning (FF), linear probing (LP), and linear probing first and then full fine-tuning (LP-FF). We also incorporated various optimization strategies employed during model training, such as label smoothing, SAM (Foret et al., 2020), SWA (Izmailov et al., 2018), Mixout (Lee et al.), and L2-SP (Xuhong et al., 2018). Comparison results are shown in the Appendix Table 10 due to limited space. More details about datasets, evaluations, and implementation details are shown in Appendix A.4.

## 5 EXPERIMENTAL RESULTS

We conduct extensive experiments to validate the effectiveness of the proposed smoothed fine-tuning, including 8 TSF datasets and 250 anomaly detection datasets, also involving 8 popular LTSMs. To ensure the effectiveness of our method is not a random occurrence, we have conducted experiments with multiple random seeds under varying available data proportions. Our method also achieves improvements on the imputation task, as shown in Appendix Figure 10 due to limited space.

We also evaluate different initialization schemes and random seeds on SFF in Appendix Section A.6.1, Tables 11 and 12. The results show that mainstream initializations (e.g., Kaiming, Xavier) consistently yield stable gains, and SFF shows no noticeable sensitivity to the initialization seed.

Table 1: MSE results of fine-tuning the LTSM Timer for TSF with prediction length 96 under different data proportions. SFF, FF, and TFS denote *smoothed full fine-tuning*, *full fine-tuning*, and *training from scratch*, respectively. Full standard deviations and additional results under 1%–20% data proportions, as well as MAE results, are reported in Appendix Tables 15, 16, 17 18, 19, and 20.

| Data proportion | 25% | | | 50% | | | 75% | | | 100% | | |
|---|---|---|---|---|---|---|---|---|---|---|---|---|
| Methods | SFF | FF | TFS | SFF | FF | TFS | SFF | FF | TFS | SFF | FF | TFS |
| Exchange | **0.0805** | 0.0865 | 0.1441 | **0.0802** | 0.0891 | 0.114 | **0.0802** | 0.0914 | 0.1026 | **0.08** | 0.091 | 0.0981 |
| Standard deviation | ±4.5e-4 | ±1.9e-4 | ±2.0e-3 | ±5.4e-4 | ±2.3e-3 | ±9.9e-4 | ±1.2e-3 | ±1.6e-3 | ±8.8e-4 | ±7.6e-4 | ±1.3e-4 | ±1.2e-3 |
| ETTh1 | **0.3506** | 0.355 | 0.3788 | **0.3494** | 0.3573 | 0.367 | **0.3493** | 0.358 | 0.3593 | **0.3547** | 0.3709 | 0.36 |
| ETTh2 | **0.271** | 0.2866 | 0.2891 | **0.273** | 0.2905 | 0.2775 | **0.2772** | 0.3032 | 0.2796 | **0.2737** | 0.3047 | 0.2777 |
| ETTm1 | **0.298** | 0.3049 | 0.333 | **0.2955** | 0.3069 | 0.3189 | **0.2956** | 0.3092 | 0.3116 | **0.2954** | 0.3128 | 0.3093 |
| ETTm2 | **0.1594** | 0.1707 | 0.1741 | **0.1605** | 0.1718 | 0.1627 | **0.1623** | 0.1838 | 0.1651 | **0.16** | 0.1784 | 0.1644 |
| Weather | **0.144** | 0.1472 | 0.1627 | **0.1441** | 0.1523 | 0.1538 | **0.1466** | 0.1665 | 0.1559 | **0.1443** | 0.1612 | 0.1526 |
| Electricity | **0.1303** | 0.1344 | 0.1365 | **0.1301** | 0.1347 | 0.1327 | **0.13** | 0.1367 | 0.1326 | **0.1304** | 0.1344 | 0.1324 |
| Traffic | **0.3488** | 0.3582 | 0.3688 | **0.3497** | 0.3586 | 0.3552 | **0.3478** | 0.361 | 0.3606 | **0.3551** | 0.3599 | 0.3609 |

### 5.1 FINE-TUNING TIMER FOR MULTIVARIATE TIME SERIES FORECASTING (TSF) AND ANOMALY DETECTION

Experiments show that training from scratch (TFS) requires substantially more data to achieve strong accuracy, whereas pretrained LSTMs can reach or even surpass TFS's full-data performance using

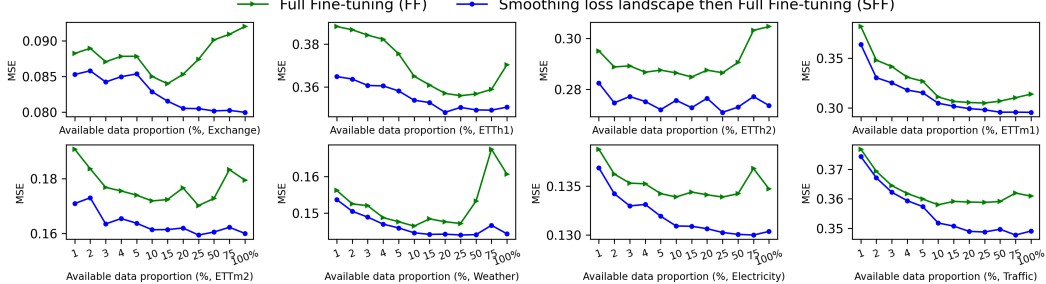

Figure 3: MSE of different fine-tuning strategies on Timer with prediction length 96, across multiple datasets (test sets) under varying data proportions (1%–100%).

only 10%–25% of the data. However, pretrained LSTMs may suffer from poor trainability caused by overfitting, leading to steep and unsmoothed loss landscapes that hinder effective fine-tuning. As a result, full fine-tuning (FF) may struggle to improve and can even degrade as the data scale increases (Figure 3). In contrast, Smoothed Full Fine-tuning (SFF) consistently surpasses FF, exhibiting steadily improving performance as more data becomes available. Across nine public datasets, SFF reduces MSE by an average of 3% and up to 6.5% compared to FF (Table 1). These results demonstrate that SFF effectively smooths the loss landscape, enhances trainability, and better exploits pretrained knowledge without introducing additional memory or computational overhead.

Similarly, in the anomaly detection task, we observe that TFS generally outperforms FF. When MSE is used as the anomaly score, higher predicted MSE values on anomalous segments indicate better detection performance. As shown in Table 2, TFS yields a higher average MSE on anomalous segments than FF. In contrast, SFF achieves significantly higher MSE scores than both FF and TFS, demonstrating its clear advantage in anomaly detection.

Table 2: Anomaly detection results with Timer. Predicted MSE on anomalous segments is reported, with higher values indicating better performance. Results over 250 datasets and four seeds are summarized in six groups, with full results and standard deviations in Appendix Tables 22 and 23.

| Group 1 (41 datasets) | | | | | | Group 2 (41 datasets) | | | | | | Group 3 (41 datasets) | | | | | |
|---|---|---|---|---|---|---|---|---|---|---|---|---|---|---|---|---|---|
| SFF | | FF | | TFS | | SFF | | FF | | TFS | | SFF | | FF | | TFS | |
| MSE | Wins | MSE | Wins | MSE | Wins | MSE | Wins | MSE | Wins | MSE | Wins | MSE | Wins | MSE | Wins | MSE | Wins |
| **0.136** | **31.3** | 0.072 | 4.3 | 0.073 | 5.3 | **0.206** | **32.0** | 0.098 | 5.3 | 0.089 | 3.7 | **0.209** | **30.3** | 0.104 | 5.0 | 0.112 | 5.7 |
| ±1.4e-2 | ±1.7 | ±1.3e-2 | ±1.9 | ±1.6e-2 | ±0.47 | 7.7e-3 | 0.82 | 4.0e-3 | 0.94 | 1.0e-2 | 0.47 | 2.9e-2 | 1.7 | 1.0e-2 | 0.0 | 2.1e-2 | 1.7 |
| Group 4 (41 datasets) | | | | | | Group 5 (41 datasets) | | | | | | Group 6 (45 datasets) | | | | | |
| SFF | | FF | | TFS | | SFF | | FF | | TFS | | SFF | | FF | | TFS | |
| MSE | Wins | MSE | Wins | MSE | Wins | MSE | Wins | MSE | Wins | MSE | Wins | MSE | Wins | MSE | Wins | MSE | Wins |
| **0.201** | **33.7** | 0.084 | 3.3 | 0.087 | 4.0 | **0.163** | **29.0** | 0.078 | 3.3 | 0.09 | 8.7 | **0.157** | **34.3** | 0.085 | 5.0 | 0.09 | 5.7 |
| ±2.9e-2 | ±0.47 | ±9.9e-3 | ±0.47 | ±8.1e-3 | ±0.82 | 3.4e-2 | 2.2 | 6.2e-3 | 0.47 | 2.4e-2 | 1.9 | 6.7e-3 | 1.2 | 9.8e-3 | 2.4 | 3.6e-3 | 2.4 |

Table 3: Fine-tuning MSE of applying SFF to TimesFM and MOMENT with prediction length 96. Full standard deviations and MAE results are reported in Appendix Tables 24 and 25.

| Data proportion | 25% (TimesFM) | | | 100% (TimesFM) | | | 25% (MOMENT) | | | 100% (MOMENT) | | |
|---|---|---|---|---|---|---|---|---|---|---|---|---|
| Methods | SFF | FF | TFS | SFF | FF | TFS | SFF | FF | TFS | SFF | FF | TFS |
| Exchange | **0.1139** | 0.1276 | 0.1209 | **0.1149** | 0.1452 | 0.1199 | **0.1502** | 0.2648 | 0.1564 | **0.1064** | 0.1448 | 0.1091 |
| Standard deviation | 2.0e-3 | 4.2e-3 | 2.9e-4 | 6.3e-4 | 1.7e-2 | 2.3e-3 | 2.4e-3 | 4.6e-4 | 3.6e-3 | 5.8e-4 | 1.4e-4 | 2.6e-4 |
| ETTh1 | **0.3955** | 0.4382 | 0.4638 | **0.406** | 0.5101 | 0.4358 | **0.4287** | 0.4454 | 0.454 | **0.3757** | 0.3951 | 0.387 |
| ETTh2 | **0.3232** | 0.3384 | 0.3325 | **0.3198** | 0.3483 | 0.347 | **0.3199** | 0.3328 | 0.3326 | **0.2818** | 0.2936 | 0.2979 |
| ETTm1 | **0.3429** | 0.4001 | 0.3903 | **0.3478** | 0.3756 | 0.3926 | **0.3457** | 0.3587 | 0.3538 | **0.3139** | 0.3148 | 0.3272 |
| ETTm2 | **0.1983** | 0.2061 | 0.2091 | **0.2026** | 0.2122 | 0.225 | **0.1793** | 0.192 | 0.1846 | **0.1692** | 0.172 | 0.1736 |
| Weather | **0.0865** | 0.0885 | 0.1995 | **0.082** | 0.1184 | 0.1902 | **0.1673** | 0.1682 | 0.169 | **0.1548** | 0.1558 | 0.161 |

## 5.2 Applying smoothed full fine-tuning (SFF) for other LTSMs

As shown in Table 3, for TimesFM and MOMENT, FF performs worse than TFS in several cases, indicating that pretrained LTSMs may indeed suffer from overfitting. In contrast, SFF consistently outperforms both FF and TFS. Compared with FF, SFF achieves average improvements of 11.45% for TimesFM and 8.31% for MOMENT across different available data proportions.

Moreover, as reported in Table 4, experiments on additional LTSMs, including UniTS, MOIRAI, Chronos, TTMs, and Sundial, further demonstrate that SFF consistently surpasses FF, confirming the effectiveness of the interpolation-based smoothing technology. This is because interpolation perturbs sharp minima without significantly affecting inherently flat regions, thereby facilitating the escape from sharp basins toward smoother and better optima.

Overall, our method generalizes well across diverse LTSM architectures, including encoder-only, decoder-only, encoder-decoder, and MLP-only models, demonstrating universality and robustness.

Table 4: MSE of fine-tuning additional LTSMs for TSF with prediction length 96. "-" indicates unavailable preprocessed datasets (Chronos) or out-of-memory errors (Sundial). Results for prediction length 720 are reported in Appendix Table 13.

| | UniTS-SFF | UniTS-FF | MOIRAI-SFF | MOIRAI-FF | Chronos-SFF | Chronos-FF | TTMs-SFF | TTMs-FF | Sundial-SFF | Sundial-FF |
|---|---|---|---|---|---|---|---|---|---|---|
| ETTh1 | **0.656** | 0.678 | **0.448** | 0.501 | **0.773** | 0.799 | **0.367** | 0.371 | **0.368** | 0.372 |
| ETTh2 | **0.364** | 0.374 | **0.32** | 0.321 | - | - | **0.275** | 0.278 | **0.293** | 0.31 |
| ETTm1 | **0.355** | 0.365 | **0.308** | 0.348 | **0.687** | 0.724 | **0.309** | 0.308 | **0.419** | 0.428 |
| ETTm2 | **0.179** | 0.184 | **0.181** | 0.184 | - | - | **0.17** | 0.178 | **0.182** | 0.195 |
| Weather | **0.171** | 0.198 | **0.166** | 0.173 | **1.137** | 1.276 | **0.157** | 0.159 | **0.179** | 0.186 |
| Elect. | **0.309** | 0.476 | **0.221** | 0.226 | **0.861** | 0.885 | **0.149** | 0.151 | - | - |
| Traffic | **0.877** | 1.195 | **0.476** | 0.497 | **0.831** | 0.834 | **0.453** | 0.462 | - | - |

Table 5: MSE comparison of SFF with linear probing (LP), and linear probing followed by full fine-tuning (LPFF) (Kumar et al.) on TSF with Timer and prediction length 96. Average performance is computed over each group of three data proportions. Full standard deviations and MAE results are reported in Tables 26 and 27. Detailed MSE, MAE, and standard deviations for individual proportions are provided in Appendix Tables 28, 29, 30, 31, 32, and 33.

| Data proportion | Avg. on 1%, 2%, 3% | | | Avg. on 4%, 5%, 10% | | | Avg. on 15%, 20%, 25% | | | Avg. on 50%, 75%, 100% | | |
|---|---|---|---|---|---|---|---|---|---|---|---|---|
| Methods | SFF | LP | LPFF | SFF | LP | LPFF | SFF | LP | LPFF | SFF | LP | LPFF |
| Exchange | **0.0856** | 0.5943 | 0.4801 | **0.0848** | 0.5906 | 0.4186 | **0.0816** | 0.563 | 0.1743 | **0.0812** | 0.474 | 0.0962 |
| Standard deviation | 4.4e-4 | 7.3e-3 | 3.9e-3 | 3.7e-4 | 7.2e-3 | 6.7e-3 | 6.1e-4 | 6.6e-3 | 7.4e-3 | 8.3e-4 | 4.7e-3 | 1.9e-3 |
| ETTh1 | **0.3722** | 0.8806 | 0.7171 | **0.3641** | 0.8594 | 0.6367 | **0.3523** | 0.7955 | 0.4127 | **0.3529** | 0.6356 | 0.3731 |
| ETTh2 | **0.28** | 0.4427 | 0.4026 | **0.278** | 0.4375 | 0.3707 | **0.2768** | 0.4234 | 0.3113 | **0.2758** | 0.3849 | 0.3001 |
| ETTm1 | **0.3448** | 1.046 | 0.7038 | **0.3162** | 1.0043 | 0.4772 | **0.301** | 0.8975 | 0.3245 | **0.2976** | 0.6608 | 0.3124 |
| ETTm2 | **0.1723** | 0.3555 | 0.3024 | **0.1663** | 0.3499 | 0.2559 | **0.1623** | 0.3297 | 0.1847 | **0.1616** | 0.2771 | 0.1804 |
| Weather | **0.1515** | 0.324 | 0.2478 | **0.146** | 0.3082 | 0.1741 | **0.1441** | 0.2699 | 0.1481 | **0.1453** | 0.2013 | 0.1565 |
| Electricity | **0.1346** | 0.6069 | 0.181 | **0.132** | 0.3242 | 0.1398 | **0.1305** | 0.2023 | 0.132 | **0.1301** | 0.1561 | 0.1335 |
| Traffic | **0.3678** | 0.9577 | 0.4081 | **0.3562** | 0.5999 | 0.3638 | **0.3494** | 0.4529 | 0.3572 | **0.3516** | 0.4079 | 0.3575 |

## 5.3 Compare SFF with other fine-tuning strategies

We compare SFF with popular fine-tuning strategies, including linear probing (LP) and linear probing followed by full fine-tuning (LPFF) (Kumar et al.). As reported in Table 5, SFF consistently outperforms both LP and LPFF across multiple data proportions, achieving average MSE reductions ranging from 7.17% to 41.57% relative to the best competitor, LPFF. These results demonstrate that SFF is a highly effective fine-tuning strategy, as it first smooths sharp regions of the loss landscape and subsequently enables improved trainability and more effective fine-tuning.

Due to limited space, comparisons with additional baselines, including label smoothing, SAM, SWA, Mixout, and L2-SP, are provided in Appendix Table 10. SFF consistently outperforms them.

## 5.4 DISCUSSION OF RETAINED PRETRAINING KNOWLEDGE AFTER SMOOTHING THE LOSS LANDSCAPE

**Impact of smoothing the loss landscape on convergence speed.** We record the test loss of different fine-tuning strategies at each epoch, as shown in Figure 4. Pretrained models capture general knowledge and can rapidly extract universal features from time series, thus requiring only a few fine-tuning steps to converge. As illustrated in Figure 4, both SFF and FF converge within the first epoch (highlighted by the red rectangles), indicating that SFF effectively preserves pretrained knowledge after smoothing the loss landscape via model interpolation. Moreover, SFF converges to a lower MSE than FF, demonstrating that the smoothing process further enhances model trainability, which aligns well with our design motivation.

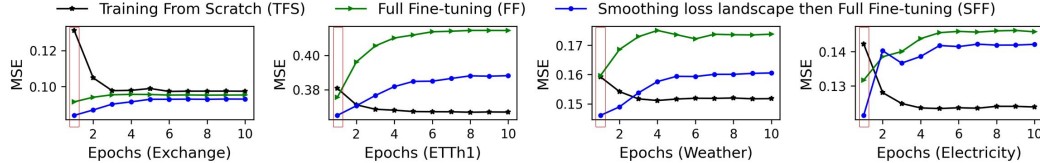

Figure 4: Test MSE over epochs for different fine-tuning methods on Timer in TSF with prediction length 96 across multiple datasets.

**Impact of smoothing the loss landscape on zero-shot forecasting.** Since pretrained knowledge is well preserved after smoothing, we further investigate its impact on zero-shot prediction. As shown in Table 6, the smoothed LTSM consistently improves zero-shot accuracy on seven datasets, achieving average gains of 6.13% for Timer and 35.75% for TimesFM. These results, averaged over multiple random seeds, indicate that smoothing guides the pretrained model toward a better local optimum. Furthermore, Figure 5 shows that an interpolation coefficient of $\alpha \approx 0.85$ yields the best zero-shot performance, striking a favorable balance between sufficient smoothing and effective preservation of pretrained knowledge. In addition, as reported in Table 7, smoothing also generally improves zero-shot accuracy on additional LTSMs, including MOIRAI, Chronos, TTMs, and Sundial, demonstrating the strong generalizability of our method.

Table 6: Zero-shot forecasting results with prediction length 96 before and after smoothing the loss landscape. They both outperform the "Re-initialize" version. MAE results are reported in Table 21.

| | ETTh1 | | ETTh2 | | ETTm1 | | ETTm2 | | Weather | | Electricity | | Traffic | |
| | Timer | +Smooth | Timer | +Smooth | Timer | +Smooth | Timer | +Smooth | Timer | +Smooth | Timer | +Smooth | Timer | +Smooth |
|---|---|---|---|---|---|---|---|---|---|---|---|---|---|---|
| MSE | 0.454 | **0.399** | 0.316 | **0.289** | 0.816 | **0.794** | 0.225 | **0.209** | 0.19 | **0.179** | 0.210 | **0.203** | 0.479 | **0.463** |
| Std. | $\pm 0$ | $\pm$1.3e-3 | $\pm 0$ | $\pm$2.2e-3 | $\pm 0$ | $\pm$1.2e-2 | $\pm 0$ | $\pm$1.6e-3 | $\pm 0$ | $\pm$9.0e-4 | $\pm 0$ | $\pm$6.2e-4 | $\pm 0$ | $\pm$6.6e-4 |

| | TimesFM | +Smooth | TimesFM | +Smooth | TimesFM | +Smooth | Tim.FM | +Smooth | Time.FM | +Smooth | Tim.FM | +Smooth | Tim.FM | +Smooth |
|---|---|---|---|---|---|---|---|---|---|---|---|---|---|---|
| MSE | 0.782 | **0.741** | 1.865 | **0.382** | 1.359 | **0.993** | 1.375 | **0.256** | 0.397 | **0.227** | 0.94 | **0.886** | 1.665 | **1.521** |
| Std. | $\pm 0$ | $\pm$7.9e-3 | $\pm 0$ | $\pm$4.8e-4 | $\pm 0$ | $\pm$2.4e-2 | $\pm 0$ | $\pm$5.8e-3 | $\pm 0$ | $\pm$4.3e-4 | $\pm 0$ | $\pm$3.6e-3 | $\pm 0$ | $\pm$1.3e-2 |

Figure 5: Impact of interpolation coefficient $\alpha$ on zero-shot forecasting with prediction length 96. The "Re-initialize" model performs the worst, which is reasonable.

## 5.5 HYPERPARAMETER SENSITIVITY ANALYSIS

**Influence of the interpolation coefficient $\alpha$.** As shown in Figure 6, although the interpolation coefficient $\alpha$ in SFF influences performance, SFF consistently achieves lower MSE than standard full fine-tuning (red dashed line) across a wide range of values on the Timer model. This further confirms that smoothing enhances trainability and enables pretrained LTSMs to exploit their pretrained knowledge better, thereby improving fine-tuning accuracy. Similarly, the zero-shot results in Figure 5 indicate that a broad range of $\alpha$ values also benefits Timer and TimesFM, further demonstrating the effectiveness and robustness of SFF.

Table 7: Zero-shot forecasting MSE of additional LTSMs with prediction length 96. Results for length 720 are reported in Appendix Table 14. UniTS is excluded as it targets few-shot learning.

|  | MOIRAI | +Smooth | Chronos | +Smooth | TTMs | +Smooth | Sundial | +Smooth |
|---|---|---|---|---|---|---|---|---|
| ETTh1 | 0.419 | **0.405** | 0.816 | **0.779** | 0.364 | **0.362** | 0.394 | **0.385** |
| ETTh2 | 0.305 | **0.295** | - | - | 0.277 | **0.275** | 0.306 | **0.303** |
| ETTm1 | 0.557 | **0.552** | 0.697 | **0.655** | 0.322 | **0.315** | 0.365 | **0.362** |
| ETTm2 | 0.227 | **0.219** | - | - | **0.171** | 0.172 | 0.2 | **0.191** |
| Weather | 0.192 | **0.189** | 1.259 | **1.087** | 0.158 | **0.158** | 0.175 | **0.173** |
| Elect. | 0.21 | **0.197** | 0.823 | **0.822** | **0.166** | 0.167 | - | - |
| Traffic | 0.555 | **0.544** | 0.854 | **0.836** | **0.514** | 0.516 | - | - |

Figure 6: Fine-tuning Timer for TSF with prediction length 96 (100% training data) under different interpolation coefficients $\alpha$ on the test set. ETTh12 denotes the average MSE of ETTh1 and ETTh2.

Table 8: MSE of fine-tuning pre-trained UniTS by adding different proportions of randomly initialized parameters. The percentages indicate the ratio of parameters subject to weight perturbation.

|  | ETTh1 | ETTh2 | ETTm1 | ETTm2 | Weather | Electricity | Traffic |
|---|---|---|---|---|---|---|---|
| Proportion (17.91%)-96 | 0.678 | 0.373 | 0.359 | 0.181 | 0.192 | 0.474 | 1.194 |
| Proportion (35.82%)-96 | 0.665 | 0.366 | 0.356 | 0.181 | 0.182 | 0.464 | 1.135 |
| Proportion (53.73%)-96 | 0.656 | 0.364 | 0.36 | 0.181 | 0.183 | 0.462 | 1.138 |
| Proportion (100%)-96 | 0.662 | 0.367 | 0.355 | 0.179 | 0.171 | 0.309 | 0.877 |
| Proportion (17.91%)-720 | 0.735 | 0.431 | 0.626 | 0.419 | 0.339 | 0.492 | 1.305 |
| Proportion (35.82%)-720 | 0.711 | 0.434 | 0.627 | 0.416 | 0.337 | 0.466 | 1.27 |
| Proportion (53.73%)-720 | 0.704 | 0.431 | 0.595 | 0.418 | 0.334 | 0.431 | 1.238 |
| Proportion (100%)-720 | 0.707 | 0.436 | 0.496 | 0.419 | 0.324 | 0.355 | 1.01 |

**Influence of the interpolation proportion of model parameters.** Notably, one of the key factors influencing SFF is the proportion of parameters being adjusted (i.e., interpolated). To verify this, we start from the first UniTS block and progressively increase the proportion of parameters undergoing weight interpolation, evaluating fine-tuning performance across datasets. As shown in Table 8, larger datasets benefit from higher interpolation ratios (MSE: 100% < 53.73% < 35.82% < 17.91% for Electricity and Traffic), whereas smaller datasets achieve better performance with lower ratios (MSE: 53.73% < 100% for ETTh1 and ETTh2). This is intuitive, as larger datasets support broader parameter updates that explore more non-convex regions and facilitate escape from sharp minima, while excessive interpolation on smaller datasets may lead to under-training and degraded performance.

# 6 CONCLUSION

In this work, we identify a key challenge: pretrained LSTMs may exhibit poor trainability during fine-tuning due to steep and unsmoothed loss landscapes, which limit the benefits of pretraining. We address this with a novel technology that first smooths the loss landscape without additional memory or computational overhead, followed by downstream fine-tuning, thereby improving trainability while preserving pretrained knowledge and achieving consistently better performance. Theoretically, SFF perturbs sharp minima without affecting inherently flat regions, enabling sharp minima escape from unfavorable basins. Extensive experiments on eight public datasets using eight pretrained LSTMs with diverse architectures and model sizes demonstrate the robustness of our method across different seeds and data regimes. We believe these insights may also generalize to broader pretrained-model fine-tuning scenarios.

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

# A   TECHNICAL APPENDICES AND SUPPLEMENTARY MATERIAL

## A.1   PROOF FOR THEOREMS ABOUT SHARP MINIMA AND FLAT MINIMA

We begin by performing a second-order Taylor expansion of the loss function $\mathcal{L}(\Theta^*)$ around the minimum point $\Theta^*$:

$$\begin{aligned} \mathcal{L}(\Theta^* + \delta) &\approx \mathcal{L}(\Theta^*) + \frac{1}{2}\delta^\top \nabla^2 \mathcal{L}(\Theta^*)\delta \\ &\approx \frac{1}{2}\delta^\top \nabla^2 \mathcal{L}(\Theta^*)\delta \end{aligned} \tag{9}$$

where $\delta$ is a small parameter perturbation (e.g., $|\delta| \ll 1$).

The loss change $\Delta\mathcal{L}$ after perturbation (interpolation) can be formulated as:

$$\Delta\mathcal{L} = \mathcal{L}(\Theta^* + \delta) - \mathcal{L}(\Theta^*) \approx \frac{1}{2}\delta^\top \nabla^2 \mathcal{L}(\Theta^*)\delta \approx \frac{1}{2}\delta^\top H\delta \tag{10}$$

The largest $\Delta\mathcal{L}$ is governed by the $\lambda_{\max}$ of the Hessian $H = \nabla^2 \mathcal{L}(\Theta^*)$.

Let the unit vector $v_{\max}$ be the eigenvector corresponding to the eigenvalue $\lambda_{\max}$. Perturbing in the direction of $v_{\max}$ yields the largest $\Delta\mathcal{L}$, because $v_{\max}$ represents the direction of maximum curvature (Foret et al., 2020), i.e., the steepest direction in the loss landscape. Let $\delta = \epsilon \cdot v_{\max}$ with $|\epsilon| \ll 1$. Then Eq. 10 can be reformulated as:

$$\Delta\mathcal{L} \approx \frac{1}{2}\epsilon \cdot v_{\max}^\top H\epsilon \cdot v_{\max} \tag{11}$$

According to the eigenvalue equation $Hv_{\max} = \lambda_{\max}v_{\max}$, we can further derive:

$$\Delta\mathcal{L} \approx \frac{1}{2}\epsilon^2 v_{\max}^\top \lambda_{\max}v_{\max} \approx \frac{1}{2}\epsilon^2 v_{\max}^\top v_{\max}\lambda_{\max} \approx \frac{1}{2}\epsilon^2 \lambda_{\max} \tag{12}$$

Hence, $\Delta\mathcal{L}$ is closely related to $\lambda_{\max}$: a larger $\lambda_{\max}$ corresponds to a sharper loss landscape, where perturbations along many directions can significantly increase the loss. Conversely, a smaller $\lambda_{\max}$ indicates a flatter landscape, in which perturbations have little effect on the loss (Hochreiter & Schmidhuber, 1997).

## A.2   RELATED WORKS

**Fine-tuning large time series models.** Most works focus on designing pre-training architecture and collecting large-scale time series data Das et al. (2024); Goswami et al. (2024); Woo et al. (2024b); Liu et al. (2024) to enrich and improve the foundations of LTSM. The fine-tuning technique has received relatively less attention in the field of large time series models. The traditional fine-tuning strategies are either *full fine-tuning* (adjusting all model parameters) or only fine-tuning the prediction head (also called *linear probing*). In the computer vision domain, Kumar et al. points out that full fine-tuning can achieve worse accuracy than linear probing in the condition of meeting out-of-distribution (OOD) data when the pretrained features are good and the distribution shift is large. To address this, they propose to linear probing first and then full fine-tuning (LP-FF) to improve the fine-tuning performance of the pre-trained model in the OOD condition. In our study, we also apply LP-FF to fine-tune the LTSM as a baseline to compare with our method.

Various optimization strategies can also be used for fine-tuning. Specifically, SAM (Sharpness-Aware Minimization) (Foret et al., 2020) updates parameters not only by the loss at the current point, but also by the flatness in a small neighborhood, so the optimizer is less likely to fall into high-curvature sharp minima. Note, however, that SAM needs one extra forward–backward pass, so its training cost is twice that of ordinary training. SWA (Stochastic Weight Averaging) (Izmailov et al., 2018) is an ensemble-like technology: it saves several weight snapshots taken after the model has converged and averages them to produce the final weights, reducing randomness and over-fitting. Mixout (Lee et al.) randomly replaces a subset of the fine-tuned weights with their pre-trained counterparts during training, mitigating catastrophic forgetting and over-fitting. L2-SP (Xuhong et al., 2018)

adds an L2 penalty between the current weights and the pre-trained weights to the loss, preventing the fine-tuned model from drifting too far away from the pre-trained solution and thus preserving pre-trained knowledge while curbing over-fitting.

**Large time series models (LTSM).** Existing efforts toward LTSMs can be categorized into two groups, with one being large language models for time series. FPT Zhou et al. (2023) partially fine-tunes GPT-2 on different downstream tasks. LLM4TS Chang et al. (2023) encodes time series into numerical tokens to utilize LLMs for time series forecasting. TimeLLM Jin et al. aligns the text prompt with time series to enhance prediction. These methods demonstrate the potential of LLMs for time series analysis. **Another category** includes pre-trained models on large-scale time series. Moirai Woo et al. (2024b), an encoder-only architecture, transforms time series into varied token sizes for better handling varied frequencies and then performs the pre-training strategy of mask modeling for time series forecasting. MOMENT Goswami et al. (2024), an encoder-decoder architecture, adopts a BERT-style mask modeling pre-training strategy and supports various downstream time series tasks. TimesFM Das et al. (2024) is a decoder-only Transformer pre-trained on Google Trends for forecasting, exhibiting notable zero-shot ability. Timer Liu et al. (2024) conducts GPT-style pre-training on the carefully processed and collected UTSD dataset and has achieved advanced accuracy on various tasks, including forecasting (Zhang et al., 2026b;c; 2025c), imputation Zhang et al. (2026a), and anomaly detection Liu et al. (2024); Zhang et al. (2025b).

## A.3    PYTORCH CODES TO SMOOTH THE LOSS LANDSCAPE OF THE PRETRAINED LTSM

We show the example codes to use the randomly initialized LTSM to smooth the loss landscape of the pretrained one in Algorithm 1. By combining the strengths of the randomly initialized LTSM (good trainability with a smoother loss landscape) and the pretrained LTSM (good pretrained knowledge), the convergence of the smoothed LTSM can be improved during fine-tuning.

---

**Algorithm 1:** Smoothing the loss landscape of the pre-trained LTSM for fine-tuning

```
def Smoothing_Landscape (model1,model2):
    """model1: pre-trained LTSM, model2: randomly initialized LTSM"""
    for param1, param2 in zip(model1.parameters(), model2.parameters()):
        # Smoothing the loss landscape of model1 through interpolation
        model1.copy_((model1 * alpha + model2 * (1 - alpha)))
    # Next, the smoothed model1 is applied for fine-tuning
    # without increasing memory and computational overhead
    return model1
```

---

## A.4    MORE DETAILS ABOUT REPRODUCING PAPER RESULTS

**Datasets.** In forecasting and imputation, we conduct extensive experiments on eight well-known datasets, including Exchange rate, Weather, Electricity, Traffic, and four ETT datasets (ETTh1, ETTh2, ETTm1, ETTm2). Details can be seen in Appendix A.5 and Table 9. **In the anomaly detection task,** following previous work Liu et al. (2024); Wu & Keogh (2021), we use UCR Anomaly Archive (containing 250 datasets) Wu & Keogh (2021) for anomaly detection.

**Evaluations.** Following Timer Liu et al. (2024), we uniformly use MSE (Mean Squared Error) and MAE (Mean Absolute Error) to evaluate the performance of methods on forecasting, imputation, and anomaly detection tasks. **In forecasting**, we investigate the performance of the proposed *smoothed fine-tuning* under different proportions of available fine-tuning data. The proportions range from 1% to 100%. The prediction length is 96 or 720. **In anomaly detection**, similar to Timer Liu et al. (2024), MSE is used as a confidence level to evaluate the effectiveness of anomaly detection. The higher the predicted MSE of the anomalous segments, the better, as this reduces the risk of normal segments being misjudged as anomalies.

**Implementation details.** The interpolation coefficient $\alpha$ is selected from 0.3, 0.5, 0.7, 0.9 for all tasks. For fairness, we use the source codes of each baseline and follow their recommended settings. Within each baseline, the configurations for "baseline" and "baseline-finetuning" are kept identical, ensuring that any performance change is attributable solely to the fine-tuning method (e.g., direct full FT vs. smoothed full FT). Settings may differ across baselines due to their public implementations,

so accuracies between baselines are not directly comparable. When the recommended input length causes out-of-memory issues (e.g., Sundial), we reduce it while still keeping "baseline" and "baseline-finetuning" consistent. All experiments run four random seeds with NVIDIA 3090 GPUs using PyTorch, reporting the mean and standard deviation.

Following Liu et al. (2024), the input and prediction lengths of Timer are fixed at 672 and 96. Based on the limited computing resources and settings supported by each model, the input lengths for MOMENT and TimesFM are 512 and 256, while the forecast lengths are 96, and 128, respectively. Following Timer Liu et al. (2024), the fine-tuning epochs are fixed at 10, and we report the best metric in all epochs. The learning rate is 3e-5 and the optimizer is Adam. More details can be found in our source code.

When implementing the baseline LTSMs, we download the weights from their official links, e.g., which are listed as follows:

- Timer's codes and pre-trained weights can be downloaded from the links[1,2].
- TimesFM's codes and pre-trained weights can be downloaded from the links[3,4]. The model weight path on Hugging Face is "google/timesfm-2.0-500m-pytorch". We adopt the latest 2.0 version.
- MOMENT's codes and pre-trained weights can be downloaded from the links[5,6]. The model weight path on Hugging Face is "AutonLab/MOMENT-1-large".

## A.5 More details about public datasets

The statistics of the eight well-known datasets are shown in Table 9, covering a range of variables and sampling frequency. These datasets involve applications in industrial machines, energy, and weather domains. They have been widely employed in the literature for time series analysis tasks Nie et al. (2023); Liu et al. (2024); Goswami et al. (2024); Woo et al. (2024b); Zhou et al. (2023); Chang et al. (2023); Jin et al..

Table 9: Statistics of eleven public datasets. *Data size* denotes the number of samples in train, validation, and test set for the single **variable**. In the experiment, each variable is separately split to construct samples and then merged, so the total number of samples needs to be multiplied by the number of variables. *Frequency* denotes the sampling interval of time points.

| Datasets | Variable | Data size (single variable) | Frequency |
|----------|----------|-----------------------------|-----------|
| ETTh1,ETTh2 | 7 | (8545, 2881, 2881) | Hourly |
| ETTm1,ETTm2 | 7 | (34465, 11521, 11521) | 15min |
| Weather | 21 | (36792, 5271, 10540) | 10min |
| Exchange rate | 9 | (5120, 665, 1422) | Daily |
| Electricity | 321 | (18317, 2633, 5261) | Hourly |
| Traffic | 862 | (12185, 1757, 3509) | Hourly |

These datasets used in this paper are extensively used for TSF algorithm evaluation, including exchange rate forecasting in the financial field, electricity consumption forecasting in the energy field, climate parameter forecasting in the weather domain, and machine parameter (e.g., loads and oil temperature) forecasting in the industrial field:

- Electricity dataset[7] collects the electricity consumption (kWh) every 15 minutes of 321 clients from 2012 to 2014.

---

[1] https://github.com/thuml/Large-Time-Series-Model?tab=readme-ov-file
[2] https://drive.google.com/drive/folders/15oaiAl4OO5gFqZMJD2lOtX2fxHbpgcU8
[3] https://github.com/google-research/timesfm
[4] https://huggingface.co/google/timesfm-2.0-500m-pytorch
[5] https://github.com/moment-timeseries-foundation-model/moment-research
[6] https://huggingface.co/AutonLab/MOMENT-1-large
[7] https://archive.ics.uci.edu/dataset/321/electricity

- ETT dataset[8] comprises two sub-datasets, ETT1 and ETT2, collected from two separate counties. Each sub-dataset offers two versions with varying sampling resolutions (15 minutes and 1 hour). ETT dataset includes multiple time series of electrical loads and a single time sequence of oil temperature.
- Weather dataset[9] contains 21 meteorological indicators, such as air temperature, humidity, etc, recorded every 10 minutes for the entirety of 2020.
- Traffic[10] dataset contains the occupation rate of freeway systems in California, USA. 5).

All datasets can be downloaded from the link[11].

## A.6 Additional experiment results and discussions

The experiments in the appendix serve as supplements to those in the main paper, including complete standard deviations, MAE results, and interpolation experiments. All figures and tables in the appendix have been appropriately linked and referenced in the main paper. They can be located by clicking the hyperlinks in the main paper while reading it.

We hope that these extensive experiments can help demonstrate that directly fine-tuning pre-trained LTSMs may indeed lead to limited performance, as they may overfit during pre-training, resulting in a steep and unsmoothed loss landscape and poor trainability, thereby degrading and limiting the fine-tuning performance of pre-trained LTSMs on downstream tasks. Meanwhile, Our proposed *smoothed finetuning* can indeed help pre-trained LTSMs achieve better fine-tuning performance on downstream tasks.

### A.6.1 Comparisons between SFF and more baselines

**Comparison with LoRA.** We additionally add LoRA fine-tuning as a baseline to highlight the contribution of our method. Since LTSM is mostly < 1 B parameters, we follow the official recommendation and set the low-rank factor r = 8. As reported in Table 10, our approach outperforms LoRA. This is reasonable: LoRA trades full fine-tuning for a low-rank constraint, achieving appealing parameter efficiency, yet this restriction can limit the model's fine-tuning capacity.

**Comparison with popular optimization strategies.** We further compare our method with several widely used optimization strategies during training. As shown in Table 10, while some of these approaches offer modest improvements over standard full fine-tuning, their performance still falls far short of that achieved by our proposed SFF fine-tuning. The key limitation is that they don't address the underlying problem, namely, the highly steep and non-smooth loss landscape of the pre-trained model. In contrast, SFF explicitly mitigates this problem by first smoothing the landscape and then fine-tuning, resulting in substantially stronger fine-tuning and adaptation performance.

**Influence of different parameter initialization schemes.** We have conducted ablations with several perturbation-based smoothing strategies: standard Gaussian noise (mean = 0, variance = 1), Xavier Gaussian, Xavier uniform, Kaiming Gaussian, and Kaiming uniform. Table 11 shows that Xavier- and Kaiming-based schemes maintain stable performance improvements. Because they consider the stable gradient variance and can supply a flat loss landscape (demonstrated by (Fort & Scherlis, 2019)) that is used to smooth the sharp landscape of the pre-trained model for better fine-tuning effect, which is also aligned with our Theoretical analysis. In contrast, standard Gaussian initialization without variance control often pushes parameters into sharper regions of the loss landscape, leading to weaker smoothing and noticeably degraded downstream performance. These findings provide strong empirical support for our design choice.

**Influence of random seeds on parameter initializations.** To assess sensitivity, we further evaluate SFF under different random seeds while using widely adopted initialization schemes (e.g., Kaiming uniform). The results in Table 12 indicate that improved performance remains highly stable across

---

[8]https://github.com/zhouhaoyi/Informer2020
[9]https://www.bgc-jena.mpg.de/wetter/
[10]http://pems.dot.ca.gov
[11]https://drive.google.com/drive/folders/1ZOYpTUa82_
jCcxIdTmyr0LXQfvaM9vIy

seeds. This suggests that SFF does not rely on a carefully engineered initialization. Instead, the mainstream initialization strategy suffices to obtain consistent smoothing and fine-tuning gains. This is reasonable because the prior work (Fort & Scherlis, 2019) has proven that the underlying design of mainstream initialization methods ensures the initialized parameters indeed lie in the flat region of the loss landscape, without being influenced by the random states (seeds). We believe this robustness is a desirable property for practical deployment.

Table 10: Comparison with more baselines on the LTSM Timer with prediction length 96. We independently run four times with four random seeds to enhance the solidity of the results and report the mean value and standard deviation.

| | Exchange | ETTh1 | ETTh2 | ETTm1 | ETTm2 | Weather |
|---|---|---|---|---|---|---|
| Original full fine-tuning | 0.09±0.0007 | 0.367±0.0027 | 0.304±0.0049 | 0.312±0.0008 | 0.176±0.0013 | 0.158±0.0012 |
| LoRA | 0.122±0.0003 | 0.418±0.0005 | 0.304±0.0006 | 0.401±0.0019 | 0.197±0.0001 | 0.155±0.0004 |
| Label-smoothing | 0.09±0.0018 | 0.364±0.0036 | 0.303±0.0043 | 0.312±0.0011 | 0.177±0.0013 | 0.158±0.0008 |
| SAM | 0.088±0.0016 | 0.362±0.003 | 0.296±0.0027 | 0.309±0.0002 | 0.175±0.0017 | 0.157±0.0 |
| SWA | 0.094±0.0017 | 0.366±0.0024 | 0.304±0.0045 | 0.319±0.0009 | 0.178±0.0014 | 0.162±0.0005 |
| MixOut | 0.09±0.0003 | 0.376±0.0006 | 0.297±0.0001 | 0.348±0.0018 | 0.184±0.0006 | 0.16±0.0007 |
| L2-SP | 0.09±0.0007 | 0.368±0.0031 | 0.304±0.0051 | 0.315±0.0007 | 0.177±0.0013 | 0.16±0.0004 |
| Ours (SFF) | **0.081**±0.0008 | **0.355**±0.0013 | **0.274**±0.0008 | **0.297**±0.0016 | **0.161**±0.0009 | **0.145**±0.0006 |

Table 11: The effectiveness of SFF (smoothed full fine-tuning) across different parameter initialization schemes on the LTSM Timer.

| | Exchange | ETTh1 | ETTh2 | ETTm1 | ETTm2 | Weather |
|---|---|---|---|---|---|---|
| Original full fine-tuning | 0.09±0.0007 | 0.367±0.0027 | 0.304±0.0049 | 0.312±0.0008 | 0.176±0.0013 | 0.158±0.0012 |
| Standard Gaussian perturbation-SFF | 5.986±0.261 | 0.723±0.003 | 1.879±0.275 | 3.876±0.128 | 19.031±0.963 | 0.447±0.03 |
| Kaiming Normal Distribution-SFF | 0.081±0.0006 | 0.353±0.0009 | 0.277±0.001 | 0.299±0.0011 | 0.164±0.0016 | 0.146±0.0008 |
| Kaiming Uniform Distribution-SFF | 0.081±0.0008 | 0.355±0.0013 | 0.274±0.0008 | 0.297±0.0016 | 0.161±0.0009 | 0.145±0.0006 |
| Xavier Normal Distribution-SFF | 0.081±0.0007 | 0.353±0.001 | 0.276±0.0012 | 0.3±0.0009 | 0.162±0.0001 | 0.145±0.0002 |
| Xavier Uniform Distribution-SFF | 0.082±0.0008 | 0.353±0.0007 | 0.277±0.0006 | 0.3±0.0003 | 0.162±0.0001 | 0.145±0.0002 |

Table 12: Experiments with four different random seeds (r1, r2,r3, and r4 here) on the LTSM Timer. The results show that SFF (smoothed full fine-tuning) is insensitive to the choice of random initialization distribution. FF denotes original full fine-tuning.

| | Exchange | ETTh1 | ETTh2 | ETTm1 | ETTm2 | Weather |
|---|---|---|---|---|---|---|
| SFF (Ours)-r1 | 0.07996 | 0.3547 | 0.27379 | 0.29542 | 0.16003 | 0.14605 |
| FF-r1 | 0.08937 | 0.36941 | 0.31021 | 0.31134 | 0.17645 | 0.16067 |
| SFF (Ours)-r2 | 0.08182 | 0.35772 | 0.27368 | 0.29902 | 0.16259 | 0.14432 |
| FF-r2 | 0.09071 | 0.36245 | 0.3035 | 0.31282 | 0.17843 | 0.15912 |
| SFF (Ours)-r3 | 0.08101 | 0.35766 | 0.27453 | 0.29824 | 0.16129 | 0.14481 |
| FF-r3 | 0.09102 | 0.36848 | 0.29699 | 0.31167 | 0.17515 | 0.15851 |
| SFF (Ours)-r4 | 0.08191 | 0.35588 | 0.27571 | 0.29938 | 0.16173 | 0.14525 |
| FF-r4 | 0.08978 | 0.36815 | 0.30666 | 0.31057 | 0.17546 | 0.15649 |

### A.6.2 GUIDANCE FOR SELECTING $\alpha$

In practice, we suggest the following guidance to select $\alpha$:

**(1) Empirically recommended values: Empirically Recommended Values:** As shown in Figure 6 and Figure 5 of the main text, while different values introduce some variation, the sensitivity analysis demonstrates that SFF consistently outperforms vanilla fine-tuning across a wide range. These values can thus serve as a reference and be prioritized in experimental setups.

**(2) Validation-based tuning:** We observe that the trend of test performance with respect to closely mirrors that on the validation set. Therefore, once a candidate search range is defined, selecting the that minimizes validation error provides a straightforward and computationally efficient strategy.

**(3) Data-driven selection:** Automatically learning is indeed a promising direction. In the current framework, however, the interpolation weights are fixed prior to fine-tuning, which makes adaptive selection non-trivial. Approaches such as meta-learning could potentially be explored to determine optimal values across models and datasets. We regard this as an important avenue for future research.

### A.6.3 Influence of SFF on normalization or scale between layers

We discuss this influence in two scenarios:

**(1) Fine-tuning after loss landscape smoothing:** When weights are smoothed, and the model is subsequently fine-tuned, the potential mismatch in normalization or scale is negligible. The model is free to update the relevant parameters during fine-tuning, effectively correcting any minor discrepancies introduced by interpolation.

**(2) Zero-shot forecasting after loss landscape smoothing:** First, the random initializations used for smoothing follow standard schemes (e.g., Kaiming, Xavier), whose typical scales are consistent with the learnable scale parameters in normalization layers. Second, the "flat" and "sharp" minima we analyze encompass the entire parameter space, including the weight matrices of normalization layers. Consequently, in theory, our method does not introduce significant scale or alignment inconsistencies. Instead, it smooths sharp minima located in suboptimal regions without harming flat minima, effectively relocating the sharp minima of the normalization layers to more favorable convergence points and thereby achieving improved and more generalizable performance. Moreover, we have included formal theoretical derivations and analyses demonstrating that the interpolation technology improves sharp minima while not harming flat minima.

Moreover, empirically, as shown in Tables 6 and 7, zero-shot forecasting following weight smoothing consistently demonstrates accuracy improvements, supporting the theoretical reasoning outlined above.

### A.6.4 More cases about the loss landscape of the pretrained LTSM (Figure 10 and Figure 8)

As shown in Figure 10 in main paper and Figure 7 and Figure 8 in the Appendix, we visualize the loss landscape of different datasets and empirically find that LTSMs initialized randomly typically have a smooth loss landscape. In contrast, pre-trained LTSMs consistently exhibit steep and non-smooth loss landscapes, indicating that this is not a random occurrence. Hence, after pre-training, LTSMs may indeed show lower trainability, which can affect their fine-tuning performance on downstream tasks.

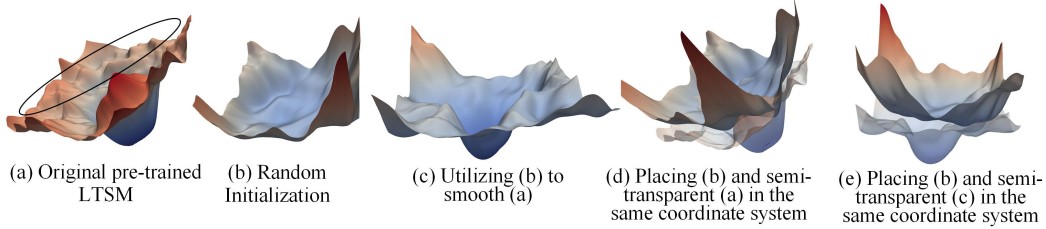

   (a) Original pre-trained    (b) Random    (c) Utilizing (b) to    (d) Placing (b) and semi-    (e) Placing (b) and semi-
   LTSM    Initialization    smooth (a)    transparent (a) in the    transparent (c) in the
   same coordinate system    same coordinate system

Figure 7: Loss landscape comparisons based on the LTSM Timer and weather dataset. The smoother the surface, the better.

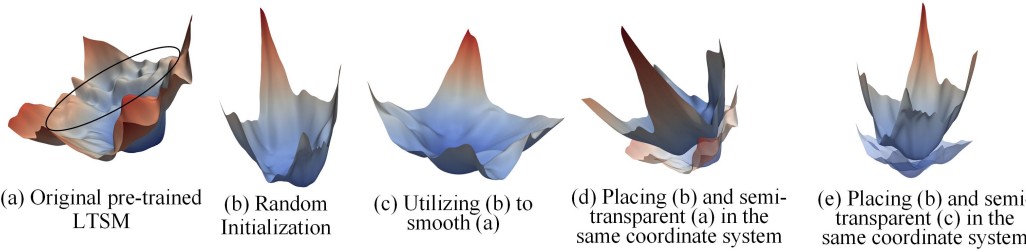

   (a) Original pre-trained    (b) Random    (c) Utilizing (b) to    (d) Placing (b) and semi-    (e) Placing (b) and semi-
   LTSM    Initialization    smooth (a)    transparent (a) in the    transparent (c) in the
   same coordinate system    same coordinate system

Figure 8: Loss landscape comparisons based on the LTSM Timer and electricity dataset. The smoother the surface, the better.

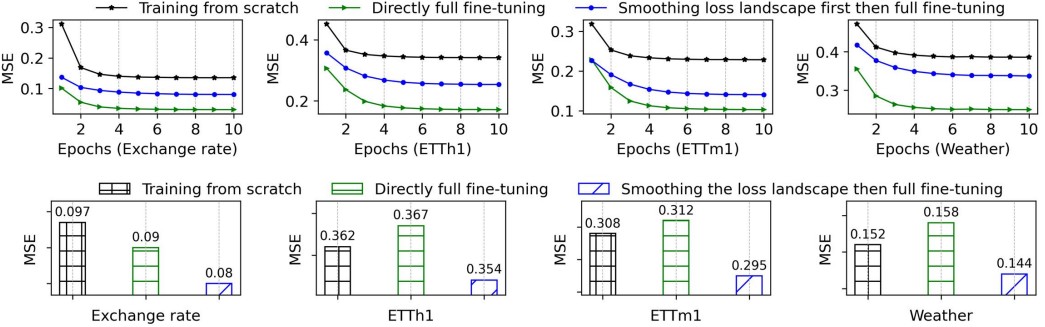

Figure 9: Time series forecasting of the LTSM Timer on various datasets with 100% available data proportion. We show the comparisons of training and testing losses for training from scratch on (black lines and bars), direct full fine-tuning (green lines and bars), and smoothing the loss landscape then full fine-tuning (blue lines and bars).

### A.6.5 TRAINING LOSS AND TEST LOSS DURING FINE-TUNING THE PRE-TRAINED LTSM (FIGURE 9)

We also empirically observe severe overfitting during the fine-tuning of pre-trained LTSM on downstream tasks, which is consistent with our analysis of the loss landscape. A steep and non-smooth loss landscape may cause the model to fall into poor local optima, leading to severe overfitting Li et al. (2018). Specifically, as shown in Figure 9, the training loss of directly fine-tuning the Timer (green lines) is significantly the lowest. However, the test MSE of the fine-tuned Timer (green bars) is even significantly worse than that of training from scratch on the Timer (black bars) on the downstream datasets (e.g., ETTh1, ETTm1, and Weather) without pre-training. This suggests that directly fine-tuning Timer leads to severe overfitting Hastie (2009), which causes pre-trained knowledge of it not to be fully utilized for improving the accuracy of downstream tasks.

### A.6.6 APPLYING *smoothed fine-tuning* FOR TIME SERIES IMPUTATION TASK (FIGURE 10)

As shown in Figure 10, our method also shows improvement for the imputation task. This indicates that the poor trainability of the LTSM, caused by overfitting during the pre-training phase, may impact their performance on various downstream time series tasks, including forecasting, anomaly detection, and imputation tasks. Our proposed method offers a potential solution to address this issue and provides a new perspective for fine-tuning the pretrained LTSMs.

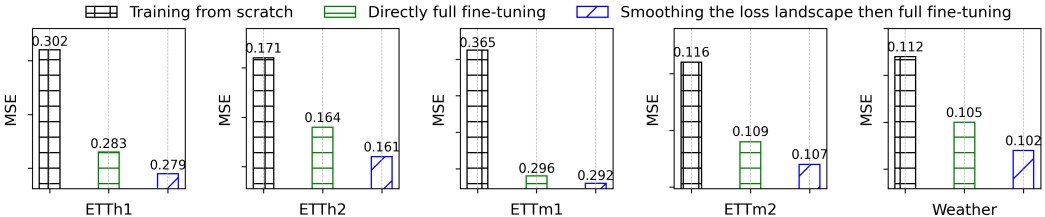

Figure 10: Time series imputation task on Timer with the mask ratio 25%. The experimental settings follow the Timer's codes: `https://github.com/thuml/Large-Time-Series-Model?tab=readme-ov-file.`

### A.6.7 SUMMARY OF THE CONTENT IN THE FOLLOWING SECTIONS

The experimental results in the following sections serve as supplements to the experiments in the main paper regarding complete standard deviations and MAE results under different available data proportions. Through multiple experiments with different random seeds, we ensure that our proposed fine-tuning method indeed helps the LTSM achieve better fine-tuning performance and that this is not a random occurrence. Moreover, our method consistently outperforms other fine-tuning methods

in terms of the MAE metric and across different data proportions, which further demonstrates the effectiveness of our approach. Our work provides new insights for fine-tuning large models. In the subsequent content, we provide relevant titles (in a table of contents format) for reference to the supplementary experimental results without further redundant explanations:

Table 13: MSE of fine-tuning more LTSMs for the TSF task with prediction length 720. SFF, and FF are *smoothed full fine-tuning* and *full fine-tuning*.

| | UniTS-SFF | UniTS-FF | MOIRAI-SFF | MOIRAI-FF | Chronos-SFF | Chronos-FF | TTMs-SFF | TTMs-FF | Sundial-SFF | Sundial-FF |
|---|---|---|---|---|---|---|---|---|---|---|
| ETTh1 | **0.704** | 0.741 | **0.582** | 0.634 | - | - | **0.421** | 0.424 | **0.485** | 0.49 |
| ETTh2 | **0.431** | 0.436 | **0.582** | 0.634 | - | - | **0.402** | 0.407 | **0.404** | 0.41 |
| ETTm1 | **0.496** | 0.625 | **0.402** | 0.451 | - | - | **0.433** | 0.435 | **0.552** | 0.562 |
| ETTm2 | **0.416** | 0.419 | **0.361** | 0.356 | - | - | **0.374** | 0.376 | **0.376** | 0.387 |
| Weather | **0.324** | 0.346 | **0.319** | 0.33 | - | - | **0.328** | 0.328 | **0.36** | 0.362 |
| Elect. | **0.355** | 0.495 | **0.843** | 0.999 | - | - | **0.241** | 0.24 | - | - |
| Traffic | **1.01** | 1.307 | **0.554** | 0.576 | - | - | **0.612** | 0.611 | - | - |

Table 14: Zero-shot forecasting MSE of more LTSMs with prediction length 720. "-" indicates that the preprocessed dataset is not included (Chronos) or out of memory (Sundial).

| | MOIRAI | +Smooth | Chronos | +Smooth | TTMs | +Smooth | Sundial | +Smooth |
|---|---|---|---|---|---|---|---|---|
| ETTh1 | **0.439** | 0.447 | - | - | **0.421** | 0.424 | **0.437** | 0.453 |
| ETTh2 | **0.386** | 0.4 | - | - | **0.404** | 0.408 | **0.409** | 0.417 |
| ETTm1 | **0.601** | 0.603 | - | - | **0.435** | 0.439 | **0.409** | 0.417 |
| ETTm2 | **0.407** | 0.438 | - | - | **0.409** | 0.41 | **0.397** | 0.408 |
| Weather | **0.399** | 0.424 | - | - | **0.328** | 0.328 | **0.354** | 0.357 |
| Elect. | **0.253** | 0.255 | - | - | **0.257** | 0.255 | - | - |
| Traffic | **0.62** | 0.622 | - | - | **0.624** | 0.622 | - | - |

Table 15: MSE of fine-tuning LTSM Timer for time series forecasting under **1%, 2%, 3% and 4%** proportion of available data. SFF, FF, and TFS are *smoothed full fine-tuning*, *full fine-tuning*, and *training from scratch*, respectively.

| Data proportion | 1% | | | 2% | | | 3% | | | 4% | | |
|---|---|---|---|---|---|---|---|---|---|---|---|---|
| Methods | SFF | FF | TFS | SFF | FF | TFS | SFF | FF | TFS | SFF | FF | TFS |
| Exchange | **0.0853** | 0.0887 | 0.2831 | **0.0858** | 0.0884 | 0.2808 | **0.0842** | 0.0876 | 0.2803 | **0.085** | 0.0879 | 0.2796 |
| Standard deviation | ±3.2e-4 | ±9.0e-4 | ±7.2e-3 | ±4.2e-4 | ±5.5e-4 | ±6.8e-3 | ±5.8e-4 | ±8.1e-4 | ±6.8e-3 | ±2.4e-4 | ±6.2e-4 | ±6.7e-3 |
| ETTh1 | **0.3649** | 0.3873 | 0.5631 | **0.3637** | 0.3861 | 0.5614 | **0.3608** | 0.3846 | 0.5588 | **0.3606** | 0.3823 | 0.545 |
| Standard deviation | ±6.3e-3 | ±1.2e-4 | ±9.8e-3 | ±6.6e-3 | ±3.6e-4 | ±1.1e-2 | ±6.6e-3 | ±7.6e-4 | ±1.1e-2 | ±7.5e-3 | ±1.4e-3 | ±4.7e-3 |
| ETTh2 | **0.2825** | 0.2945 | 0.363 | **0.2747** | 0.2893 | 0.362 | **0.2772** | 0.2896 | 0.3617 | **0.2752** | 0.2872 | 0.3434 |
| Standard deviation | ±1.0e-3 | ±4.4e-4 | ±2.0e-3 | ±1.7e-3 | ±1.5e-3 | ±2.3e-3 | ±1.9e-3 | ±7.0e-4 | ±2.7e-3 | ±2.7e-3 | ±2.1e-3 | ±2.3e-3 |
| ETTm1 | **0.364** | 0.3826 | 0.5342 | **0.3304** | 0.3484 | 0.4193 | **0.3252** | 0.3418 | 0.4159 | **0.3179** | 0.3312 | 0.3884 |
| Standard deviation | ±4.4e-3 | ±1.3e-3 | ±1.1e-2 | ±2.7e-3 | ±1.0e-3 | ±3.0e-3 | ±3.4e-3 | ±8.0e-4 | ±3.2e-3 | ±3.5e-3 | ±7.9e-4 | ±2.3e-3 |
| ETTm2 | **0.1709** | 0.1897 | 0.2531 | **0.173** | 0.184 | 0.2524 | **0.1635** | 0.1758 | 0.2157 | **0.1654** | 0.1755 | 0.216 |
| Standard deviation | ±2.4e-3 | ±1.7e-3 | ±3.2e-3 | ±2.1e-3 | ±7.4e-4 | ±3.4e-3 | ±2.4e-3 | ±3.4e-4 | ±1.7e-3 | ±1.4e-3 | ±6.1e-4 | ±1.5e-3 |
| Weather | **0.1537** | 0.1564 | 0.2403 | **0.1505** | 0.153 | 0.2259 | **0.1489** | 0.1525 | 0.2162 | **0.147** | 0.1492 | 0.2111 |
| Standard deviation | ±3.9e-4 | ±6.5e-4 | ±3.0e-3 | ±1.5e-4 | ±9.1e-4 | ±2.3e-3 | ±8.7e-4 | ±1.2e-3 | ±5.7e-4 | ±2.9e-4 | ±8.2e-4 | ±5.0e-4 |
| Electricity | **0.1369** | 0.1393 | 0.2285 | **0.1342** | 0.1367 | 0.2035 | **0.133** | 0.1356 | 0.1857 | **0.1331** | 0.1358 | 0.1708 |
| Standard deviation | ±7.8e-5 | ±7.6e-4 | ±1.5e-3 | ±1.9e-4 | ±7.1e-4 | ±1.4e-3 | ±3.3e-4 | ±6.9e-4 | ±1.3e-3 | ±2.5e-4 | ±7.4e-4 | ±6.4e-4 |
| Traffic | **0.3743** | 0.3768 | 0.5803 | **0.3671** | 0.3698 | 0.4794 | **0.3622** | 0.365 | 0.4388 | **0.3594** | 0.3623 | 0.4199 |
| Standard deviation | ±2.6e-4 | ±6.8e-4 | ±4.1e-3 | ±1.0e-4 | ±9.9e-4 | ±2.4e-3 | ±2.4e-4 | ±7.9e-4 | ±8.0e-4 | ±2.3e-4 | ±9.9e-4 | ±3.2e-4 |
| Avg. Improvements | - | 4.07% | 37.9 | - | 3.64% | 34.04 | - | 3.91% | 31.47 | - | 3.4% | 28.96 |
| Max. Improvements | - | 9.91% | 69.87 | - | 5.98% | 69.44 | - | 7.0% | 69.96 | - | 5.75% | 69.6 |

Table 16: MSE of fine-tuning LTSM Timer for time series forecasting under **5%, 10%, 15% and 20%** proportion of available data. SFF, FF, and TFS are *smoothed full fine-tuning*, *full fine-tuning*, and *training from scratch*, respectively.

| Data proportion | 5% | | | 10% | | | 15% | | | 20% | | |
|---|---|---|---|---|---|---|---|---|---|---|---|---|
| Methods | SFF | FF | TFS | SFF | FF | TFS | SFF | FF | TFS | SFF | FF | TFS |
| Exchange | **0.0854** | 0.0883 | 0.2721 | **0.0829** | 0.0854 | 0.1919 | **0.0815** | 0.0845 | 0.1715 | **0.0805** | 0.0858 | 0.157 |
| Standard deviation | ±1.6e-4 | ±8.4e-4 | ±6.4e-3 | ±7.2e-4 | ±6.9e-4 | ±1.7e-3 | ±3.1e-4 | ±9.4e-4 | ±9.1e-4 | ±1.1e-3 | ±1.4e-3 | ±2.5e-3 |
| ETTh1 | **0.3582** | 0.3745 | 0.4509 | **0.3539** | 0.3654 | 0.4162 | **0.3528** | 0.3615 | 0.3963 | **0.3483** | 0.3565 | 0.382 |
| Standard deviation | ±4.6e-3 | ±7.7e-4 | ±3.2e-3 | ±2.1e-3 | ±1.0e-3 | ±1.2e-3 | ±1.7e-3 | ±1.2e-3 | ±1.3e-3 | ±1.9e-3 | ±4.6e-4 | ±1.2e-3 |
| ETTh2 | **0.272** | 0.2866 | 0.3288 | **0.2757** | 0.2855 | 0.3157 | **0.2728** | 0.2854 | 0.3038 | **0.2765** | 0.2865 | 0.2929 |
| Standard deviation | ±3.5e-3 | ±1.4e-3 | ±1.2e-3 | ±2.2e-3 | ±6.2e-4 | ±7.0e-4 | ±3.8e-3 | ±1.4e-3 | ±1.3e-3 | ±1.5e-3 | ±1.1e-3 | ±3.7e-4 |
| ETTm1 | **0.3152** | 0.3273 | 0.385 | **0.3046** | 0.3115 | 0.3532 | **0.3016** | 0.3067 | 0.3428 | **0.2992** | 0.3059 | 0.3378 |
| Standard deviation | ±2.6e-3 | ±9.4e-4 | ±2.7e-3 | ±1.7e-3 | ±8.7e-4 | ±1.3e-3 | ±1.0e-3 | ±6.1e-4 | ±1.1e-3 | ±1.1e-3 | ±7.1e-4 | ±1.1e-3 |
| ETTm2 | **0.1637** | 0.1745 | 0.2151 | **0.1614** | 0.1714 | 0.1943 | **0.1614** | 0.1729 | 0.1846 | **0.162** | 0.1763 | 0.1793 |
| Standard deviation | ±2.9e-3 | ±9.4e-4 | ±1.4e-3 | ±1.8e-3 | ±4.9e-4 | ±5.6e-4 | ±1.3e-3 | ±1.1e-3 | ±2.9e-4 | ±7.8e-4 | ±6.7e-4 | ±3.7e-4 |
| Weather | **0.1459** | 0.1478 | 0.2024 | **0.1446** | 0.1464 | 0.1852 | **0.1442** | 0.1489 | 0.1739 | **0.1442** | 0.1466 | 0.166 |
| Standard deviation | ±1.6e-4 | ±6.5e-4 | ±3.6e-4 | ±1.2e-4 | ±4.9e-4 | ±2.0e-4 | ±1.2e-4 | ±1.2e-3 | ±8.3e-5 | ±2.9e-5 | ±2.1e-4 | ±2.6e-4 |
| Electricity | **0.1319** | 0.1346 | 0.1621 | **0.1309** | 0.1338 | 0.1461 | **0.1309** | 0.1342 | 0.1411 | **0.1306** | 0.1345 | 0.1378 |
| Standard deviation | ±2.6e-4 | ±7.0e-4 | ±4.4e-4 | ±1.8e-4 | ±5.2e-4 | ±2.0e-4 | ±2.3e-4 | ±5.5e-4 | ±1.4e-4 | ±1.3e-4 | ±7.8e-4 | ±8.5e-5 |
| Traffic | **0.3574** | 0.3604 | 0.4095 | **0.3518** | 0.3582 | 0.3874 | **0.3508** | 0.3596 | 0.3788 | **0.349** | 0.3579 | 0.373 |
| Standard deviation | ±1.2e-4 | ±9.4e-4 | ±2.7e-4 | ±8.5e-4 | ±7.1e-4 | ±1.6e-4 | ±1.1e-3 | ±8.6e-4 | ±1.0e-4 | ±1.7e-3 | ±4.4e-4 | ±1.8e-4 |
| Avg. Improvements | - | 3.34% | 25.97 | - | 2.84% | 19.58 | - | 3.34% | 16.24 | - | 3.66% | 13.63 |
| Max. Improvements | - | 6.19% | 68.61 | - | 5.83% | 56.8 | - | 6.65% | 52.48 | - | 8.11% | 48.73 |

Table 17: MSE of fine-tuning LTSM Timer for time series forecasting under **25%, 50%, 75% and 100%** proportion of available data. SFF, FF, and TFS are *smoothed full fine-tuning*, *full fine-tuning*, and *training from scratch*, respectively.

| Data proportion | 25% | | | 50% | | | 75% | | | 100% | | |
|---|---|---|---|---|---|---|---|---|---|---|---|---|
| Methods | SFF | FF | TFS | SFF | FF | TFS | SFF | FF | TFS | SFF | FF | TFS |
| Exchange | **0.0805** | 0.0865 | 0.1441 | **0.0802** | 0.0891 | 0.114 | **0.0802** | 0.0914 | 0.1026 | **0.08** | 0.091 | 0.0981 |
| Standard deviation | ±4.5e-4 | ±1.9e-4 | ±2.0e-3 | ±5.4e-4 | ±2.3e-3 | ±9.9e-4 | ±1.2e-3 | ±1.6e-3 | ±8.8e-4 | ±7.6e-4 | ±1.3e-4 | ±1.2e-3 |
| ETTh1 | **0.3506** | 0.355 | 0.3788 | **0.3494** | 0.3573 | 0.367 | **0.3493** | 0.358 | 0.3593 | **0.3547** | 0.3709 | 0.36 |
| Standard deviation | ±6.1e-4 | ±5.8e-4 | ±1.2e-3 | ±1.1e-3 | ±1.3e-3 | ±8.4e-4 | ±1.4e-3 | ±9.3e-4 | ±1.1e-3 | ±1.4e-3 | ±3.6e-3 | ±1.2e-3 |
| ETTh2 | **0.271** | 0.2866 | 0.2891 | **0.273** | 0.2905 | 0.2775 | **0.2772** | 0.3042 | 0.2796 | **0.2737** | 0.3117 | 0.2777 |
| Standard deviation | ±2.5e-3 | ±1.6e-3 | ±3.6e-4 | ±2.0e-3 | ±8.4e-4 | ±2.9e-4 | ±4.8e-4 | ±5.2e-4 | ±7.6e-4 | ±3.8e-4 | ±6.0e-3 | ±1.6e-3 |
| ETTm1 | **0.298** | 0.3049 | 0.333 | **0.2955** | 0.3069 | 0.3189 | **0.2956** | 0.3092 | 0.3116 | **0.2954** | 0.3128 | 0.3093 |
| Standard deviation | ±1.1e-3 | ±5.6e-4 | ±9.3e-4 | ±1.6e-3 | ±1.1e-3 | ±7.1e-4 | ±1.3e-3 | ±7.3e-4 | ±1.2e-3 | ±1.5e-3 | ±5.5e-4 | ±1.1e-3 |
| ETTm2 | **0.1594** | 0.1707 | 0.1741 | **0.1605** | 0.1718 | 0.1627 | **0.1623** | 0.1838 | 0.1651 | **0.16** | 0.1784 | 0.1644 |
| Standard deviation | ±9.7e-4 | ±1.3e-3 | ±2.8e-4 | ±3.8e-4 | ±5.9e-4 | ±1.8e-4 | ±4.3e-4 | ±2.5e-3 | ±9.3e-4 | ±1.0e-3 | ±1.4e-3 | ±1.1e-3 |
| Weather | **0.144** | 0.1472 | 0.1627 | **0.1441** | 0.1523 | 0.1538 | **0.1466** | 0.1665 | 0.1559 | **0.1443** | 0.1612 | 0.1526 |
| Standard deviation | ±5.2e-5 | ±6.1e-4 | ±5.7e-5 | ±2.1e-4 | ±7.4e-4 | ±4.2e-4 | ±1.3e-4 | ±1.5e-3 | ±1.1e-3 | ±7.3e-4 | ±1.6e-3 | ±9.2e-4 |
| Electricity | **0.1303** | 0.1344 | 0.1365 | **0.1301** | 0.1347 | 0.1327 | **0.13** | 0.1367 | 0.1326 | **0.1304** | 0.1344 | 0.1324 |
| Standard deviation | ±1.7e-4 | ±9.4e-4 | ±5.7e-5 | ±2.4e-4 | ±9.5e-4 | ±1.3e-4 | ±3.6e-4 | ±5.8e-4 | ±6.5e-4 | ±2.0e-4 | ±5.4e-4 | ±8.0e-4 |
| Traffic | **0.3488** | 0.3582 | 0.3688 | **0.3497** | 0.3586 | 0.3552 | **0.3478** | 0.361 | 0.3606 | **0.3551** | 0.3599 | 0.3609 |
| Standard deviation | ±2.0e-3 | ±9.2e-4 | ±2.0e-4 | ±1.5e-3 | ±5.8e-4 | ±1.5e-4 | ±3.2e-3 | ±1.1e-3 | ±3.0e-3 | ±2.7e-4 | ±2.2e-4 | ±2.5e-3 |
| Avg. Improvements | - | 3.79% | 12.28 | - | 4.97% | 6.82 | - | 7.52% | 5.47 | - | 7.41% | 4.64 |
| Max. Improvements | - | 6.94% | 44.14 | - | 9.99% | 29.65 | - | 12.25% | 21.83 | - | 12.19% | 18.45 |

Table 18: MAE of fine-tuning LTSM Timer for time series forecasting under **1%, 2%, 3% and 4%** proportion of available data. SFF, FF, and TFS are *smoothed full fine-tuning*, *full fine-tuning*, and *training from scratch*, respectively.

| Data proportion | 1% | | | 2% | | | 3% | | | 4% | | |
|---|---|---|---|---|---|---|---|---|---|---|---|---|
| Methods | SFF | FF | TFS | SFF | FF | TFS | SFF | FF | TFS | SFF | FF | TFS |
| Exchange | **0.2048** | 0.208 | 0.3946 | **0.2045** | 0.2083 | 0.3928 | **0.2036** | 0.2067 | 0.3923 | **0.2038** | 0.2071 | 0.3919 |
| Standard deviation | ±2.3e-4 | ±3.9e-4 | ±5.3e-3 | ±4.8e-4 | ±5.4e-4 | ±5.1e-3 | ±3.5e-4 | ±1.4e-4 | ±5.1e-3 | ±4.3e-4 | ±2.0e-4 | ±5.0e-3 |
| ETTh1 | **0.3973** | 0.4101 | 0.5202 | **0.3962** | 0.4085 | 0.5184 | **0.3943** | 0.4072 | 0.5168 | **0.3939** | 0.4044 | 0.5055 |
| Standard deviation | ±3.3e-3 | ±1.6e-4 | ±4.5e-3 | ±3.4e-3 | ±4.5e-5 | ±5.2e-3 | ±3.5e-3 | ±6.2e-5 | ±5.1e-3 | ±5.3e-3 | ±4.7e-4 | ±4.1e-3 |
| ETTh2 | **0.3362** | 0.3433 | 0.4109 | **0.3319** | 0.3401 | 0.41 | **0.3322** | 0.3394 | 0.4098 | **0.3339** | 0.3392 | 0.3944 |
| Standard deviation | ±2.0e-3 | ±1.1e-4 | ±1.8e-3 | ±3.2e-4 | ±5.4e-4 | ±2.0e-3 | ±5.2e-4 | ±6.5e-5 | ±2.4e-3 | ±5.2e-4 | ±1.8e-3 | ±2.7e-3 |
| ETTm1 | **0.4085** | 0.4166 | 0.5138 | **0.3838** | 0.3952 | 0.4507 | **0.3807** | 0.3909 | 0.4488 | **0.3755** | 0.3824 | 0.4303 |
| Standard deviation | ±2.5e-3 | ±3.8e-4 | ±6.0e-3 | ±2.6e-3 | ±4.8e-4 | ±1.9e-3 | ±1.9e-3 | ±3.8e-4 | ±2.0e-3 | ±1.8e-3 | ±1.6e-4 | ±1.7e-3 |
| ETTm2 | **0.2586** | 0.2738 | 0.3275 | **0.2591** | 0.2679 | 0.3271 | **0.2523** | 0.262 | 0.2979 | **0.2541** | 0.2609 | 0.2982 |
| Standard deviation | ±3.0e-3 | ±1.2e-3 | ±2.5e-3 | ±1.9e-3 | ±1.7e-4 | ±2.7e-3 | ±2.5e-3 | ±8.5e-4 | ±1.9e-3 | ±1.6e-3 | ±7.8e-5 | ±1.6e-3 |
| Weather | **0.2028** | 0.2062 | 0.2942 | **0.1996** | 0.2022 | 0.2808 | **0.1978** | 0.2015 | 0.2704 | **0.1954** | 0.1984 | 0.265 |
| Standard deviation | ±5.6e-4 | ±4.0e-4 | ±2.7e-3 | ±2.6e-4 | ±1.3e-4 | ±2.5e-3 | ±7.0e-4 | ±5.4e-4 | ±5.8e-4 | ±2.3e-4 | ±3.4e-5 | ±4.1e-4 |
| Electricity | **0.2342** | 0.2367 | 0.3243 | **0.2307** | 0.2334 | 0.2976 | **0.229** | 0.2315 | 0.2817 | **0.2292** | 0.2318 | 0.2702 |
| Standard deviation | ±8.3e-5 | ±6.8e-5 | ±1.4e-3 | ±3.4e-4 | ±8.8e-5 | ±1.2e-3 | ±5.0e-4 | ±5.1e-5 | ±9.6e-4 | ±2.7e-4 | ±8.6e-5 | ±5.5e-4 |
| Traffic | **0.2672** | 0.27 | 0.3921 | **0.2611** | 0.2636 | 0.3452 | **0.2575** | 0.26 | 0.3179 | **0.2559** | 0.2585 | 0.3037 |
| Standard deviation | ±2.0e-4 | ±2.4e-4 | ±1.7e-3 | ±2.1e-4 | ±1.9e-4 | ±1.4e-3 | ±6.8e-5 | ±7.4e-5 | ±5.0e-4 | ±9.9e-5 | ±2.5e-4 | ±2.0e-4 |
| Avg. Improvements | - | 2.25% | 27.77 | - | 2.1% | 25.24 | - | 2.12% | 23.22 | - | 1.72% | 21.26 |
| Max. Improvements | - | 5.55% | 48.1 | - | 3.28% | 47.94 | - | 3.7% | 48.1 | - | 2.61% | 48.0 |

Table 19: MAE of fine-tuning LTSM Timer for time series forecasting under **5%, 10%, 15% and 20%** proportion of available data. SFF, FF, and TFS are *smoothed full fine-tuning*, *full fine-tuning*, and *training from scratch*, respectively.

| Data proportion | 5% | | | 10% | | | 15% | | | 20% | | |
|---|---|---|---|---|---|---|---|---|---|---|---|---|
| Methods | SFF | FF | TFS | SFF | FF | TFS | SFF | FF | TFS | SFF | FF | TFS |
| Exchange | **0.2044** | 0.2078 | 0.3867 | **0.2019** | 0.2059 | 0.3252 | **0.2009** | 0.2047 | 0.3085 | **0.2008** | 0.2072 | 0.2936 |
| Standard deviation | ±2.0e-4 | ±1.7e-4 | ±4.9e-3 | ±9.0e-4 | ±1.8e-4 | ±1.7e-3 | ±6.1e-4 | ±1.8e-4 | ±1.2e-3 | ±1.0e-3 | ±1.5e-3 | ±2.9e-3 |
| ETTh1 | **0.3923** | 0.4011 | 0.462 | **0.3882** | 0.3952 | 0.4422 | **0.3884** | 0.3946 | 0.4274 | **0.3856** | 0.3915 | 0.4173 |
| Standard deviation | ±3.4e-3 | ±3.8e-4 | ±1.5e-3 | ±1.8e-3 | ±1.8e-4 | ±6.8e-4 | ±1.5e-3 | ±5.2e-4 | ±1.3e-3 | ±1.4e-3 | ±6.1e-5 | ±1.1e-3 |
| ETTh2 | **0.3325** | 0.3399 | 0.3826 | **0.3327** | 0.339 | 0.3686 | **0.3348** | 0.3385 | 0.3583 | **0.3332** | 0.3382 | 0.3512 |
| Standard deviation | ±1.2e-3 | ±9.5e-4 | ±1.9e-3 | ±2.3e-3 | ±9.5e-4 | ±9.4e-4 | ±1.7e-3 | ±8.9e-4 | ±2.1e-4 | ±2.6e-3 | ±7.9e-4 | ±1.9e-4 |
| ETTm1 | **0.3735** | 0.3801 | 0.4283 | **0.3663** | 0.3706 | 0.4054 | **0.3644** | 0.3678 | 0.3967 | **0.3631** | 0.3679 | 0.3926 |
| Standard deviation | ±1.3e-3 | ±1.9e-4 | ±2.0e-3 | ±8.2e-4 | ±4.5e-5 | ±1.1e-3 | ±3.1e-4 | ±7.6e-5 | ±9.8e-4 | ±8.2e-4 | ±2.5e-5 | ±1.0e-3 |
| ETTm2 | **0.2522** | 0.2599 | 0.2976 | **0.2516** | 0.2584 | 0.2787 | **0.2532** | 0.2576 | 0.2707 | **0.2527** | 0.2622 | 0.2672 |
| Standard deviation | ±1.6e-3 | ±8.6e-5 | ±1.6e-3 | ±1.3e-3 | ±5.0e-4 | ±6.1e-4 | ±5.0e-4 | ±2.9e-4 | ±2.0e-4 | ±4.1e-4 | ±4.0e-4 | ±2.5e-4 |
| Weather | **0.1936** | 0.1968 | 0.257 | **0.1928** | 0.1967 | 0.2385 | **0.1928** | 0.1978 | 0.2271 | **0.1932** | 0.1988 | 0.2198 |
| Standard deviation | ±2.3e-4 | ±2.2e-5 | ±5.3e-4 | ±1.4e-4 | ±5.3e-4 | ±7.3e-5 | ±1.3e-4 | ±6.9e-4 | ±2.9e-5 | ±3.0e-4 | ±1.2e-3 | ±3.8e-4 |
| Electricity | **0.2273** | 0.2303 | 0.2616 | **0.2263** | 0.2319 | 0.2444 | **0.226** | 0.2312 | 0.2384 | **0.225** | 0.23 | 0.2346 |
| Standard deviation | ±9.1e-5 | ±5.4e-5 | ±4.5e-4 | ±3.2e-4 | ±1.4e-3 | ±3.0e-4 | ±1.8e-4 | ±1.1e-3 | ±2.1e-4 | ±4.5e-4 | ±5.0e-4 | ±1.4e-4 |
| Traffic | **0.2537** | 0.2571 | 0.2948 | **0.2501** | 0.2554 | 0.2754 | **0.2488** | 0.2573 | 0.2678 | **0.2473** | 0.2573 | 0.2631 |
| Standard deviation | ±6.5e-5 | ±1.8e-4 | ±2.9e-4 | ±9.1e-4 | ±9.1e-5 | ±2.3e-4 | ±1.3e-3 | ±1.9e-3 | ±1.1e-4 | ±1.5e-3 | ±1.2e-3 | ±1.8e-4 |
| Avg. Improvements | - | 1.87% | 19.39 | - | 1.98% | 14.37 | - | 1.9% | 11.57 | - | 2.48% | 9.94 |
| Max. Improvements | - | 2.96% | 47.14 | - | 2.63% | 37.92 | - | 3.3% | 34.88 | - | 3.89% | 31.61 |

Table 20: MAE of fine-tuning LTSM Timer for time series forecasting under **25%, 50%, 75% and 100%** proportion of available data. SFF, FF, and TFS are *smoothed full fine-tuning*, *full fine-tuning*, and *training from scratch*, respectively.

| Data proportion | 25% | | | 50% | | | 75% | | | 100% | | |
|---|---|---|---|---|---|---|---|---|---|---|---|---|
| Methods | SFF | FF | TFS | SFF | FF | TFS | SFF | FF | TFS | SFF | FF | TFS |
| Exchange | **0.1997** | 0.2095 | 0.2782 | **0.1993** | 0.2122 | 0.2421 | **0.2002** | 0.2136 | 0.2266 | **0.2006** | 0.2153 | 0.2209 |
| Standard deviation | ±5.3e-4 | ±5.3e-4 | ±2.6e-3 | ±1.0e-3 | ±2.6e-3 | ±1.2e-3 | ±1.7e-3 | ±7.8e-4 | ±1.2e-4 | ±5.3e-4 | ±1.1e-3 | ±4.0e-4 |
| ETTh1 | **0.3879** | 0.3902 | 0.4145 | **0.388** | 0.3905 | 0.404 | **0.3858** | 0.3907 | 0.3956 | **0.3921** | 0.3955 | 0.399 |
| Standard deviation | ±4.5e-4 | ±5.3e-4 | ±9.8e-4 | ±3.0e-4 | ±2.4e-4 | ±6.6e-4 | ±2.4e-3 | ±5.0e-4 | ±5.5e-4 | ±1.4e-3 | ±1.1e-3 | ±7.6e-4 |
| ETTh2 | **0.3325** | 0.337 | 0.3467 | **0.3327** | 0.3421 | 0.3378 | **0.3387** | 0.3516 | 0.3435 | **0.3353** | 0.3531 | 0.3436 |
| Standard deviation | ±1.3e-3 | ±4.7e-4 | ±1.4e-4 | ±1.4e-3 | ±7.4e-4 | ±1.9e-4 | ±1.8e-4 | ±1.1e-3 | ±7.8e-4 | ±5.7e-4 | ±1.2e-3 | ±2.4e-3 |
| ETTm1 | **0.3599** | 0.3656 | 0.3877 | **0.3576** | 0.3667 | 0.374 | **0.3566** | 0.3698 | 0.3696 | **0.3558** | 0.3703 | 0.3669 |
| Standard deviation | ±1.7e-3 | ±1.7e-4 | ±8.7e-4 | ±1.8e-3 | ±1.0e-4 | ±6.1e-4 | ±2.0e-3 | ±5.1e-4 | ±4.9e-4 | ±1.6e-3 | ±2.8e-4 | ±4.9e-4 |
| ETTm2 | **0.249** | 0.2562 | 0.2626 | **0.2497** | 0.2566 | 0.253 | **0.2534** | 0.2635 | 0.2571 | **0.2472** | 0.2591 | 0.2533 |
| Standard deviation | ±6.4e-4 | ±4.6e-4 | ±2.4e-4 | ±5.8e-4 | ±5.6e-4 | ±1.6e-4 | ±3.8e-4 | ±1.5e-3 | ±6.4e-4 | ±1.5e-3 | ±8.1e-4 | ±5.7e-4 |
| Weather | **0.1921** | 0.1961 | 0.2146 | **0.1929** | 0.199 | 0.2037 | **0.1971** | 0.217 | 0.2098 | **0.192** | 0.2046 | 0.203 |
| Standard deviation | ±2.9e-4 | ±4.3e-4 | ±9.0e-5 | ±2.1e-4 | ±1.5e-3 | ±1.5e-4 | ±2.9e-4 | ±1.5e-3 | ±7.3e-4 | ±1.5e-3 | ±1.1e-3 | ±6.1e-4 |
| Electricity | **0.2245** | 0.2289 | 0.2328 | **0.2247** | 0.2273 | 0.228 | **0.2245** | 0.2292 | 0.2274 | **0.2239** | 0.2273 | 0.2268 |
| Standard deviation | ±4.2e-4 | ±8.4e-4 | ±1.5e-4 | ±2.7e-4 | ±2.3e-4 | ±9.0e-5 | ±4.3e-4 | ±4.4e-4 | ±2.1e-5 | ±4.2e-4 | ±3.3e-4 | ±2.2e-4 |
| Traffic | **0.2463** | 0.2545 | 0.2599 | **0.2486** | 0.2527 | 0.2512 | **0.2485** | 0.2538 | 0.2573 | **0.2444** | 0.2517 | 0.2553 |
| Standard deviation | ±1.9e-3 | ±1.8e-3 | ±1.5e-4 | ±1.4e-3 | ±1.3e-3 | ±1.6e-4 | ±4.6e-4 | ±9.0e-4 | ±2.6e-3 | ±3.0e-4 | ±2.0e-3 | ±1.3e-3 |
| Avg. Improvements | - | 2.27% | 8.8 | - | 2.56% | 4.58 | - | 3.99% | 3.9 | - | 3.97% | 3.72 |
| Max. Improvements | - | 4.68% | 28.22 | - | 6.08% | 17.68 | - | 9.17% | 11.65 | - | 6.83% | 9.19 |

Table 21: MAE of Smoothing the loss landscape then perform zero-shot forecasting.

| | ETTh1 | | ETTh2 | | ETTm1 | | ETTm2 | | Weather | | Electricity | | Traffic | |
|---|---|---|---|---|---|---|---|---|---|---|---|---|---|---|
| | Timer | +Smooth | Timer | +Smooth | Timer | +Smooth | Timer | +Smooth | Timer | +Smooth | Timer | +Smooth | Timer | +Smooth |
| MAE | 0.434 | **0.418** | 0.359 | **0.352** | 0.61 | **0.607** | 0.3 | **0.294** | 0.236 | **0.231** | 0.312 | **0.308** | 0.343 | **0.335** |
| Std. | ±0 | ±5.0e-4 | ±0 | ±1.1e-3 | ±0 | ±2.7e-3 | ±0 | ±1.2e-3 | ±0 | ±1.1e-3 | ±0 | ±5.4e-4 | ±0 | ±5.2e-4 |
| Imp. | - | 3.69% | - | 1.95% | - | 0.49% | - | 2.0% | - | 2.12% | - | 1.28% | - | 2.33% |
| | TimesFM | +Smooth | TimesFM | +Smooth | TimesFM | +Smooth | Tim.FM | +Smooth | Time.FM | +Smooth | Tim.FM | +Smooth | Tim.FM | +Smooth |
| MAE | 0.559 | **0.55** | 0.541 | **0.419** | 0.749 | **0.682** | 0.404 | **0.335** | 0.278 | **0.262** | 0.756 | **0.747** | 0.867 | **0.834** |
| Std. | ±0 | ±1.4e-3 | ±0 | ±2.1e-3 | ±0 | ±3.0e-3 | ±0 | ±8.8e-4 | ±0 | ±3.8e-3 | ±0 | ±4.0e-3 | ±0 | ±4.0e-3 |
| Imp. | - | 1.61% | - | 22.55% | - | 8.95% | - | 17.08% | - | 5.76% | - | 1.19% | - | 3.81% |

Table 22: The complete anomaly detection results on 250 datasets, reporting the average MSE values of anomalous segments in each dataset under four random seeds, where higher values are better, because this reduces the risk of normal segments being misjudged as anomalies.

| Index | 1 | 2 | 3 | 4 | 5 | 6 | 7 | 8 | 9 | 10 | 11 | 12 | 13 | 14 | 15 |
|---|---|---|---|---|---|---|---|---|---|---|---|---|---|---|---|
| 1 (SFF) | 0.051 | 0.01 | 0.004 | 0.435 | 0.011 | 0.09 | 0.112 | 0.016 | 0.046 | 0.166 | 0.129 | 0.045 | 1.112 | 0.042 | 0.13 |
| 1 (FF) | 0.031 | 0.003 | 0.002 | 0.706 | 0.004 | 0.002 | 0.039 | 0.006 | 0.005 | 0.003 | 0.055 | 0.033 | 0.071 | 0.024 | 0.096 |
| 1 (TFS) | 0.019 | 0.005 | 0.003 | 0.262 | 0.005 | 0.003 | 0.032 | 0.009 | 0.028 | 0.002 | 0.139 | 0.008 | 0.652 | 0.034 | 0.1 |
| 2 (SFF) | 0.107 | 0.077 | 0.002 | 0.164 | 0.005 | 0.002 | 0.093 | 0.004 | 0.082 | 0.065 | 0.071 | 0.053 | 0.27 | 0.324 | 0.043 |
| 2 (FF) | 0.012 | 0.024 | 0.0 | 0.11 | 0.005 | 0.0 | 0.036 | 0.002 | 0.024 | 0.038 | 0.069 | 0.013 | 0.231 | 0.213 | 0.023 |
| 2 (TFS) | 0.012 | 0.026 | 0.001 | 0.103 | 0.006 | 0.001 | 0.055 | 0.002 | 0.043 | 0.049 | 0.02 | 0.02 | 0.13 | 0.232 | 0.028 |
| 3 (SFF) | 0.024 | 0.059 | 0.007 | 0.138 | 0.203 | 0.583 | 0.085 | 0.04 | 0.314 | 0.238 | 0.118 | 0.24 | 0.004 | 0.07 | 0.029 |
| 3 (FF) | 0.001 | 0.014 | 0.003 | 0.092 | 0.114 | 0.206 | 0.038 | 0.01 | 0.297 | 0.195 | 0.1 | 0.096 | 0.002 | 0.05 | 0.011 |
| 3 (TFS) | 0.001 | 0.013 | 0.004 | 0.094 | 0.124 | 0.329 | 0.05 | 0.026 | 0.081 | 0.117 | 0.108 | 0.144 | 0.001 | 0.05 | 0.0 |
| 4 (SFF) | 0.121 | 0.32 | 0.106 | 0.085 | 0.092 | 0.348 | 0.512 | 0.129 | 0.54 | 0.122 | 0.027 | 0.413 | 0.36 | 0.033 | 0.006 |
| 4 (FF) | 0.026 | 0.095 | 0.073 | 0.013 | 0.016 | 0.062 | 0.259 | 0.003 | 0.223 | 0.013 | 0.026 | 0.396 | 0.352 | 0.005 | 0.003 |
| 4 (TFS) | 0.087 | 0.184 | 0.175 | 0.031 | 0.079 | 0.095 | 0.323 | 0.005 | 0.213 | 0.012 | 0.015 | 0.212 | 0.218 | 0.004 | 0.003 |
| 5 (SFF) | 0.912 | 1.184 | 0.043 | 0.255 | 0.017 | 0.072 | 0.093 | 0.042 | 0.074 | 0.032 | 0.062 | 0.331 | 0.255 | 0.058 | 0.021 |
| 5 (FF) | 0.618 | 0.078 | 0.014 | 0.114 | 0.001 | 0.038 | 0.029 | 0.014 | 0.026 | 0.014 | 0.007 | 0.224 | 0.007 | 0.063 | 0.003 |
| 5 (TFS) | 0.602 | 0.133 | 0.036 | 0.113 | 0.0 | 0.029 | 0.053 | 0.029 | 0.019 | 0.015 | 0.02 | 0.238 | 0.004 | 0.019 | 0.013 |
| 6 (SFF) | 0.29 | 0.015 | 0.034 | 0.142 | 0.105 | 0.497 | 0.352 | 0.134 | 0.013 | 0.021 | 0.141 | 0.002 | 0.14 | 0.011 | 0.068 |
| 6 (FF) | 0.334 | 0.004 | 0.011 | 0.12 | 0.007 | 0.432 | 0.156 | 0.002 | 0.005 | 0.006 | 0.14 | 0.001 | 0.135 | 0.006 | 0.002 |
| 6 (TFS) | 0.158 | 0.005 | 0.007 | 0.074 | 0.007 | 0.169 | 0.051 | 0.001 | 0.002 | 0.001 | 0.081 | 0.0 | 0.104 | 0.007 | 0.003 |
| 7 (SFF) | 0.345 | 0.003 | 0.013 | 0.039 | 1.293 | 0.131 | 0.055 | 0.318 | 0.06 | 0.111 | 0.097 | 0.026 | 0.82 | 0.198 | 0.314 |
| 7 (FF) | 0.028 | 0.001 | 0.006 | 0.025 | 0.779 | 0.063 | 0.005 | 0.065 | 0.016 | 0.11 | 0.031 | 0.006 | 0.583 | 0.085 | 0.055 |
| 7 (TFS) | 0.13 | 0.001 | 0.014 | 0.037 | 0.159 | 0.114 | 0.036 | 0.332 | 0.015 | 0.073 | 0.026 | 0.006 | 0.267 | 0.078 | 0.094 |
| 8 (SFF) | 0.182 | 0.007 | 0.119 | 0.135 | 0.182 | 0.004 | 0.172 | 0.233 | 0.073 | 0.005 | 0.117 | 0.081 | 0.045 | 0.518 | 0.014 |
| 8 (FF) | 0.155 | 0.004 | 0.036 | 0.058 | 0.201 | 0.003 | 0.07 | 0.157 | 0.051 | 0.007 | 0.04 | 0.04 | 0.038 | 0.41 | 0.008 |
| 8 (TFS) | 0.105 | 0.009 | 0.111 | 0.142 | 0.145 | 0.003 | 0.135 | 0.073 | 0.046 | 0.001 | 0.105 | 0.039 | 0.036 | 0.402 | 0.014 |
| 9 (SFF) | 2.133 | 0.067 | 0.126 | 0.118 | 0.151 | 0.728 | 0.213 | 0.174 | 0.141 | 0.035 | 0.07 | 0.082 | 0.05 | 0.471 | 0.008 |
| 9 (FF) | 0.795 | 0.03 | 0.009 | 0.026 | 0.072 | 0.458 | 0.067 | 0.151 | 0.077 | 0.008 | 0.035 | 0.029 | 0.006 | 0.478 | 0.002 |
| 9 (TFS) | 1.539 | 0.031 | 0.096 | 0.03 | 0.144 | 0.437 | 0.079 | 0.089 | 0.08 | 0.024 | 0.043 | 0.055 | 0.027 | 0.203 | 0.002 |
| 10 (SFF) | 0.066 | 0.031 | 0.478 | 0.029 | 0.055 | 0.004 | 0.087 | 0.069 | 0.202 | 0.423 | 0.835 | 0.103 | 0.08 | 0.981 | 0.085 |
| 10 (FF) | 0.028 | 0.015 | 0.094 | 0.004 | 0.014 | 0.001 | 0.043 | 0.02 | 0.017 | 0.135 | 0.431 | 0.008 | 0.056 | 0.255 | 0.008 |
| 10 (TFS) | 0.07 | 0.027 | 0.167 | 0.005 | 0.06 | 0.001 | 0.045 | 0.014 | 0.043 | 0.273 | 0.462 | 0.008 | 0.012 | 0.382 | 0.005 |
| 11 (SFF) | 0.054 | 0.073 | 0.485 | 0.005 | 0.143 | 0.005 | 0.054 | 0.176 | 0.003 | 1.351 | 0.048 | 0.007 | 0.003 | 0.046 | 0.596 |
| 11 (FF) | 0.014 | 0.057 | 0.453 | 0.003 | 0.049 | 0.023 | 0.031 | 0.143 | 0.002 | 0.077 | 0.009 | 0.002 | 0.001 | 0.02 | 0.158 |
| 11 (TFS) | 0.049 | 0.05 | 0.35 | 0.001 | 0.017 | 0.001 | 0.027 | 0.108 | 0.002 | 0.115 | 0.007 | 0.003 | 0.005 | 0.039 | 0.371 |
| 12 (SFF) | 0.511 | 0.026 | 0.184 | 0.082 | 0.085 | 0.26 | 0.637 | 0.221 | 0.092 | 0.096 | 0.13 | 0.014 | 0.002 | 0.011 | 0.137 |
| 12 (FF) | 0.349 | 0.016 | 0.137 | 0.016 | 0.006 | 0.281 | 0.091 | 0.085 | 0.026 | 0.102 | 0.132 | 0.013 | 0.001 | 0.003 | 0.059 |
| 12 (TFS) | 0.126 | 0.025 | 0.107 | 0.022 | 0.159 | 0.034 | 0.297 | 0.071 | 0.04 | 0.057 | 0.07 | 0.012 | 0.002 | 0.004 | 0.091 |
| 13 (SFF) | 0.045 | 0.067 | 0.003 | 0.005 | 0.075 | 0.001 | 0.005 | 0.045 | 0.16 | 0.172 | 0.034 | 0.022 | 0.204 | 0.023 | 0.033 |
| 13 (FF) | 0.026 | 0.025 | 0.001 | 0.002 | 0.01 | 0.002 | 0.002 | 0.028 | 0.01 | 0.015 | 0.021 | 0.018 | 0.118 | 0.001 | 0.015 |
| 13 (TFS) | 0.019 | 0.038 | 0.001 | 0.004 | 0.006 | 0.002 | 0.002 | 0.021 | 0.014 | 0.005 | 0.034 | 0.016 | 0.162 | 0.005 | 0.03 |
| 14 (SFF) | 0.98 | 0.579 | 0.012 | 0.022 | 0.014 | 0.003 | 0.728 | 0.02 | 0.002 | 0.354 | 0.042 | 0.023 | 0.055 | 0.068 | 0.132 |
| 14 (FF) | 0.248 | 0.226 | 0.003 | 0.009 | 0.006 | 0.001 | 0.683 | 0.009 | 0.0 | 0.244 | 0.041 | 0.018 | 0.032 | 0.059 | 0.074 |
| 14 (TFS) | 0.166 | 0.406 | 0.01 | 0.009 | 0.011 | 0.002 | 0.803 | 0.008 | 0.0 | 0.43 | 0.023 | 0.009 | 0.053 | 0.041 | 0.039 |
| 15 (SFF) | 0.007 | 0.052 | 0.094 | 0.471 | 0.19 | 0.003 | 0.182 | 0.522 | 0.193 | 0.003 | 0.028 | 0.31 | 0.005 | 0.006 | 0.356 |
| 15 (FF) | 0.004 | 0.013 | 0.072 | 0.263 | 0.015 | 0.002 | 0.098 | 0.315 | 0.038 | 0.001 | 0.016 | 0.183 | 0.001 | 0.0 | 0.073 |
| 15 (TFS) | 0.007 | 0.014 | 0.001 | 0.447 | 0.035 | 0.001 | 0.114 | 0.186 | 0.182 | 0.003 | 0.023 | 0.222 | 0.001 | 0.0 | 0.222 |
| 16 (SFF) | 0.003 | 0.283 | 0.335 | 0.013 | 0.141 | 0.185 | 0.165 | 0.638 | 1.156 | 0.14 | 0.009 | 0.043 | 0.105 | 0.015 | 0.013 |
| 16 (FF) | 0.001 | 0.175 | 0.275 | 0.03 | 0.049 | 0.129 | 0.016 | 0.299 | 0.694 | 0.048 | 0.006 | 0.031 | 0.073 | 0.009 | 0.004 |
| 16 (TFS) | 0.001 | 0.102 | 0.335 | 0.009 | 0.142 | 0.109 | 0.1 | 0.153 | 0.774 | 0.004 | 0.012 | 0.033 | 0.026 | 0.009 | 0.008 |
| 17 (SFF) | 0.067 | 0.049 | 0.358 | 0.008 | 0.031 | 0.107 | 0.09 | 0.321 | 0.024 | 0.005 | | | | | |
| 17 (FF) | 0.031 | 0.02 | 0.405 | 0.004 | 0.023 | 0.091 | 0.044 | 0.051 | 0.015 | 0.001 | | | | | |
| 17 (TFS) | 0.024 | 0.032 | 0.211 | 0.002 | 0.026 | 0.073 | 0.084 | 0.122 | 0.022 | 0.002 | | | | | |

Table 23: Standard deviations on 250 anomaly detection datasets under four random seeds.

| Index | 1 | 2 | 3 | 4 | 5 | 6 | 7 | 8 | 9 | 10 | 11 | 12 | 13 | 14 | 15 |
|---|---|---|---|---|---|---|---|---|---|---|---|---|---|---|---|
| 1 (SFF) | ±6.8e-3 | ±3.0e-3 | ±1.3e-3 | ±5.0e-2 | ±1.2e-3 | ±0.12 | 1.0e-2 | 1.1e-3 | 3.6e-3 | 0.11 | 1.6e-2 | 2.9e-2 | 0.68 | 9.2e-3 | 6.8e-3 |
| 1 (FF) | ±1.2e-2 | ±2.3e-3 | ±7.9e-4 | ±0.43 | ±1.7e-3 | ±1.9e-3 | 5.0e-2 | 1.3e-3 | 2.2e-5 | 1.3e-3 | 7.5e-4 | 1.8e-2 | 5.8e-2 | 1.0e-3 | 8.2e-4 |
| 1 (TFS) | ±1.1e-2 | ±1.4e-3 | ±2.4e-4 | ±0.15 | ±2.7e-3 | ±1.7e-3 | 2.0e-2 | 4.6e-4 | 1.8e-2 | 1.4e-3 | 6.3e-2 | 3.9e-3 | 0.74 | 5.3e-3 | 3.4e-2 |
| 2 (SFF) | ±5.6e-2 | ±3.9e-2 | ±1.4e-3 | ±1.3e-3 | ±1.6e-3 | ±6.9e-4 | 1.5e-2 | 3.8e-4 | 2.8e-2 | 2.1e-2 | 1.0e-2 | 2.0e-2 | 2.7e-2 | 4.1e-2 | 3.1e-3 |
| 2 (FF) | ±5.5e-3 | ±4.4e-3 | ±3.9e-4 | ±2.1e-4 | ±2.1e-3 | ±3.2e-4 | 2.6e-2 | 8.7e-4 | 1.7e-2 | 3.7e-3 | 1.7e-2 | 7.3e-3 | 4.2e-2 | 6.3e-2 | 4.0e-2 |
| 2 (TFS) | ±1.1e-2 | ±4.8e-4 | ±8.5e-4 | ±7.9e-2 | ±4.5e-3 | ±4.7e-4 | 6.3e-3 | 3.0e-4 | 3.0e-2 | 4.7e-3 | 4.7e-3 | 1.5e-2 | 5.7e-2 | 0.12 | 2.4e-3 |
| 3 (SFF) | ±1.1e-2 | ±2.6e-2 | ±5.1e-4 | ±1.4e-2 | ±2.4e-2 | ±9.6e-2 | 2.4e-2 | 6.9e-3 | 2.3e-2 | 3.9e-2 | 4.9e-3 | 5.3e-2 | 1.1e-3 | 1.3e-2 | 1.6e-2 |
| 3 (FF) | ±2.6e-4 | ±6.2e-3 | ±3.9e-4 | ±7.7e-4 | ±3.2e-2 | ±2.2e-4 | 1.2e-2 | 7.3e-4 | 5.5e-2 | 1.9e-2 | 3.5e-2 | 5.4e-2 | 2.7e-5 | 2.3e-2 | 8.4e-3 |
| 3 (TFS) | ±8.4e-4 | ±6.9e-3 | ±2.2e-3 | ±2.2e-2 | ±6.4e-2 | ±0.18 | 1.0e-2 | 4.5e-3 | 1.5e-2 | 4.1e-2 | 4.5e-3 | 7.5e-2 | 1.2e-4 | 2.3e-3 | 1.4e-4 |
| 4 (SFF) | ±4.2e-2 | ±3.4e-2 | ±1.3e-2 | ±3.0e-2 | ±1.7e-2 | ±7.8e-3 | 0.1 | 9.1e-2 | 0.23 | 7.7e-2 | 3.8e-3 | 3.6e-2 | 0.11 | 3.1e-2 | 3.4e-4 |
| 4 (FF) | ±5.8e-3 | ±1.5e-2 | ±7.7e-4 | ±2.4e-3 | ±3.0e-3 | ±2.2e-2 | 0.18 | 1.9e-3 | 1.9e-3 | 5.2e-3 | 5.7e-5 | 5.5e-2 | 2.6e-2 | 4.3e-3 | 3.4e-4 |
| 4 (TFS) | ±5.8e-3 | ±5.2e-2 | ±7.8e-2 | ±2.1e-2 | ±5.3e-3 | ±1.1e-2 | 0.22 | 9.8e-4 | 3.9e-2 | 5.7e-3 | 3.5e-3 | 2.5e-2 | 5.5e-2 | 2.5e-3 | 9.9e-4 |
| 5 (SFF) | ±0.14 | ±0.47 | ±3.1e-3 | ±3.3e-2 | ±2.2e-2 | ±1.1e-2 | 3.4e-2 | 1.5e-2 | 1.9e-2 | 8.7e-3 | 4.6e-2 | 2.4e-2 | 0.35 | 2.0e-3 | 1.0e-3 |
| 5 (FF) | ±0.15 | ±3.9e-2 | ±5.1e-3 | ±5.0e-2 | ±5.7e-4 | ±1.8e-2 | 1.0e-2 | 1.6e-2 | 1.3e-2 | 9.3e-3 | 4.6e-4 | 6.9e-2 | 4.8e-3 | 1.2e-3 | 2.6e-3 |
| 5 (TFS) | ±0.1 | ±0.11 | ±2.4e-2 | ±4.6e-2 | ±7.5e-5 | ±7.2e-3 | 1.5e-3 | 4.7e-3 | 6.3e-3 | 3.0e-3 | 4.6e-3 | 8.8e-2 | 2.0e-3 | 4.4e-3 | 5.9e-3 |
| 6 (SFF) | ±4.1e-2 | ±4.6e-3 | ±1.7e-2 | ±1.6e-2 | ±5.4e-2 | ±1.2e-2 | 0.2 | 9.4e-2 | 1.6e-2 | 1.3e-2 | 5.1e-3 | 7.5e-5 | 4.0e-3 | 1.0e-3 | 9.2e-2 |
| 6 (FF) | ±6.8e-3 | ±2.9e-3 | ±8.2e-3 | ±6.9e-3 | ±6.2e-3 | ±7.6e-2 | 5.6e-2 | 1.7e-3 | 6.0e-3 | 5.4e-3 | 4.7e-3 | 9.2e-5 | 3.3e-4 | 3.4e-5 | 1.8e-3 |
| 6 (TFS) | ±6.8e-3 | ±5.1e-5 | ±2.3e-3 | ±4.3e-3 | ±5.2e-3 | ±2.1e-2 | 7.8e-3 | 4.9e-4 | 3.9e-4 | 3.0e-4 | 1.1e-2 | 2.9e-4 | 1.4e-2 | 1.3e-3 | 1.1e-3 |
| 7 (SFF) | ±0.28 | ±6.8e-4 | ±3.5e-3 | ±4.4e-3 | ±1.0 | ±9.3e-3 | 1.1e-2 | 4.9e-2 | 1.9e-2 | 4.4e-3 | 6.1e-3 | 6.7e-3 | 0.16 | 1.8e-2 | 6.2e-3 |
| 7 (FF) | ±2.0e-3 | ±7.6e-4 | ±3.1e-3 | ±2.7e-3 | ±0.41 | ±2.4e-2 | 2.1e-4 | 5.1e-2 | 1.9e-4 | 2.7e-2 | 3.4e-2 | 2.1e-3 | 0.1 | 5.4e-2 | 2.2e-2 |
| 7 (TFS) | ±3.7e-3 | ±8.3e-4 | ±9.1e-3 | ±1.9e-3 | ±1.8e-2 | ±7.0e-3 | 1.4e-2 | 2.7e-2 | 3.0e-3 | 2.5e-2 | 2.1e-2 | 1.5e-3 | 4.0e-2 | 5.1e-2 | 1.1e-2 |
| 8 (SFF) | ±1.6e-3 | ±2.3e-4 | ±5.8e-3 | ±2.8e-2 | ±3.2e-2 | ±1.7e-4 | 2.8e-2 | 1.2e-2 | 2.5e-2 | 1.5e-3 | 1.5e-2 | 5.1e-2 | 5.4e-3 | 4.9e-2 | 4.5e-4 |
| 8 (FF) | ±4.8e-2 | ±9.8e-5 | ±1.2e-3 | ±1.1e-2 | ±2.1e-2 | ±1.5e-3 | 1.0e-2 | 2.6e-2 | 3.8e-3 | 5.6e-3 | 1.3e-2 | 2.0e-2 | 8.4e-4 | 1.1e-2 | 1.4e-3 |
| 8 (TFS) | ±5.9e-3 | ±5.5e-3 | ±1.6e-3 | ±1.0e-2 | ±2.0e-2 | ±9.2e-4 | 2.0e-2 | 4.9e-2 | 1.0e-2 | 1.2e-4 | 2.7e-2 | 2.2e-2 | 1.6e-3 | 8.6e-2 | 3.8e-3 |
| 9 (SFF) | ±0.36 | ±5.6e-2 | ±0.1 | ±4.5e-2 | ±2.9e-2 | ±7.5e-2 | 1.5e-2 | 2.4e-2 | 3.1e-2 | 1.2e-2 | 1.8e-2 | 3.4e-2 | 3.3e-2 | 1.6e-3 | 5.5e-3 |
| 9 (FF) | ±0.46 | ±1.3e-2 | ±2.0e-3 | ±3.4e-2 | ±2.3e-2 | ±8.5e-2 | 7.9e-2 | 3.2e-2 | 2.8e-2 | 9.5e-4 | 5.4e-4 | 4.8e-3 | 1.9e-3 | 1.3e-3 | 6.1e-4 |
| 9 (TFS) | ±0.96 | ±2.3e-2 | ±0.12 | ±4.0e-2 | ±9.2e-3 | ±8.7e-2 | 8.7e-2 | 2.6e-2 | 2.4e-2 | 1.7e-2 | 1.5e-2 | 6.9e-3 | 1.8e-2 | 4.7e-2 | 6.4e-4 |
| 10 (SFF) | ±6.5e-3 | ±7.5e-3 | ±0.27 | ±2.3e-2 | ±2.9e-2 | ±3.4e-4 | 2.1e-2 | 2.2e-2 | 5.0e-2 | 2.4e-2 | 0.13 | 6.3e-2 | 4.9e-3 | 0.16 | 9.7e-2 |
| 10 (FF) | ±2.9e-3 | ±3.3e-3 | ±7.9e-3 | ±5.2e-4 | ±9.2e-4 | ±2.0e-4 | 5.4e-3 | 2.2e-2 | 3.5e-2 | 2.9e-3 | 4.8e-2 | 4.3e-3 | 9.0e-3 | 0.14 | 3.3e-3 |
| 10 (TFS) | ±1.5e-2 | ±1.8e-3 | ±0.16 | ±1.1e-3 | ±5.3e-2 | ±2.9e-4 | 6.8e-3 | 7.0e-3 | 8.6e-3 | 0.15 | 0.24 | 2.8e-3 | 2.2e-3 | 0.18 | 1.3e-3 |
| 11 (SFF) | ±5.4e-2 | ±1.6e-2 | ±3.2e-2 | ±6.6e-4 | ±5.0e-2 | ±1.7e-3 | 3.6e-3 | 2.4e-2 | 1.3e-3 | 0.86 | 4.7e-2 | 3.6e-3 | 7.6e-4 | 6.4e-3 | 0.11 |
| 11 (FF) | ±1.2e-2 | ±6.8e-4 | ±3.1e-2 | ±1.5e-3 | ±3.7e-2 | ±1.5e-2 | 4.7e-3 | 6.1e-3 | 1.1e-3 | 7.5e-2 | 4.6e-3 | 9.3e-4 | 2.9e-4 | 1.3e-2 | 0.19 |
| 11 (TFS) | ±6.3e-2 | ±1.1e-2 | ±3.3e-2 | ±3.4e-4 | ±9.3e-3 | ±2.6e-4 | 7.4e-3 | 2.5e-2 | 8.6e-4 | 2.9e-2 | 3.1e-3 | 1.0e-3 | 3.6e-3 | 9.9e-3 | 8.9e-2 |
| 12 (SFF) | ±5.2e-2 | ±1.4e-3 | ±2.2e-3 | ±4.1e-2 | ±5.8e-2 | ±2.6e-2 | 0.39 | 1.5e-2 | 2.7e-2 | 4.5e-3 | 1.2e-2 | 1.5e-3 | 2.3e-4 | 6.3e-4 | 2.0e-2 |
| 12 (FF) | ±1.7e-2 | ±5.2e-3 | ±1.2e-2 | ±3.8e-3 | ±2.0e-3 | ±3.3e-2 | 7.9e-3 | 5.2e-2 | 9.2e-3 | 4.8e-2 | 8.3e-3 | 7.4e-5 | 6.3e-5 | 1.1e-4 | 2.8e-2 |
| 12 (TFS) | ±0.15 | ±8.7e-3 | ±1.8e-2 | ±3.8e-3 | ±5.6e-2 | ±4.1e-3 | 0.35 | 7.3e-2 | 2.0e-2 | 2.0e-2 | 1.4e-2 | 1.1e-3 | 6.9e-4 | 2.1e-4 | 1.1e-2 |
| 13 (SFF) | ±8.6e-3 | ±3.9e-2 | ±6.5e-4 | ±1.0e-3 | ±3.8e-2 | ±2.2e-4 | 7.7e-4 | 2.5e-3 | 0.1 | 0.19 | 3.5e-3 | 2.1e-3 | 3.9e-2 | 1.6e-2 | 6.8e-3 |
| 13 (FF) | ±1.6e-2 | ±7.7e-3 | ±3.6e-4 | ±4.5e-4 | ±2.7e-3 | ±2.8e-4 | 4.6e-4 | 1.4e-2 | 1.4e-3 | 5.5e-3 | 1.4e-3 | 7.4e-3 | 1.6e-2 | 7.6e-4 | 3.1e-3 |
| 13 (TFS) | ±1.8e-2 | ±3.3e-2 | ±3.1e-4 | ±2.5e-3 | ±2.2e-3 | ±6.2e-4 | 2.2e-4 | 5.4e-2 | 2.3e-3 | 6.1e-4 | 2.1e-3 | 5.9e-3 | 5.9e-2 | 5.4e-3 | 5.8e-3 |
| 14 (SFF) | ±0.91 | ±0.15 | ±3.1e-3 | ±3.6e-2 | ±4.6e-3 | ±5.6e-4 | 8.4e-2 | 7.6e-3 | 5.1e-4 | 2.9e-2 | 6.6e-3 | 2.8e-3 | 1.0e-2 | 6.5e-3 | 2.4e-2 |
| 14 (FF) | ±5.4e-2 | ±4.4e-2 | ±2.1e-3 | ±2.9e-5 | ±2.3e-3 | ±3.7e-4 | 2.4e-3 | 8.8e-4 | 2.3e-4 | 8.8e-2 | 2.5e-2 | 1.0e-2 | 5.6e-3 | 1.3e-2 | 4.3e-3 |
| 14 (TFS) | ±3.1e-2 | ±0.19 | ±5.9e-3 | ±9.0e-3 | ±3.2e-3 | ±5.3e-4 | 1.3e-2 | 1.3e-2 | 1.0e-4 | 0.4 | 1.6e-2 | 5.5e-3 | 9.4e-3 | 1.4e-2 | 2.5e-2 |
| 15 (SFF) | ±5.2e-4 | ±3.9e-2 | ±8.6e-3 | ±1.7e-2 | ±6.7e-2 | ±1.4e-3 | 9.9e-3 | 0.11 | 5.7e-3 | 8.1e-4 | 8.0e-4 | 3.6e-2 | 2.0e-3 | 3.1e-3 | 9.7e-2 |
| 15 (FF) | ±8.1e-5 | ±7.1e-3 | ±5.3e-3 | ±7.0e-4 | ±1.2e-3 | ±1.2e-3 | 3.6e-2 | 0.13 | 3.2e-2 | 1.5e-4 | 3.5e-3 | 9.7e-2 | 3.0e-4 | 1.8e-4 | 1.9e-2 |
| 15 (TFS) | ±3.3e-3 | ±5.9e-3 | ±5.1e-4 | ±3.7e-2 | ±2.0e-2 | ±3.6e-4 | 5.1e-2 | 9.3e-2 | 1.5e-2 | 1.4e-3 | 8.2e-3 | 5.1e-2 | 1.2e-3 | 1.6e-4 | 7.8e-2 |
| 16 (SFF) | ±3.6e-4 | ±4.2e-2 | ±1.5e-2 | ±7.2e-4 | ±2.2e-2 | ±8.0e-3 | 1.9e-2 | 0.13 | 0.12 | 5.8e-4 | 1.8e-3 | 8.4e-4 | 1.0e-2 | 2.5e-3 | 1.8e-3 |
| 16 (FF) | ±6.4e-4 | ±1.7e-2 | ±5.5e-2 | ±3.7e-2 | ±1.3e-2 | ±1.9e-2 | 4.5e-3 | 0.2 | 8.6e-2 | 6.5e-2 | 2.0e-3 | 1.8e-2 | 4.8e-2 | 1.1e-2 | 4.6e-3 |
| 16 (TFS) | ±5.4e-4 | ±2.4e-2 | ±0.13 | ±1.7e-3 | ±1.8e-2 | ±6.8e-2 | 5.7e-3 | 0.18 | 3.8e-2 | 1.1e-3 | 6.6e-3 | 1.1e-2 | 2.1e-2 | 5.5e-4 | 1.1e-2 |
| 17 (SFF) | ±1.8e-2 | ±1.2e-2 | ±7.7e-2 | ±3.0e-3 | ±3.4e-3 | ±6.8e-3 | 4.2e-3 | 0.13 | 1.4e-3 | 4.2e-3 | | | | | |
| 17 (FF) | ±1.8e-2 | ±3.0e-3 | ±3.4e-3 | ±2.1e-3 | ±4.1e-3 | ±1.5e-3 | 1.8e-3 | 5.7e-3 | 5.6e-4 | 8.5e-5 | | | | | |
| 17 (TFS) | ±1.1e-2 | ±1.5e-3 | ±7.7e-2 | ±3.8e-4 | ±3.0e-3 | ±6.5e-3 | 4.7e-3 | 4.9e-3 | 7.9e-3 | 1.4e-3 | | | | | |

Table 24: Complete standard deviation and MSE of applying our *smoothed fine-tuning* (SFF) on other LTSMs TimesFM and MOMENT.

| Data proportion | 25% (TimesFM) | | | 100% (TimesFM) | | | 25% (MOMENT) | | | 100% (MOMENT) | | |
|---|---|---|---|---|---|---|---|---|---|---|---|---|
| Methods | SFF | FF | TFS | SFF | FF | TFS | SFF | FF | TFS | SFF | FF | TFS |
| Exchange | **0.1139** | 0.1276 | 0.1209 | **0.1149** | 0.1452 | 0.1199 | **0.1502** | 0.2648 | 0.1564 | **0.1064** | 0.1448 | 0.1091 |
| Standard deviation | 2.0e-3 | 4.2e-3 | 2.9e-4 | 6.3e-4 | 1.7e-2 | 2.3e-3 | 2.4e-3 | 4.6e-4 | 3.6e-3 | 5.8e-4 | 1.4e-4 | 2.6e-4 |
| ETTh1 | **0.3955** | 0.4382 | 0.4638 | **0.406** | 0.5101 | 0.4358 | **0.4287** | 0.4454 | 0.454 | **0.3757** | 0.3951 | 0.387 |
| Standard deviation | 1.9e-3 | 2.4e-2 | 6.0e-4 | 3.6e-3 | 8.8e-3 | 2.0e-3 | 2.1e-3 | 9.9e-4 | 1.8e-3 | 4.0e-4 | 6.5e-5 | 1.4e-3 |
| ETTh2 | **0.3232** | 0.3384 | 0.3325 | **0.3198** | 0.3483 | 0.347 | **0.3199** | 0.3328 | 0.3326 | **0.2818** | 0.2936 | 0.2979 |
| Standard deviation | 2.9e-3 | 9.0e-3 | 9.5e-4 | 2.0e-3 | 3.3e-3 | 4.7e-3 | 1.4e-3 | 2.6e-4 | 1.5e-3 | 7.8e-4 | 4.0e-5 | 1.6e-3 |
| ETTm1 | **0.3429** | 0.4001 | 0.3903 | **0.3478** | 0.3756 | 0.3926 | **0.3457** | 0.3587 | 0.3538 | **0.3139** | 0.3148 | 0.3272 |
| Standard deviation | 3.6e-3 | 7.3e-3 | 7.2e-4 | 2.9e-3 | 3.0e-2 | 4.8e-4 | 1.2e-3 | 1.2e-4 | 6.6e-4 | 1.0e-4 | 3.1e-5 | 2.0e-3 |
| ETTm2 | **0.1983** | 0.2061 | 0.2091 | **0.2026** | 0.2122 | 0.225 | **0.1793** | 0.192 | 0.1846 | **0.1692** | 0.172 | 0.1736 |
| Standard deviation | 3.7e-3 | 2.6e-3 | 4.0e-4 | 1.5e-3 | 6.8e-3 | 3.8e-3 | 2.1e-4 | 5.0e-4 | 6.1e-4 | 3.7e-4 | 3.7e-5 | 9.5e-4 |
| Weather | **0.0865** | 0.0885 | 0.1995 | **0.082** | 0.1184 | 0.1902 | **0.1673** | 0.1682 | 0.169 | **0.1548** | 0.1558 | 0.161 |
| Standard deviation | 4.7e-3 | 5.6e-3 | 4.5e-3 | 1.1e-2 | 3.0e-2 | 4.2e-3 | 1.1e-4 | 1.4e-4 | 1.7e-4 | 1.9e-4 | 2.2e-4 | 1.4e-4 |
| Avg. Improvements | - | 7.55% | 16.21 | - | 15.35% | 16.18 | - | 10.28% | 3.25 | - | 6.34% | 3.54 |
| Max. Improvements | - | 14.3% | 56.64 | - | 30.74% | 56.89 | - | 43.28% | 5.57 | - | 26.52% | 5.4 |

Table 25: Complete standard deviation and MAE of applying our *smoothed fine-tuning* (SFF) on other LTSMs TimesFM and MOMENT.

| Data proportion | 25% (TimesFM) | | | 100% (TimesFM) | | | 25% (MOMENT) | | | 100% (MOMENT) | | |
|---|---|---|---|---|---|---|---|---|---|---|---|---|
| Methods | SFF | FF | TFS | SFF | FF | TFS | SFF | FF | TFS | SFF | FF | TFS |
| Exchange | **0.2414** | 0.2519 | 0.2497 | **0.2422** | 0.2703 | 0.2472 | **0.282** | 0.3844 | 0.2894 | **0.2322** | 0.2751 | 0.2369 |
| Standard deviation | 9.9e-4 | 4.3e-3 | 4.2e-4 | 8.4e-4 | 1.8e-2 | 1.7e-3 | 2.3e-3 | 3.3e-4 | 3.4e-4 | 5.8e-4 | 1.0e-4 | 3.1e-4 |
| ETTh1 | **0.405** | 0.4226 | 0.4526 | **0.4149** | 0.4567 | 0.4344 | **0.4386** | 0.4455 | 0.4559 | **0.4022** | 0.4144 | 0.4112 |
| Standard deviation | 4.3e-3 | 8.0e-3 | 4.5e-4 | 3.0e-3 | 2.7e-3 | 5.3e-4 | 1.2e-3 | 7.4e-4 | 8.3e-4 | 4.2e-4 | 2.6e-5 | 1.3e-3 |
| ETTh2 | **0.3731** | 0.3742 | 0.3803 | **0.3704** | 0.3782 | 0.391 | **0.3695** | 0.3797 | 0.3784 | **0.3404** | 0.35 | 0.3514 |
| Standard deviation | 1.2e-3 | 3.9e-3 | 9.3e-4 | 8.3e-4 | 2.6e-3 | 1.4e-3 | 1.4e-3 | 3.5e-4 | 9.3e-4 | 3.3e-4 | 4.1e-5 | 8.5e-4 |
| ETTm1 | **0.3851** | 0.4119 | 0.4223 | **0.3892** | 0.3992 | 0.4227 | **0.3938** | 0.4054 | 0.4013 | **0.3783** | 0.3787 | 0.3854 |
| Standard deviation | 2.1e-3 | 3.8e-3 | 4.3e-4 | 3.2e-3 | 1.5e-2 | 7.6e-4 | 1.3e-3 | 1.3e-4 | 4.6e-4 | 1.6e-4 | 2.4e-5 | 7.5e-4 |
| ETTm2 | **0.2823** | 0.2847 | 0.2925 | **0.2748** | 0.2851 | 0.3119 | **0.2671** | 0.2769 | 0.2724 | **0.2587** | 0.2613 | 0.2638 |
| Standard deviation | 1.0e-3 | 1.4e-3 | 3.8e-4 | 1.9e-3 | 5.1e-3 | 4.4e-3 | 6.3e-5 | 4.8e-4 | 4.9e-4 | 1.4e-3 | 1.2e-5 | 7.5e-4 |
| Weather | **0.1135** | 0.1161 | 0.2523 | **0.1025** | 0.152 | 0.2424 | **0.2213** | 0.2247 | 0.2231 | **0.2107** | 0.2114 | 0.2142 |
| Standard deviation | 3.3e-3 | 6.7e-3 | 4.2e-3 | 1.4e-2 | 4.0e-2 | 4.2e-3 | 1.1e-3 | 1.2e-4 | 3.6e-4 | 1.1e-4 | 4.0e-4 | 2.5e-4 |
| Avg. Improvements | - | 3.04% | 13.84% | - | 10.05% | 14.88% | - | 6.46% | 2.22% | - | 3.78% | 2.12% |
| Max. Improvements | - | 6.51% | 55.01% | - | 32.57% | 57.71% | - | 26.64% | 3.79% | - | 15.59% | 3.13% |

Table 26: Full standard deviation and MSE of comparing our smoothed full fine-tuning (SFF) with linear-probing (LP) and linear-probing then full fine-tuning (LPFF). The average performance is reported for each group of three different data proportions, e.g., "Avg. on 1%, 2%, 3%".

| Data proportion | Avg. on 1%, 2%, 3% | | | Avg. on 4%, 5%, 10% | | | Avg. on 15%, 20%, 25% | | | Avg. on 50%, 75%, 100% | | |
|---|---|---|---|---|---|---|---|---|---|---|---|---|
| Methods | SFF | LP | LPFF | SFF | LP | LPFF | SFF | LP | LPFF | SFF | LP | LPFF |
| Exchange | **0.0856** | 0.5943 | 0.4801 | **0.0848** | 0.5906 | 0.4186 | **0.0816** | 0.563 | 0.1743 | **0.0812** | 0.474 | 0.0962 |
| Standard deviation | 4.4e-4 | 7.3e-3 | 3.9e-3 | 3.7e-4 | 7.2e-3 | 6.7e-3 | 6.1e-4 | 6.6e-3 | 7.4e-3 | 8.3e-4 | 4.7e-3 | 1.9e-3 |
| ETTh1 | **0.3722** | 0.8806 | 0.7171 | **0.3641** | 0.8594 | 0.6367 | **0.3523** | 0.7955 | 0.4127 | **0.3529** | 0.6356 | 0.3731 |
| Standard deviation | 6.5e-3 | 9.2e-3 | 3.6e-3 | 4.7e-3 | 8.2e-3 | 9.2e-3 | 1.4e-3 | 6.7e-3 | 3.4e-3 | 1.3e-3 | 4.5e-3 | 1.8e-3 |
| ETTh2 | **0.28** | 0.4427 | 0.4026 | **0.278** | 0.4375 | 0.3707 | **0.2768** | 0.4234 | 0.3113 | **0.2758** | 0.3849 | 0.3001 |
| Standard deviation | 1.5e-3 | 7.3e-3 | 3.8e-3 | 2.8e-3 | 6.8e-3 | 1.6e-3 | 2.6e-3 | 5.8e-3 | 1.5e-3 | 9.7e-4 | 3.1e-3 | 1.7e-3 |
| ETTm1 | **0.3448** | 1.046 | 0.7038 | **0.3162** | 1.0043 | 0.4772 | **0.301** | 0.8975 | 0.3245 | **0.2976** | 0.6608 | 0.3124 |
| Standard deviation | 3.5e-3 | 2.6e-2 | 7.3e-3 | 2.6e-3 | 2.3e-2 | 4.5e-3 | 1.0e-3 | 1.8e-2 | 6.8e-4 | 1.5e-3 | 9.1e-3 | 1.2e-3 |
| ETTm2 | **0.1723** | 0.3555 | 0.3024 | **0.1663** | 0.3499 | 0.2559 | **0.1623** | 0.3297 | 0.1847 | **0.1616** | 0.2771 | 0.1804 |
| Standard deviation | 2.3e-3 | 6.2e-3 | 3.1e-3 | 2.1e-3 | 5.9e-3 | 2.0e-3 | 1.0e-3 | 4.9e-3 | 9.8e-4 | 6.2e-4 | 2.7e-3 | 1.6e-3 |
| Weather | **0.1515** | 0.324 | 0.2478 | **0.146** | 0.3082 | 0.1741 | **0.1441** | 0.2699 | 0.1481 | **0.1453** | 0.2013 | 0.1565 |
| Standard deviation | 4.7e-4 | 4.2e-3 | 4.1e-3 | 1.9e-4 | 3.4e-3 | 3.5e-3 | 6.6e-5 | 1.9e-3 | 5.6e-4 | 3.6e-4 | 8.2e-4 | 1.1e-3 |
| Electricity | **0.1346** | 0.6069 | 0.181 | **0.132** | 0.3242 | 0.1398 | **0.1305** | 0.2023 | 0.132 | **0.1301** | 0.1561 | 0.1335 |
| Standard deviation | 2.0e-4 | 1.1e-3 | 8.6e-4 | 2.3e-4 | 4.3e-4 | 2.6e-4 | 1.8e-4 | 1.9e-4 | 2.6e-4 | 2.7e-4 | 8.9e-5 | 5.8e-4 |
| Traffic | **0.3678** | 0.9577 | 0.4081 | **0.3562** | 0.5999 | 0.3638 | **0.3494** | 0.4529 | 0.3572 | **0.3516** | 0.4079 | 0.3575 |
| Standard deviation | 1.3e-4 | 2.2e-3 | 4.4e-4 | 4.2e-4 | 1.9e-3 | 3.7e-4 | 1.8e-3 | 4.4e-4 | 7.4e-4 | 1.4e-3 | 1.5e-4 | 5.4e-4 |
| Avg. Improvements | - | - | 41.14% | - | - | 30.02% | - | - | 13.04% | - | - | 6.95% |
| Max. Improvements | - | - | 82.17% | - | - | 79.74% | - | - | 53.18% | - | - | 15.59% |

Table 27: Full standard deviation and MAE of comparing our smoothed full fine-tuning (SFF) with linear-probing (LP) and linear-probing then full fine-tuning (LPFF). The average performance is reported for each group of three different data proportions, e.g., "Avg. on 1%, 2%, 3%".

| Data proportion | Avg. on 1%, 2%, 3% | | | Avg. on 4%, 5%, 10% | | | Avg. on 15%, 20%, 25% | | | Avg. on 50%, 75%, 100% | | |
|---|---|---|---|---|---|---|---|---|---|---|---|---|
| Methods | SFF | LP | LPFF | SFF | LP | LPFF | SFF | LP | LPFF | SFF | LP | LPFF |
| Exchange | **0.2047** | 0.5867 | 0.5287 | **0.2039** | 0.5848 | 0.4886 | **0.2013** | 0.5714 | 0.3048 | **0.2014** | 0.5244 | 0.2202 |
| Standard deviation | 3.5e-4 | 2.6e-3 | 1.4e-3 | 5.1e-4 | 2.6e-3 | 3.9e-3 | 7.2e-4 | 2.5e-3 | 6.6e-3 | 1.1e-3 | 2.1e-3 | 2.8e-3 |
| ETTh1 | **0.4006** | 0.644 | 0.5865 | **0.3963** | 0.6364 | 0.5505 | **0.3886** | 0.6134 | 0.4376 | **0.3905** | 0.5504 | 0.4097 |
| Standard deviation | 3.4e-3 | 4.8e-3 | 1.3e-3 | 3.5e-3 | 4.4e-3 | 3.9e-3 | 1.1e-3 | 3.8e-3 | 2.1e-3 | 1.4e-3 | 2.8e-3 | 1.0e-3 |
| ETTh2 | **0.3345** | 0.46 | 0.4366 | **0.3347** | 0.457 | 0.4154 | **0.3358** | 0.4488 | 0.3671 | **0.3365** | 0.4248 | 0.3558 |
| Standard deviation | 9.5e-4 | 4.1e-3 | 2.1e-3 | 1.4e-3 | 3.9e-3 | 9.5e-4 | 1.8e-3 | 3.3e-3 | 1.3e-3 | 7.3e-4 | 1.9e-3 | 1.2e-3 |
| ETTm1 | **0.3942** | 0.7198 | 0.597 | **0.3735** | 0.7061 | 0.4873 | **0.3637** | 0.6695 | 0.3874 | **0.3592** | 0.5751 | 0.3759 |
| Standard deviation | 2.3e-3 | 8.9e-3 | 2.1e-3 | 1.3e-3 | 8.3e-3 | 1.2e-3 | 9.5e-4 | 6.7e-3 | 3.5e-4 | 1.8e-3 | 3.6e-3 | 6.5e-4 |
| ETTm2 | **0.2601** | 0.3958 | 0.3636 | **0.2547** | 0.3926 | 0.3328 | **0.2523** | 0.381 | 0.2778 | **0.2512** | 0.349 | 0.2711 |
| Standard deviation | 2.5e-3 | 3.5e-3 | 1.7e-3 | 1.5e-3 | 3.4e-3 | 1.1e-3 | 5.2e-4 | 2.9e-3 | 1.3e-3 | 8.2e-4 | 1.7e-3 | 1.5e-3 |
| Weather | **0.2005** | 0.3584 | 0.299 | **0.1941** | 0.3472 | 0.2298 | **0.1929** | 0.3189 | 0.2007 | **0.1946** | 0.2598 | 0.2072 |
| Standard deviation | 5.1e-4 | 2.8e-3 | 2.9e-3 | 2.0e-4 | 2.3e-3 | 3.1e-3 | 2.4e-4 | 1.3e-3 | 6.5e-4 | 6.8e-4 | 6.2e-4 | 1.7e-3 |
| Electricity | **0.2312** | 0.6057 | 0.2824 | **0.2275** | 0.411 | 0.2387 | **0.2251** | 0.3048 | 0.2269 | **0.2243** | 0.2599 | 0.227 |
| Standard deviation | 3.1e-4 | 2.3e-3 | 6.4e-4 | 2.2e-4 | 8.3e-4 | 2.9e-4 | 3.5e-4 | 6.6e-5 | 1.2e-4 | 3.7e-4 | 7.8e-5 | 6.9e-4 |
| Traffic | **0.2619** | 0.6062 | 0.3052 | **0.2532** | 0.4331 | 0.263 | **0.2474** | 0.3423 | 0.2519 | **0.2474** | 0.2972 | 0.2486 |
| Standard deviation | 3.9e-5 | 4.8e-4 | 2.2e-4 | 4.0e-4 | 9.6e-4 | 1.0e-4 | 1.8e-3 | 2.0e-4 | 1.1e-3 | 6.2e-4 | 1.6e-4 | 3.3e-4 |
| Avg. Improvements | - | - | 30.51% | - | - | 22.06% | - | - | 9.43% | - | - | 4.77% |
| Max. Improvements | - | - | 61.28% | - | - | 58.27% | - | - | 33.96% | - | - | 8.54% |

Table 28: Full standard deviation and **MSE** of comparing our smoothed full fine-tuning (SFF) with linear-probing (LP) and linear-probing then full fine-tuning (LPFF) under **1%, 2%, 3%, and 4%** proportion of available data.

| Data proportion | 1% | | | 2% | | | 3% | | | 4% | | |
|---|---|---|---|---|---|---|---|---|---|---|---|---|
| Methods | SFF | LP | LPFF | SFF | LP | LPFF | SFF | LP | LPFF | SFF | LP | LPFF |
| Exchange | **0.0857** | 0.5944 | 0.4841 | **0.0863** | 0.5943 | 0.4788 | **0.085** | 0.5943 | 0.4773 | **0.0852** | 0.5943 | 0.4763 |
| Standard deviation | 3.2e-4 | 7.3e-3 | 4.5e-3 | 4.2e-4 | 7.3e-3 | 3.0e-3 | 5.8e-4 | 7.3e-3 | 4.2e-3 | 2.4e-4 | 7.3e-3 | 3.9e-3 |
| ETTh1 | **0.3737** | 0.8809 | 0.7219 | **0.3731** | 0.8806 | 0.7159 | **0.37** | 0.8804 | 0.7134 | **0.371** | 0.8731 | 0.7677 |
| Standard deviation | 6.3e-3 | 9.1e-3 | 3.0e-3 | 6.6e-3 | 9.2e-3 | 3.4e-3 | 6.6e-3 | 9.2e-3 | 4.3e-3 | 7.5e-3 | 7.7e-3 | 5.3e-3 |
| ETTh2 | **0.2839** | 0.4428 | 0.4047 | **0.2769** | 0.4427 | 0.402 | **0.2797** | 0.4427 | 0.4012 | **0.279** | 0.4394 | 0.3874 |
| Standard deviation | 1.0e-3 | 7.3e-3 | 4.1e-3 | 1.7e-3 | 7.3e-3 | 3.7e-3 | 1.9e-3 | 7.3e-3 | 3.7e-3 | 2.7e-3 | 6.9e-3 | 2.5e-3 |
| ETTm1 | **0.3702** | 1.0585 | 0.8141 | **0.3343** | 1.0401 | 0.6548 | **0.3301** | 1.0393 | 0.6424 | **0.3229** | 1.0215 | 0.5353 |
| Standard deviation | 4.4e-3 | 2.7e-2 | 8.4e-3 | 2.7e-3 | 2.6e-2 | 6.6e-3 | 3.4e-3 | 2.6e-2 | 6.7e-3 | 3.5e-3 | 2.4e-2 | 5.0e-3 |
| ETTm2 | **0.1744** | 0.3569 | 0.3154 | **0.1759** | 0.3568 | 0.3139 | **0.1668** | 0.3528 | 0.2779 | **0.1674** | 0.3527 | 0.2757 |
| Standard deviation | 2.4e-3 | 6.3e-3 | 3.5e-3 | 2.1e-3 | 6.3e-3 | 3.0e-3 | 2.4e-3 | 6.0e-3 | 2.6e-3 | 1.4e-3 | 6.1e-3 | 2.1e-3 |
| Weather | **0.1541** | 0.3273 | 0.2726 | **0.1507** | 0.324 | 0.2477 | **0.1499** | 0.3207 | 0.223 | **0.1474** | 0.3175 | 0.2 |
| Standard deviation | 3.9e-4 | 4.4e-3 | 2.8e-3 | 1.5e-4 | 4.2e-3 | 4.5e-3 | 8.7e-4 | 4.0e-3 | 4.9e-3 | 2.9e-4 | 3.8e-3 | 6.5e-3 |
| Electricity | **0.1369** | 0.7664 | 0.2197 | **0.1342** | 0.5866 | 0.1698 | **0.133** | 0.4678 | 0.1535 | **0.1331** | 0.3911 | 0.1464 |
| Standard deviation | 7.8e-5 | 7.9e-4 | 9.6e-4 | 1.9e-4 | 1.4e-3 | 9.8e-4 | 3.3e-4 | 1.2e-3 | 6.5e-4 | 2.5e-4 | 5.6e-4 | 4.6e-4 |
| Traffic | **0.3743** | 1.1903 | 0.4536 | **0.3671** | 0.917 | 0.3941 | **0.3622** | 0.7658 | 0.3767 | **0.3594** | 0.6768 | 0.3688 |
| Standard deviation | 1.0e-4 | 2.4e-3 | 1.1e-3 | 1.4e-5 | 1.7e-3 | 1.7e-4 | 2.7e-4 | 2.4e-3 | 3.3e-5 | 2.6e-4 | 2.6e-3 | 2.2e-4 |
| Avg. Improvements | - | 62.35% | 44.78% | - | 61.21% | 40.11% | - | 59.87% | 37.4% | - | 58.38% | 34.83% |
| Max. Improvements | - | 85.58% | 82.3% | - | 85.48% | 81.98% | - | 85.7% | 82.19% | - | 85.66% | 82.11% |

Table 29: Full standard deviation and **MSE** of comparing our smoothed full fine-tuning (SFF) with linear-probing (LP) and linear-probing then full fine-tuning (LPFF) under **5%, 10%, 15%, and 20%** proportion of available data.

| Data proportion | 5% | | | 10% | | | 15% | | | 20% | | |
|---|---|---|---|---|---|---|---|---|---|---|---|---|
| Methods | SFF | LP | LPFF | SFF | LP | LPFF | SFF | LP | LPFF | SFF | LP | LPFF |
| Exchange | **0.0856** | 0.5936 | 0.468 | **0.0838** | 0.5838 | 0.3114 | **0.082** | 0.5737 | 0.2205 | **0.0818** | 0.5613 | 0.1646 |
| Standard deviation | 1.6e-4 | 7.4e-3 | 3.0e-3 | 7.2e-4 | 7.0e-3 | 1.3e-2 | 3.1e-4 | 6.8e-3 | 1.1e-2 | 1.1e-3 | 6.7e-3 | 6.2e-3 |
| ETTh1 | **0.3646** | 0.8618 | 0.6158 | **0.3568** | 0.8434 | 0.5265 | **0.3551** | 0.8139 | 0.4433 | **0.3509** | 0.7926 | 0.4029 |
| Standard deviation | 4.6e-3 | 8.7e-3 | 1.2e-2 | 2.1e-3 | 8.2e-3 | 1.1e-2 | 1.7e-3 | 6.6e-3 | 4.2e-3 | 1.9e-3 | 6.9e-3 | 3.5e-3 |
| ETTh2 | **0.277** | 0.4386 | 0.3743 | **0.2782** | 0.4346 | 0.3503 | **0.2776** | 0.4279 | 0.3223 | **0.2785** | 0.423 | 0.3116 |
| Standard deviation | 3.5e-3 | 7.0e-3 | 1.7e-3 | 2.2e-3 | 6.6e-3 | 6.9e-4 | 3.8e-3 | 6.2e-3 | 1.6e-3 | 1.5e-3 | 5.7e-3 | 1.5e-3 |
| ETTm1 | **0.3189** | 1.0209 | 0.5224 | **0.307** | 0.9706 | 0.374 | **0.303** | 0.9274 | 0.3353 | **0.3006** | 0.8982 | 0.3229 |
| Standard deviation | 2.6e-3 | 2.4e-2 | 5.2e-3 | 1.7e-3 | 2.1e-2 | 3.2e-3 | 1.0e-3 | 1.9e-2 | 5.1e-4 | 1.1e-3 | 1.8e-2 | 8.8e-4 |
| ETTm2 | **0.1679** | 0.3525 | 0.2738 | **0.164** | 0.3444 | 0.2182 | **0.1633** | 0.3368 | 0.1942 | **0.1631** | 0.3297 | 0.1862 |
| Standard deviation | 2.9e-3 | 6.1e-3 | 2.2e-3 | 1.8e-3 | 5.6e-3 | 1.6e-3 | 1.3e-3 | 5.3e-3 | 1.3e-3 | 7.8e-4 | 4.9e-3 | 8.5e-4 |
| Weather | **0.1461** | 0.3116 | 0.1721 | **0.1447** | 0.2956 | 0.1501 | **0.1443** | 0.2823 | 0.1476 | **0.1443** | 0.2654 | 0.1488 |
| Standard deviation | 1.6e-4 | 3.5e-3 | 3.2e-3 | 1.2e-4 | 2.8e-3 | 6.2e-4 | 1.2e-4 | 2.2e-3 | 7.4e-4 | 2.9e-5 | 2.0e-3 | 7.5e-4 |
| Electricity | **0.1319** | 0.3355 | 0.1402 | **0.1309** | 0.246 | 0.1329 | **0.1309** | 0.2179 | 0.1321 | **0.1306** | 0.1989 | 0.1321 |
| Standard deviation | 2.6e-4 | 1.4e-4 | 2.9e-4 | 1.8e-4 | 5.8e-4 | 1.9e-5 | 2.3e-4 | 2.7e-4 | 2.6e-4 | 1.3e-4 | 1.5e-4 | 3.2e-4 |
| Traffic | **0.3576** | 0.6198 | 0.3639 | **0.3518** | 0.503 | 0.3588 | **0.3508** | 0.4671 | 0.3578 | **0.349** | 0.4505 | 0.3571 |
| Standard deviation | 2.8e-5 | 2.3e-3 | 7.1e-5 | 9.8e-4 | 8.0e-4 | 8.3e-4 | 1.2e-3 | 4.4e-4 | 5.8e-4 | 1.9e-3 | 4.2e-4 | 1.2e-3 |
| Avg. Improvements | - | 57.17% | 31.11% | - | 53.5% | 21.96% | - | 51.22% | 15.9% | - | 49.36% | 12.45% |
| Max. Improvements | - | 85.58% | 81.71% | - | 85.65% | 73.09% | - | 85.71% | 62.81% | - | 85.43% | 50.3% |

Table 30: Full standard deviation and **MSE** of comparing our smoothed full fine-tuning (SFF) with linear-probing (LP) and linear-probing then full fine-tuning (LPFF) under **25%, 50%, 75%, and 100%** proportion of available data.

| Data proportion | 25% | | | 50% | | | 75% | | | 100% | | |
|---|---|---|---|---|---|---|---|---|---|---|---|---|
| Methods | SFF | LP | LPFF | SFF | LP | LPFF | SFF | LP | LPFF | SFF | LP | LPFF |
| Exchange | **0.0811** | 0.5541 | 0.1377 | **0.0809** | 0.5163 | 0.1011 | **0.0819** | 0.4667 | 0.0958 | **0.0809** | 0.439 | 0.0917 |
| Standard deviation | 4.5e-4 | 6.1e-3 | 4.7e-3 | 5.4e-4 | 5.4e-3 | 2.7e-3 | 1.2e-3 | 4.8e-3 | 2.0e-3 | 7.6e-3 | 3.8e-3 | 9.2e-4 |
| ETTh1 | **0.351** | 0.7801 | 0.3918 | **0.3509** | 0.695 | 0.3724 | **0.3513** | 0.623 | 0.3769 | **0.3567** | 0.5888 | 0.37 |
| Standard deviation | 6.1e-4 | 6.6e-3 | 2.4e-3 | 1.1e-3 | 5.1e-3 | 1.2e-3 | 1.4e-3 | 4.4e-3 | 1.5e-3 | 1.4e-3 | 4.0e-3 | 2.8e-3 |
| ETTh2 | **0.2745** | 0.4192 | 0.3001 | **0.2757** | 0.3985 | 0.2977 | **0.2778** | 0.3843 | 0.3034 | **0.274** | 0.372 | 0.2991 |
| Standard deviation | 2.5e-3 | 5.5e-3 | 1.4e-3 | 2.0e-3 | 4.0e-3 | 1.6e-3 | 4.8e-4 | 3.0e-3 | 1.9e-3 | 3.8e-4 | 2.4e-3 | 1.7e-3 |
| ETTm1 | **0.2995** | 0.8669 | 0.3152 | **0.2979** | 0.7422 | 0.3103 | **0.2974** | 0.6467 | 0.3136 | **0.2976** | 0.5935 | 0.3132 |
| Standard deviation | 1.1e-3 | 1.6e-2 | 6.4e-4 | 1.6e-3 | 1.1e-2 | 8.8e-4 | 1.3e-3 | 8.7e-3 | 1.5e-3 | 1.5e-3 | 7.2e-3 | 1.3e-3 |
| ETTm2 | **0.1608** | 0.3226 | 0.1738 | **0.1609** | 0.2947 | 0.1751 | **0.1627** | 0.2771 | 0.1863 | **0.1613** | 0.2595 | 0.1798 |
| Standard deviation | 9.7e-4 | 4.6e-3 | 7.9e-4 | 3.8e-4 | 3.4e-3 | 8.4e-4 | 4.3e-4 | 2.7e-3 | 1.2e-3 | 1.0e-3 | 2.1e-3 | 2.9e-3 |
| Weather | **0.144** | 0.2619 | 0.1479 | **0.1443** | 0.2248 | 0.1497 | **0.1468** | 0.192 | 0.1624 | **0.1451** | 0.1871 | 0.1573 |
| Standard deviation | 5.2e-5 | 1.5e-3 | 1.9e-4 | 2.1e-4 | 1.0e-3 | 7.2e-4 | 1.3e-4 | 7.9e-4 | 1.7e-3 | 7.3e-4 | 6.5e-4 | 9.1e-4 |
| Electricity | **0.1303** | 0.1902 | 0.1317 | **0.1301** | 0.1654 | 0.1324 | **0.13** | 0.1538 | 0.1345 | **0.1304** | 0.1491 | 0.1335 |
| Standard deviation | 1.7e-4 | 1.5e-4 | 2.0e-4 | 2.4e-4 | 1.0e-4 | 1.2e-4 | 3.6e-4 | 9.5e-5 | 9.7e-4 | 2.0e-4 | 6.9e-5 | 6.4e-4 |
| Traffic | **0.3488** | 0.441 | 0.3567 | **0.3497** | 0.4164 | 0.3564 | **0.3502** | 0.406 | 0.3587 | **0.3551** | 0.4014 | 0.3574 |
| Standard deviation | 2.3e-3 | 4.5e-4 | 4.8e-4 | 1.7e-3 | 2.5e-4 | 7.4e-4 | 2.5e-3 | 1.2e-4 | 3.3e-4 | 7.1e-5 | 7.5e-5 | 5.7e-4 |
| Avg. Improvements | - | 48.49% | 9.8% | - | 42.89% | 6.56% | - | 37.73% | 7.86% | - | 35.19% | 6.22% |
| Max. Improvements | - | 85.36% | 41.1% | - | 84.33% | 19.98% | - | 82.45% | 14.51% | - | 81.57% | 11.78% |

Table 31: Full standard deviation and **MAE** of comparing our smoothed full fine-tuning (SFF) with linear-probing (LP) and linear-probing then full fine-tuning (LPFF) under **1%, 2%, 3%, and 4%** proportion of available data.

| Data proportion | 1% | | | 2% | | | 3% | | | 4% | | |
|---|---|---|---|---|---|---|---|---|---|---|---|---|
| Methods | SFF | LP | LPFF | SFF | LP | LPFF | SFF | LP | LPFF | SFF | LP | LPFF |
| Exchange | **0.2051** | 0.5867 | 0.5309 | **0.2052** | 0.5867 | 0.528 | **0.204** | 0.5867 | 0.5272 | **0.2041** | 0.5866 | 0.5267 |
| Standard deviation | 2.3e-4 | 2.6e-3 | 1.5e-3 | 4.8e-4 | 2.6e-3 | 1.1e-3 | 3.5e-4 | 2.6e-3 | 1.5e-3 | 4.3e-4 | 2.6e-3 | 1.4e-3 |
| ETTh1 | **0.4019** | 0.6441 | 0.5884 | **0.401** | 0.644 | 0.5861 | **0.3992** | 0.6439 | 0.5851 | **0.4012** | 0.6409 | 0.6001 |
| Standard deviation | 3.3e-3 | 4.8e-3 | 1.2e-3 | 3.4e-3 | 4.8e-3 | 1.2e-3 | 3.5e-3 | 4.8e-3 | 1.4e-3 | 5.3e-3 | 4.4e-3 | 1.5e-3 |
| ETTh2 | **0.3389** | 0.4601 | 0.4379 | **0.3323** | 0.46 | 0.4362 | **0.3326** | 0.46 | 0.4356 | **0.3344** | 0.4581 | 0.4267 |
| Standard deviation | 2.0e-3 | 4.1e-3 | 2.3e-3 | 3.2e-4 | 4.1e-3 | 2.0e-3 | 5.2e-4 | 4.1e-3 | 2.0e-3 | 5.2e-4 | 3.9e-3 | 1.5e-3 |
| ETTm1 | **0.412** | 0.7239 | 0.6419 | **0.3875** | 0.7179 | 0.5773 | **0.3835** | 0.7176 | 0.5717 | **0.378** | 0.7119 | 0.5213 |
| Standard deviation | 2.5e-3 | 9.1e-3 | 3.6e-3 | 2.6e-3 | 8.8e-3 | 1.4e-3 | 1.9e-3 | 8.9e-3 | 1.3e-3 | 1.8e-3 | 8.6e-3 | 8.9e-4 |
| ETTm2 | **0.2628** | 0.3966 | 0.3717 | **0.2617** | 0.3965 | 0.3709 | **0.2559** | 0.3942 | 0.3483 | **0.2564** | 0.3942 | 0.347 |
| Standard deviation | 3.0e-3 | 3.6e-3 | 2.0e-3 | 1.9e-3 | 3.6e-3 | 1.7e-3 | 2.5e-3 | 3.5e-3 | 1.5e-3 | 1.6e-3 | 3.5e-3 | 1.1e-3 |
| Weather | **0.2034** | 0.3607 | 0.3198 | **0.1999** | 0.3584 | 0.2994 | **0.1987** | 0.3562 | 0.2779 | **0.1957** | 0.3539 | 0.2567 |
| Standard deviation | 5.6e-4 | 3.0e-3 | 1.7e-3 | 2.6e-4 | 2.8e-3 | 3.2e-3 | 7.0e-4 | 2.7e-3 | 3.7e-3 | 2.3e-4 | 2.6e-3 | 5.8e-3 |
| Electricity | **0.2342** | 0.6951 | 0.3184 | **0.2307** | 0.5986 | 0.2729 | **0.229** | 0.5233 | 0.256 | **0.2292** | 0.4676 | 0.2476 |
| Standard deviation | 8.3e-5 | 2.5e-3 | 5.1e-4 | 3.4e-4 | 2.3e-3 | 8.5e-4 | 5.0e-4 | 2.0e-3 | 5.6e-4 | 2.7e-4 | 1.4e-3 | 4.4e-4 |
| Traffic | **0.2672** | 0.7112 | 0.3433 | **0.2611** | 0.59 | 0.2946 | **0.2575** | 0.5175 | 0.2776 | **0.2559** | 0.4736 | 0.2694 |
| Standard deviation | 1.1e-4 | 5.1e-4 | 4.6e-4 | 4.7e-6 | 1.5e-4 | 1.5e-4 | 4.7e-6 | 7.8e-4 | 7.1e-5 | 9.4e-5 | 1.2e-3 | 8.5e-5 |
| Avg. Improvements | - | 47.27% | 33.23% | - | 46.5% | 29.86% | - | 45.41% | 27.81% | - | 44.14% | 25.73% |
| Max. Improvements | - | 66.31% | 61.37% | - | 65.02% | 61.14% | - | 65.23% | 61.31% | - | 65.21% | 61.25% |

Table 32: Full standard deviation and **MAE** of comparing our smoothed full fine-tuning (SFF) with linear-probing (LP) and linear-probing then full fine-tuning (LPFF) under **5%, 10%, 15%, and 20%** proportion of available data.

| Data proportion | 5% | | | 10% | | | 15% | | | 20% | | |
|---|---|---|---|---|---|---|---|---|---|---|---|---|
| Methods | SFF | LP | LPFF | SFF | LP | LPFF | SFF | LP | LPFF | SFF | LP | LPFF |
| Exchange | **0.2046** | 0.5863 | 0.5221 | **0.2031** | 0.5816 | 0.4169 | **0.2017** | 0.5766 | 0.3476 | **0.2019** | 0.5705 | 0.298 |
| Standard deviation | 2.0e-4 | 2.6e-3 | 1.0e-3 | 9.0e-4 | 2.6e-3 | 9.3e-3 | 6.1e-4 | 2.6e-3 | 8.9e-3 | 1.0e-3 | 2.5e-3 | 5.7e-3 |
| ETTh1 | **0.3971** | 0.6374 | 0.5467 | **0.3907** | 0.6308 | 0.5046 | **0.3904** | 0.6203 | 0.4575 | **0.3874** | 0.6122 | 0.4314 |
| Standard deviation | 3.4e-3 | 4.6e-3 | 4.9e-3 | 1.8e-3 | 4.4e-3 | 5.1e-3 | 1.5e-3 | 3.8e-3 | 2.8e-3 | 1.4e-3 | 3.9e-3 | 2.1e-3 |
| ETTh2 | **0.3339** | 0.4576 | 0.4186 | **0.3359** | 0.4554 | 0.4008 | **0.3365** | 0.4515 | 0.3769 | **0.3367** | 0.4485 | 0.3674 |
| Standard deviation | 1.2e-3 | 3.9e-3 | 8.9e-4 | 2.3e-3 | 3.7e-3 | 4.2e-4 | 1.7e-3 | 3.6e-3 | 1.3e-3 | 2.6e-3 | 3.2e-3 | 1.3e-3 |
| ETTm1 | **0.3753** | 0.7116 | 0.5149 | **0.3674** | 0.6949 | 0.4256 | **0.3646** | 0.6801 | 0.3959 | **0.3642** | 0.6701 | 0.3869 |
| Standard deviation | 1.3e-3 | 8.6e-3 | 8.9e-4 | 8.2e-4 | 7.8e-3 | 1.8e-3 | 3.1e-4 | 7.1e-3 | 1.5e-4 | 8.2e-4 | 6.8e-3 | 5.4e-4 |
| ETTm2 | **0.2544** | 0.3941 | 0.3458 | **0.2534** | 0.3896 | 0.3057 | **0.2539** | 0.3852 | 0.2866 | **0.2533** | 0.3811 | 0.2807 |
| Standard deviation | 1.6e-3 | 3.5e-3 | 1.1e-3 | 1.3e-3 | 3.3e-3 | 1.0e-3 | 5.0e-4 | 3.1e-3 | 1.4e-3 | 4.1e-4 | 2.9e-3 | 1.4e-3 |
| Weather | **0.1939** | 0.3497 | 0.2289 | **0.193** | 0.3381 | 0.2038 | **0.193** | 0.3282 | 0.1996 | **0.1935** | 0.3161 | 0.2025 |
| Standard deviation | 2.3e-4 | 2.4e-3 | 3.0e-3 | 1.4e-4 | 1.9e-3 | 6.9e-4 | 1.3e-4 | 1.5e-3 | 4.3e-4 | 3.0e-4 | 1.4e-3 | 9.5e-4 |
| Electricity | **0.2273** | 0.4226 | 0.2394 | **0.2263** | 0.3427 | 0.2291 | **0.226** | 0.3184 | 0.2274 | **0.225** | 0.3021 | 0.2266 |
| Standard deviation | 9.1e-5 | 7.8e-4 | 3.2e-4 | 3.2e-4 | 3.1e-4 | 1.2e-4 | 1.8e-4 | 1.5e-4 | 1.6e-4 | 4.5e-4 | 4.5e-5 | 2.6e-5 |
| Traffic | **0.2537** | 0.4449 | 0.2639 | **0.2501** | 0.3808 | 0.2557 | **0.2488** | 0.3548 | 0.2515 | **0.2473** | 0.3404 | 0.2511 |
| Standard deviation | 7.5e-5 | 1.3e-3 | 2.8e-5 | 1.0e-3 | 3.1e-4 | 1.9e-4 | 1.4e-3 | 9.9e-5 | 3.3e-5 | 1.6e-3 | 2.0e-4 | 1.8e-3 |
| Avg. Improvements | - | 43.28% | 23.27% | - | 40.34% | 16.19% | - | 38.52% | 11.46% | - | 37.14% | 9.14% |
| Max. Improvements | - | 65.1% | 60.81% | - | 65.08% | 51.28% | - | 65.02% | 41.97% | - | 64.61% | 32.25% |

Table 33: Full standard deviation and **MAE** of comparing our smoothed full fine-tuning (SFF) with linear-probing (LP) and linear-probing then full fine-tuning (LPFF) under **25%, 50%, 75%, and 100%** proportion of available data.

| Data proportion | 25% | | | 50% | | | 75% | | | 100% | | |
|---|---|---|---|---|---|---|---|---|---|---|---|---|
| Methods | SFF | LP | LPFF | SFF | LP | LPFF | SFF | LP | LPFF | SFF | LP | LPFF |
| Exchange | **0.2004** | 0.567 | 0.2689 | **0.2003** | 0.5476 | 0.2251 | **0.2026** | 0.5206 | 0.2204 | **0.2013** | 0.505 | 0.2151 |
| Standard deviation | 5.3e-4 | 2.3e-3 | 5.3e-3 | 1.0e-3 | 2.3e-3 | 3.7e-3 | 1.7e-3 | 2.2e-3 | 2.5e-3 | 5.3e-4 | 2.0e-3 | 2.0e-3 |
| ETTh1 | **0.3883** | 0.6078 | 0.4238 | **0.3883** | 0.5753 | 0.4091 | **0.3892** | 0.5457 | 0.4127 | **0.394** | 0.5301 | 0.4074 |
| Standard deviation | 4.5e-4 | 3.7e-3 | 1.3e-3 | 3.0e-4 | 3.0e-3 | 1.5e-3 | 2.4e-3 | 2.8e-3 | 9.0e-4 | 1.4e-3 | 2.7e-3 | 6.8e-4 |
| ETTh2 | **0.3343** | 0.4463 | 0.3571 | **0.3346** | 0.4337 | 0.3536 | **0.339** | 0.4245 | 0.3599 | **0.3359** | 0.4163 | 0.3539 |
| Standard deviation | 1.3e-3 | 3.1e-3 | 1.3e-3 | 1.4e-3 | 2.4e-3 | 1.4e-3 | 1.8e-4 | 1.8e-3 | 1.4e-3 | 5.7e-4 | 1.5e-3 | 8.3e-4 |
| ETTm1 | **0.3623** | 0.6584 | 0.3794 | **0.3602** | 0.6101 | 0.3731 | **0.3595** | 0.5709 | 0.3797 | **0.358** | 0.5444 | 0.3748 |
| Standard deviation | 1.7e-3 | 6.2e-3 | 3.5e-4 | 1.8e-3 | 4.4e-3 | 5.6e-4 | 2.0e-3 | 3.4e-3 | 7.2e-4 | 1.6e-3 | 2.9e-3 | 6.7e-4 |
| ETTm2 | **0.2499** | 0.3768 | 0.2661 | **0.2504** | 0.3601 | 0.2653 | **0.2539** | 0.3494 | 0.2797 | **0.2493** | 0.3375 | 0.2683 |
| Standard deviation | 6.4e-4 | 2.7e-3 | 1.2e-3 | 5.8e-4 | 2.1e-3 | 9.2e-4 | 3.8e-4 | 1.7e-3 | 1.3e-3 | 1.5e-3 | 1.4e-3 | 2.3e-3 |
| Weather | **0.1925** | 0.3125 | 0.1999 | **0.193** | 0.2815 | 0.2002 | **0.1975** | 0.252 | 0.2164 | **0.1934** | 0.2459 | 0.2049 |
| Standard deviation | 2.9e-4 | 1.0e-3 | 5.9e-4 | 2.1e-4 | 6.9e-4 | 1.4e-3 | 2.9e-4 | 6.3e-4 | 2.3e-3 | 1.5e-3 | 5.2e-4 | 1.5e-3 |
| Electricity | **0.2245** | 0.2939 | 0.2268 | **0.2247** | 0.2704 | 0.2266 | **0.2245** | 0.2578 | 0.2278 | **0.2239** | 0.2516 | 0.2266 |
| Standard deviation | 4.2e-4 | 4.7e-6 | 1.7e-4 | 2.7e-4 | 5.4e-5 | 1.1e-3 | 4.3e-4 | 1.0e-4 | 6.0e-4 | 4.2e-4 | 7.9e-5 | 3.9e-4 |
| Traffic | **0.2463** | 0.3316 | 0.2532 | **0.2486** | 0.3062 | 0.2465 | **0.2485** | 0.2951 | 0.2498 | **0.245** | 0.2902 | 0.2494 |
| Standard deviation | 2.2e-3 | 2.9e-4 | 1.4e-3 | 1.4e-3 | 2.5e-4 | 4.7e-4 | 4.6e-4 | 1.4e-4 | 3.5e-4 | 0.0e+00 | 8.5e-5 | 1.8e-4 |
| Avg. Improvements | - | 36.53% | 7.28% | - | 32.17% | 4.27% | - | 28.07% | 5.6% | - | 26.68% | 4.36% |
| Max. Improvements | - | 64.66% | 25.47% | - | 63.42% | 11.02% | - | 61.08% | 9.22% | - | 60.14% | 7.08% |

