# OpenReview forum: "Lost in the Non-convex Loss Landscape: How to Fine-tune the Large Time Series Model?"
_ICLR.cc/2026/Conference — ICLR 2026 Poster_

### Official Review · Reviewer_r8RN · 2025-10-23

**Soundness:** 3
**Presentation:** 3
**Contribution:** 2
**Rating:** 4
**Confidence:** 4

**Summary:**

The paper proposes the Smoothed Full Fine-tunning (SFF) to finetune LTSMs. SFF linearly interpolate the parameters of a pretrained LTSM with a randomly initialized auxiliary model, with the goal of smoothing sharp, non-convex regions of the loss landscape. The main contribution lies in the interpolation that perturbs the parameter of the model to reach flatter regions. Such perturbation towards flat minima is a well-known strategy for enhance generalization. Experimental results show the method can obtain better results than compared baselines and that SFF can perform fine-tuning with minimal memory and computational overhead.

**Strengths:**

- The authors identify a significant and practical problem, the poor fine-tuning performance of modern LTSMs due to the non-convexity of their loss landscape.
- SFF is simple to implement, requiring only a one-time linear interpolation of weights before fine-tuning. It adds no computational or memory overhead during the actual fine-tuning process.

**Weaknesses:**

- Exploration of interpolation strategies. The paper states that $\alpha$ controls the proportion of pre-trained knowledge retained, and the experiments show a sensitivity to $\alpha$. However, the optimal $\alpha$ appears to be model-dependent. A more robust, perhaps adaptive, method for selecting or tuning $\alpha$ would significantly strengthen the work.

- The auxiliary model is defined as a randomly initialized LTSM. The paper claims this model has a smooth loss landscape, however is not clear why a randomly initialized model is guaranteed to be smoother than a pre-trained one, especially considering the vast differences in initialization schemes and model architectures. A more rigorous analysis or reference to why random initialization leads to flat minima would be beneficial.

- The theoretical analysis in Section 3.1 is relatively brief and qualitative, relying heavily on existing literature. I suggest the authors to present a more in depth analysis of the loss landscape for finetuning, such analysis would provide strong theoretical justification.

**Questions:**

Refer to the weakness part.

- Can the authors provide a more detailed explanation or empirical evidence as to why a randomly initialized LTSM is guaranteed to have a smoother loss landscape than a pre-trained one?

---

> ### Author Response · Authors · 2025-11-26
> **Response to Weakness 1**
>
> **Weakness 1**: Exploration of interpolation strategies. The paper states that $\alpha$ controls the proportion of pre-trained knowledge retained, and the experiments show a sensitivity to $\alpha$. However, the optimal $\alpha$ appears to be model-dependent. A more robust, perhaps adaptive, method for selecting or tuning $\alpha$ would significantly strengthen the work.
>
> **Response to Weakness 1**: We sincerely thank the reviewer for the insightful question regarding the selection of the interpolation parameter $\alpha$. We address this from three perspectives:
>
> **(1) Empirically recommended values:** As illustrated in Figures 5 and 6 of the manuscript, although different $\alpha$ values introduce some variation, the sensitivity analysis demonstrates that SFF consistently outperforms vanilla fine-tuning across a broad range. Specifically, $\alpha \approx 0.8$ performs best for zero-shot prediction, while $\alpha \approx 0.6$ yields the strongest overall performance under full fine-tuning. These values can thus serve as reliable initial starting points.
>
> **(2) Validation-based tuning:** We observe that the trend of test performance with respect to $\alpha$ closely mirrors that on the validation set. Therefore, once a candidate search range is defined, selecting the $\alpha$ that minimizes validation error provides a straightforward and computationally efficient strategy.
>
> **(3) Data-driven selection:** Automatically learning $\alpha$ is indeed a promising direction. In the current framework, however, the interpolation weights are fixed prior to fine-tuning, which makes adaptive selection non-trivial. Approaches such as meta-learning could potentially be explored to determine optimal $\alpha$ values across models and datasets. We regard this as an important avenue for future research.

---

> ### Author Response · Authors · 2025-11-26
> **Response to Weakness 2 (Part 1)**
>
> **Weakness 2**:  The auxiliary model is defined as a randomly initialized LTSM. The paper claims this model has a smooth loss landscape, however is not clear why a randomly initialized model is guaranteed to be smoother than a pre-trained one, especially considering the vast differences in initialization schemes and model architectures. A more rigorous analysis or reference to why random initialization leads to flat minima would be beneficial.
>
> **Response to Weakness 2**:
>
> We appreciate the reviewer’s thoughtful question. Pre-trained models are known to accumulate sharp minima due to potential overfitting and task-specific specialization, often resulting in a steep and highly non-smooth loss landscape [1]. In contrast, prior work [2] has shown that parameters drawn from widely used initialization schemes—such as Kaiming [3] and Xavier [4]—tend to lie in what they term the  *“Goldilocks zone”* , a region of the loss surface characterized by flatness and favorable optimization geometry. Furthermore, **this property is not tied to the particular initialization scheme or model architecture**. Mainstream methods such as Kaiming uniform/Gaussian and Xavier uniform/Gaussian all reliably exhibit this characteristic.
>
> Building upon these findings, we provide a formal theoretical analysis demonstrating why randomly initialized parameters inherently induce a flatter and smoother local loss landscape. This theoretical justification is included to address the reviewer’s concern directly and to clarify the role of the auxiliary model within our method.
>
> Given an MSE loss function $\mathcal{L}(\Theta)$ and the $\Theta$  denotes model parameters. To obtain a minimum $\mathcal{L}(\Theta)$, the corresponding parameters $\Theta^*$ can be a sharp minimum or a flat minimum. Inspired by [1], we use the Hessian matrix to formally characterize the sharpness and flatness of the loss landscape in the following analysis.
>
> By analyzing the maximum eigenvalue $\lambda_{{max}}(\cdot)$ of the Hessian matrix $H=\nabla^2 \mathcal{L}(\Theta^*)$ ($H\in \mathbb{R}^{d \times d}$ where $d$ denotes feature dimension), we can obtain the definitions of the sharp and flat minimum as follows:
>
>
> **Theorem 1: Sharp minimum**. The Hessian $\nabla^2 \mathcal{L}(\Theta^\*)$ has large eigenvalues (i.e., $\lambda_{{max}}(\nabla^2 \mathcal{L}(\Theta^\*)) \gg \tau$ where $\tau>0$ is a threshold), meaning small parameter perturbations lead to large loss increases.
>
> **Theorem 2: Flat minimum**. The Hessian $\nabla^2 \mathcal{L}(\Theta^\*)$ has small eigenvalues (i.e., $\lambda_{\text{max}}(\nabla^2 \mathcal{L}(\Theta^\*)) \leq \tau$ where $\tau>0$ is a threshold), meaning the loss is robust to parameter perturbations.
>
>
> **Next, we provide proofs about the two above Theorems and demonstrate why mainstream Kaiming or Xavier initializations yield a flat loss landscape.**
>
>  *[1] On large-batch training for deep learning: Generalization gap and sharp minima. ICLR.*
>
> *[2] The goldilocks zone: Towards better understanding of neural network loss landscapes. AAAI.*
>
> *[3] Delving deep into rectifiers: Surpassing human-level performance on imagenet classification. ICCV.*
>
> *[4] Understanding the difficulty of training deep feedforward neural networks. AISTATS.*
>
> *[5] Sharpness-aware minimization for efficiently improving generalization. ICLR.*
>
> *[6] Flat minima. Neural computation.*

---

> ### Author Response · Authors · 2025-11-26
> **Response to Weakness 2 (Part 2)**
>
> **Proving Theorem 1 and Theorem 2 about the Sharp minimum and Flat minimum:**
>
> We start by performing a second-order Taylor expansion of the loss function $\mathcal{L}(\Theta^\*)$ at the minimum point $\Theta^\*$:
>
> \begin{equation}
> \mathcal{L}(\Theta^\* + \delta) \approx \mathcal{L}(\Theta^*) + \frac{1}{2} \delta^\top \nabla^2 \mathcal{L}(\Theta^\*) \delta \approx \frac{1}{2} \delta^\top \nabla^2 \mathcal{L}(\Theta^\*) \delta
> \end{equation}
>
> where $\delta$ is a small parameter perturbation (e.g., $|\delta| \ll 1$).
>
> The loss change after perturbation can be formulated as:
>
> \begin{equation}
> \Delta \mathcal{L} = \mathcal{L}(\Theta^\* + \delta) - \mathcal{L}(\Theta^\*) \approx \frac{1}{2} \delta^\top \nabla^2 \mathcal{L}(\Theta^*) \delta \approx \frac{1}{2} \delta^\top H \delta
> \end{equation}
>
> The largest $\Delta \mathcal{L}$ is governed by the $\lambda_{\text{max}}$ of the Hessian $H=\nabla^2 \mathcal{L}(\Theta^\*)$.
> Let unit vector $v_\text{max}$ be the eigenvector corresponding to eigenvalue $\lambda_{{max}}$. Perturbing in the direction of $v_{max}$ yields the largest $\Delta \mathcal{L}$ because $v_{max}$ represents the largest curvature [5], i.e., steepest direction, in the loss landscape. Let $\delta = \epsilon \cdot v_{\text{max}}$ ($|\epsilon| \ll 1$), the above Equation can be reformulated as:
>
> \begin{equation}
> \Delta \mathcal{L}  \approx \frac{1}{2} \epsilon \cdot v_\text{max}^\top H \epsilon \cdot v_\text{max}
> \end{equation}
>
> According to eigen equation, $Hv_{max}=\lambda_{{max}}v_{max}$, we can further obtain:
>
> \begin{equation}
> \Delta \mathcal{L}  \approx \frac{1}{2} \epsilon^2 v_\text{max}^\top \lambda_{\text{max}}v_{max}\approx \frac{1}{2} \epsilon^2 v_{max}^\top v_{max} \lambda_{{max}} \approx \frac{1}{2} \epsilon^2 \lambda_{{max}}
> \end{equation}
>
> Hence, $\Delta \mathcal{L}$ is closely related to $\lambda_{{max}}$, the larger $\lambda_{{max}}$ is, the sharper the loss landscape, so parameter perturbations in many directions can noticeably increase the loss.
> Conversely, the smaller $\lambda_{{max}}$ is, the flatter the landscape remains, and perturbations won't obviously raise the loss [6].
>
>
>
> **Next, we further demonstrate why mainstream Kaiming or Xavier initializations yield a flat loss landscape.**
>
> Specifically, prior work [2] quantifies the smoothness of the loss landscape under Kaiming and Xavier initializations by the ratio of the trace of the Hessian matrix $Tr(H)$ to its Frobenius norm $\\|H\\|_F$ and **has demonstrated $\\frac{\operatorname{Tr}(H)}{\\|H\\|_F}  \gg 1$ from both theoretical and experimental perspectives**. According to the symmetry of $H$, we explicitly expand the formula as:
>
> \begin{equation}
> \frac{\operatorname{Tr}(H)}{\|H\|\_F} = \frac{\operatorname{Tr}(H)}{\sqrt{\sum_{i=1}^d h\_i^2}}=\frac{\operatorname{Tr}(H)}{\sqrt{\operatorname{Tr}(H^TH)}}=\frac{\sum \lambda\_i}{\sqrt{\sum \lambda\_i^2}} \gg 1
> \end{equation}
>
> This indicates that the sum of eigenvalues greatly exceeds the square root of the sum of squared eigenvalues, suggesting that **most eigenvalues are positive and relatively evenly distributed rather than containing extreme outliers, i.e., most cases belong to $\lambda(\nabla^2 \mathcal{L}(\Theta^*)) \leq \tau$ where $\tau>0$**.
>
> According to the definition of **Theorem 1** and  **Theorem 2** about sharp minimum and flat minimum, this indicates that the loss surface exhibits a smooth, valley-like geometry dominated by positive curvature. As a result, most random descent directions remain stable and low-curvature, making the optimization process easier and more consistent.
>
> In contrast, if $\frac{\operatorname{Tr}(H)}{\\|H\\|_F} \lesssim 1$, it indicates a balanced mix of positive and negative eigenvalues $\lambda$, leading to a steep and unsmooth loss landscape. As a result, convergence becomes slower and the optimizer is more likely to fall into suboptimal local minima.
>
> **Therefore, Kaiming or Xavier initialization offers a stable and smooth loss landscape**. In our work, we adopt them for randomly initializing the auxiliary LSTM for smoothed full fine-tuning.
>
> **Next, we further discuss the influence of initialization schemes on fine-tuning performance.**
>
>  *[1] On large-batch training for deep learning: Generalization gap and sharp minima. ICLR.*
>
> *[2] The goldilocks zone: Towards better understanding of neural network loss landscapes. AAAI.*
>
> *[3] Delving deep into rectifiers: Surpassing human-level performance on imagenet classification. ICCV.*
>
> *[4] Understanding the difficulty of training deep feedforward neural networks. AISTATS.*
>
> *[5] Sharpness-aware minimization for efficiently improving generalization. ICLR.*
>
> *[6] Flat minima. Neural computation.*

---

> ### Author Response · Authors · 2025-11-26
> **Response to Weakness 2 (Part 3)**
>
> **Regarding to influence of different initialization schemes on fine-tuning performance.**  As shown in **Tables 1 and 2**, we conducted two experiments to investigate this issue. In **Table 1**, we have additionally introduced several more types of perturbations, e.g., standard Gaussian (mean = 0, variance = 1), Xavier Gaussian, Xavier uniform, Kaiming Gaussian, and Kaiming uniform.
> The results in **Table 1** show that Xavier- and Kaiming-based schemes maintain stable performance improvements. Because they consider the stable gradient variance and can supply a flat loss landscape (demonstrated by [1]) that is used to smooth the sharp landscape of the pre-trained model for better fine-tuning effect, which is also aligned with our Theoretical analysis. In contrast, standard Gaussian initialization—lacking variance control—often pushes parameters into sharper regions of the landscape, resulting in weaker smoothing and noticeably degraded downstream performance. These findings provide strong empirical evidence supporting our design choice.
>
> In **Table 2**, we further evaluate SFF under different random seeds while using widely adopted initialization schemes (e.g., Kaiming uniform). **Table 2** indicates that the improved performance remains highly stable across seeds. This suggests that the auxiliary LTSM of SFF doesn't rely on a carefully engineered initialization. Instead, any mainstream initialization strategy suffices to obtain consistent smoothing and fine-tuning gains. This is reasonable because the prior work [1] has proven that the underlying design of mainstream initialization methods ensures the initialized parameters indeed lie in the flat region of the loss landscape, without being influenced by the random states (seeds). We believe this robustness is a desirable property for practical deployment.
>
> **Table 1:  The effectiveness of our SFF (smoothed full fine-tuning) across different parameter initialization schemes on the LTSM Timer. We independently run four times with four random seeds to enhance the solidity of the conclusions. and report the average performance and standard deviation (The same applies to Table 2).**
> |    | Exchange|ETTh1| ETTh2|ETTm1|ETTm2|Weather|
> |:-:|:-:|:-:|:-:|:----:|:----:|:----:|
> | Original full fine-tuning|0.09$\pm$0.0007|0.367$\pm$0.0027|0.304$\pm$0.0049|0.312$\pm$0.0008|0.176$\pm$0.0013|0.158$\pm$0.0012|
> |Standard Gaussian  perturbation-SFF|5.986$\pm$0.261|0.723$\pm$0.003|1.879$\pm$0.275|3.876$\pm$0.128|19.031$\pm$0.963|0.447$\pm$0.03|
> |Kaiming Normal Distribution-SFF|0.081$\pm$0.0006|0.353$\pm$0.0009|0.277$\pm$0.001|0.299$\pm$0.0011|0.164$\pm$0.0016|0.146$\pm$0.0008|
> |Kaiming Uniform Distribution-SFF|0.081$\pm$0.0008|0.355$\pm$0.0013|0.274$\pm$0.0008|0.297$\pm$0.0016|0.161$\pm$0.0009|0.145$\pm$0.0006|
> |Xavier Normal Distribution-SFF|0.081$\pm$0.0007|0.353$\pm$0.001|0.276$\pm$0.0012|0.3$\pm$0.0009|0.162$\pm$0.0001|0.145$\pm$0.0002|
> |Xavier Uniform Distribution-SFF|0.082$\pm$0.0008|0.353$\pm$0.0007|0.277$\pm$0.0006|0.3$\pm$0.0003|0.162$\pm$0.0001|0.145$\pm$0.0002|
>
>
> **Table 2:  Experiments with four different random seeds (r1, r2,r3, and r4 here) on the LTSM Timer. The results show that our SFF (smoothed full fine-tuning) is insensitive to the choice of random initialization distribution. FF denotes original full fine-tuning.**
> |    | Exchange|ETTh1| ETTh2|ETTm1|ETTm2|Weather|
> |:-:|:-:|:-:|:-:|:----:|:----:|:----:|
> |SFF (Ours)-r1|**0.07996**|**0.3547**|**0.27379**|**0.29542**|**0.16003**|**0.14605**|
> |FF-r1|0.08937| 0.36941| 0.31021| 0.31134| 0.17645| 0.16067|
> |SFF (Ours)-r2|**0.08182**|**0.35772**|**0.27368**|**0.29902**|**0.16259**|**0.14432**|
> |FF-r2|0.09071| 0.36245| 0.3035| 0.31282| 0.17843 |0.15912
> |SFF (Ours)-r3|**0.08101**|**0.35766**|**0.27453**|**0.29824**|**0.16129**|**0.14481**|
> |FF-r3|0.09102 |0.36848 |0.29699 |0.31167 |0.17515 |0.15851|
> |SFF (Ours)-r4|**0.08191**|**0.35588**|**0.27571**|**0.29938**|**0.16173**|**0.14525**|
> |FF-r4|0.08978|0.36815|0.30666|0.31057|0.17546|0.15649|
>
> *[1] The goldilocks zone: Towards better understanding of neural network loss landscapes. AAAI.*

---

> ### Author Response · Authors · 2025-11-26
> **Response to Weakness 3**
>
> **Weakness 3**: The theoretical analysis in Section 3.1 is relatively brief and qualitative, relying heavily on existing literature. I suggest the authors to present a more in depth analysis of the loss landscape for finetuning, such analysis would provide strong theoretical justification.
>
> **Response to Weakness 3**: We sincerely thank the reviewer for bringing this important point to our attention. In the previous response, we have already provided theoretical proof that random initialization yields a smooth loss landscape. Building on that result, **we now further prove why weight interpolation can smooth sharp minima without harming flat ones:**
>
>
> **Smoothing (perturbing) sharp minima.** SFF interpolation strategy defines smoothed parameters as $\Theta_3 = \alpha\Theta_1^\* + (1-\alpha)\Theta_2$, where $\Theta_1^\*$ (pre-trained LTSM) are easier to lie in a sharp minimum (Figure 1(a) of the manuscript) due to large-scale pretraining [1]. In contrast, $\Theta_2$ (tailored Kaiming or Xavier random initializations) lies in a flat region as shown in Figure 1(b) of the manuscrip. After pre-training the LTSM, for the sharp minimum points $\Theta_1^\*$, its largest eigenvalue is $\lambda_{\text{max}}(\nabla^2 \mathcal{L}(\Theta_{1}^*)) \gg \tau $. In contrast, $\Theta_2$ is randomly initialized in a flat region, and so its largest eigenvalue is $\lambda_{\text{max}}(\nabla^2 \mathcal{L}(\Theta_{2})) \le \tau$.
>
>
> Under a local quadratic approximation of the loss function around the interpolation path, the Hessian at $\Theta_3$ can be approximated by a convex combination of the Hessians at $\Theta_1^*$  and $\Theta_2$:
>
> \begin{equation}
> \nabla^2 \mathcal{L}(\Theta_3) \approx \alpha \nabla^2 \mathcal{L}(\Theta_1^*) + (1 - \alpha) \nabla^2 \mathcal{L}(\Theta_2)
> \end{equation}
>
> Consequently, the maximum eigenvalue satisfies:
> \begin{equation}
> \lambda_{\text{max}}(\nabla^2 \mathcal{L}(\Theta_3)) \lesssim \alpha \lambda_{\text{max}}(\nabla^2 \mathcal{L}(\Theta_1^*)) + (1-\alpha) \lambda_{\text{max}}(\nabla^2 \mathcal{L}(\Theta_2))
> \end{equation}
>
> Since $\lambda_{\text{max}}(\nabla^2 \mathcal{L}(\Theta_2)) \ll \lambda_{\text{max}}(\nabla^2 \mathcal{L}(\Theta_1^\*))$  (flat vs. sharp), it follows that $\lambda_{\text{max}}(\nabla^2 \mathcal{L}(\Theta_3)) < \lambda_{\text{max}}(\nabla^2 \mathcal{L}(\Theta_1^*))$ for $\alpha \in (0,1)$.
> This suggests that interpolation reduces local sharpness, i.e., helps "sharp weights'' escape the non-flat regions for a smoother one and a better fine-tuning effect.
> This provides theoretical support for SFF’s smoothing effect.
>
>
> **Preservation of flat regions.** After pre-training the LTSM, for the flat minimum points $\bar \Theta_1^*$, interpolation preserves its flatness. Specifically, both $\bar \Theta_1^\*$ and $\Theta_2$ lie in flat regions of the loss landscape, i.e.,
>
> \begin{equation}
> \lambda_{\text{max}}\left( \nabla^2 \mathcal{L}(\bar \Theta_1^*) \right) \leq \tau, \quad \lambda_{\text{max}}\left( \nabla^2 \mathcal{L}(\Theta_2) \right) \leq \tau
> \end{equation}
>
> Similarly, the Hessian at the interpolated point $\Theta_3=\alpha \bar \Theta_1^\*+(1-\alpha)\Theta_2$ satisfies  $\nabla^2 \mathcal{L}(\Theta_3) \approx \alpha \nabla^2 \mathcal{L}(\bar \Theta_1^*) + (1 - \alpha) \nabla^2 \mathcal{L}(\Theta_2)$.
>
> Consequently, we can derive that:
>
> \begin{equation}
> \lambda_{\text{max}}\left( \nabla^2 \mathcal{L}(\Theta_3) \right) \lesssim \alpha \lambda_{\text{max}}\left( \nabla^2 \mathcal{L}(\bar \Theta_1^\*) \right) + (1 - \alpha) \lambda_{\text{max}}\left( \nabla^2 \mathcal{L}(\Theta_2) \right) \leq \alpha \tau + (1 - \alpha) \tau = \tau
> \end{equation}
>
> Hence, $\Theta_3$ remains in a flat region. According to [2], this indicates that interpolation does not harm existing flat minima $\bar\Theta_1^\* $.
>
> In summary, from the perspective of mathematical rigor, we ensure that interpolation perturbs sharp minima without obviously harming originally flat regions, thereby aiding sharp minima escape to better and smoother basins. **The complete theoretical proofs or depth analysis about Section 3.1 in all responses are organized into lines 182 to 257 and lines 772 to 804 of the revised manuscript.**
>
> We hope that the above clarification and discussion address your concerns about the in-depth analysis and theoretical justification for our work.
>
> *[1] On large-batch training for deep learning: Generalization gap and sharp minima. ICLR.*
>
> *[2] Sharpness-aware minimization for efficiently improving generalization. ICLR.*

---

> > ### Author Response · Authors · 2025-11-26
> > **Response to Questions**
> >
> > **Q1:** Can the authors provide a more detailed explanation or empirical evidence as to why a randomly initialized LTSM is guaranteed to have a smoother loss landscape than a pre-trained one?
> >
> > **Response to Q1**:
> >
> > We sincerely thank the reviewer for this important question. We have provided a detailed theoretical analysis explaining why a randomly initialized LTSM exhibits a smoother loss landscape compared to a pre-trained model. The core theoretical proof is as follows:
> >
> > Specifically, prior work [1] quantifies the smoothness of the loss landscape under Kaiming and Xavier initializations by the ratio of the trace of the Hessian matrix $Tr(H)$ to its Frobenius norm $\\|H\\|_F$ and **has demonstrated $\\frac{\operatorname{Tr}(H)}{\\|H\\|_F}  \gg 1$ from both theoretical and experimental perspectives**. According to the symmetry of $H$, we explicitly expand the formula as:
> >
> > \begin{equation}
> > \frac{\operatorname{Tr}(H)}{\|H\|\_F} = \frac{\operatorname{Tr}(H)}{\sqrt{\sum_{i=1}^d h\_i^2}}=\frac{\operatorname{Tr}(H)}{\sqrt{\operatorname{Tr}(H^TH)}}=\frac{\sum \lambda\_i}{\sqrt{\sum \lambda\_i^2}} \gg 1
> > \end{equation}
> >
> > This indicates that the sum of eigenvalues greatly exceeds the square root of the sum of squared eigenvalues, suggesting that **most eigenvalues are positive and relatively evenly distributed rather than containing extreme outliers, i.e., most cases belong to $\lambda(\nabla^2 \mathcal{L}(\Theta^*)) \leq \tau$ where $\tau>0$**.
> >
> > According to the definition of **Theorem 1** and  **Theorem 2** about sharp minimum and flat minimum ( Please refer to Response to Weakness 2), this indicates that the loss surface exhibits a smooth, valley-like geometry dominated by positive curvature. As a result, most random descent directions remain stable and low-curvature, making the optimization process easier and more consistent.
> >
> > In contrast, if $\frac{\operatorname{Tr}(H)}{\\|H\\|_F} \lesssim 1$, it indicates a balanced mix of positive and negative eigenvalues $\lambda$, leading to a steep and unsmooth loss landscape. As a result, convergence becomes slower and the optimizer is more likely to fall into suboptimal local minima.
> >
> > **Therefore, Kaiming or Xavier initialization offers a stable and smooth loss landscape**. In our work, we adopt them for randomly initializing the auxiliary LSTM for smoothed full fine-tuning.
> >
> >
> > **For a comprehensive discussion, we kindly refer the reviewer to our Response to Weakness 2 (Part 1) and Response to Weakness 2 (Part 2).**
> >
> > *[1] The goldilocks zone: Towards better understanding of neural network loss landscapes. AAAI.*

---

### Official Review · Reviewer_wYp2 · 2025-10-28

**Soundness:** 3
**Presentation:** 3
**Contribution:** 3
**Rating:** 4
**Confidence:** 4

**Summary:**

The paper proposes Smoothed Full Fine-tuning (SFF), a lightweight method to improve fine-tuning of large time-series models by interpolating pretrained weights with a randomly initialized model before training. This simple step smooths sharp loss landscapes, enhancing optimization without extra computational cost. Tested on eight major LTSMs across forecasting and anomaly detection tasks, SFF consistently improves fine-tuning and often boosts zero-shot accuracy. The approach is practical and broadly effective, though the paper’s novelty is mainly within time-series models and lacks comparisons to established flatness-based fine-tuning methods.

**Strengths:**

- The method is simple and inexpensive. Smoothing is done with a one-shot linear interpolation using a randomly initialized copy before fine-tuning. It only needs a few lines of PyTorch and does not add compute or memory cost.
- The motivation is clear. The paper argues that pretraining can leave large time-series models stuck in sharp, non-convex regions, and interpolation helps escape these while keeping flat regions stable.
- The evaluation covers many settings. The authors test eight large time-series models across forecasting, anomaly detection, and imputation tasks.

**Weaknesses:**

- The novelty claim may be overstated. The idea of mixing weights for fine-tuning already exists in other domains, so the contribution should be limited to time-series foundation models.
- The baselines are limited. The comparisons use full fine-tuning and linear probing but omit other regularization or smoothing approaches such as SAM, SWA, Mixout, or L2-SP.
- Some zero-shot results decline after smoothing. Models like MOIRAI and Chronos perform worse on certain datasets.
- Interpolation with a random model might cause misalignment. It can disturb normalization or scale between layers, and the paper does not analyze these risks in detail.

**Questions:**

Please address the identified weaknesses and limitations noted above.

---

> ### Author Response · Authors · 2025-11-26
> **Response to Weakness 1 and Weakness 2**
>
> **Weakness 1**: The novelty claim may be overstated. The idea of mixing weights for fine-tuning already exists in other domains, so the contribution should be limited to time-series foundation models.
>
> **Response to Weakness 1**: **We would like to respectfully clarify that, to the best of our knowledge, weight mixing has not previously been explored as a mechanism for improving fine-tuning**. Prior work has primarily employed weight interpolation for purposes such as model ensembling or continual learning, rather than for modifying the loss geometry of a pre-trained model. Our contribution is, to our understanding, the first to show that interpolating weights can explicitly smooth the pre-trained loss landscape, which in turn leads to more stable and effective fine-tuning.
> The key distinctions between our method and existing interpolation-based ensemble approaches are summarized below:
>
> **(1) Different goals:** Existing interpolation methods are primarily designed for model ensembling—e.g., interpolating among multiple **well-trained models** to improve generalization or mitigate catastrophic forgetting. **In contrast, our method (SFF) leverages interpolation to *smooth the loss landscape of the pretrained model* by a **randomly initialized model**, thereby making it more trainable during fine-tuning.** From another perspective, SFF’s effect is similar to adding momentum (similar to the concept of momentum in the Adam) through random weight interpolation, allowing them to escape the non-flat regions for a smoother one and a better fine-tuning effect.
>
>
>
> **(2) Different pipelines:**
> Previous works typically use the interpolated model directly for downstream tasks without further training.
> In our case, the pretrained model begins with a steep, irregular loss landscape. After interpolation smooths this landscape, **we proceed with additional fine-tuning to utilize the smoothing effect for better performance**.
>
> **(3) New theoretical analysis:** Our method is built upon a key conceptual contrast: the flat, smooth loss landscape of a randomly initialized model versus the steep, unsmooth loss landscape of a pretrained one. We formalize this contrast through theoretical analysis and proof (please refer to **lines 182 to 257 and lines 772 to 804 of the revised manuscript**), which further shows—also theoretically—why our interpolation strategy can effectively exploit this difference to enhance fine-tuning performance.
>
> We have revised the related work section in **lines 125 to 144** to explicitly discuss the differences between SFF and prior weight interpolation/averaging methods, highlighting our unique contributions. We hope this clarification addresses your concerns and demonstrates the novelty of our work.
>
> **Weakness 2**: The baselines are limited. The comparisons use full fine-tuning and linear probing but omit other regularization or smoothing approaches such as SAM, SWA, Mixout, or L2-SP.
>
> **Response to Weakness 2:** We thank the reviewer for this valuable suggestion. In the pre-train then fine-tune setting, these optimization strategies refine pre-trained weights through different forms of regularization—Sharpness-Aware Minimization (SAM), Stochastic Weight Averaging (SWA), Mixout (weight replacement), and parameter-constrained fine-tuning (L2-SP). We have now included all of them in **Table 1** for a comprehensive comparison.
>
> While SAM yields a slight improvement over vanilla fine-tuning, the overall performance gains remain limited, and all of them fall short of our method. This suggests that they don't adequately address the underlying issue: the steep loss landscape inherited from pre-training. In contrast, SFF explicitly mitigates this problem by first smoothing the landscape and then fine-tuning, resulting in substantially stronger fine-tuning and adaptation performance.
>
>
> **Table 1:  Compare our method with more baselines with four random seeds on the LTSM Timer. We report both the mean and the standard deviation. The second-best method is enclosed in square brackets [$\cdot$].**
> |    | Exchange|ETTh1| ETTh2|ETTm1|ETTm2|Weather|
> |:-:|:-:|:-:|:-:|:----:|:----:|:----:|
> | Original full fine-tuning|0.09$\pm$0.0007|0.367$\pm$0.0027|0.304$\pm$0.0049|0.312$\pm$0.0008|0.176$\pm$0.0013|0.158$\pm$0.0012|
> |SAM|[0.088]$\pm$0.0016|[0.362]$\pm$0.003|[0.296]$\pm$0.0027|[0.309]$\pm$0.0002|[0.175]$\pm$0.0017|[0.157]$\pm$0.0|
> |SWA|0.094$\pm$0.0017|0.366$\pm$0.0024|0.304$\pm$0.0045|0.319$\pm$0.0009|0.178$\pm$0.0014|0.162$\pm$0.0005|
> |MixOut|0.09$\pm$0.0003|0.376$\pm$0.0006|0.297$\pm$0.0001|0.348$\pm$0.0018|0.184$\pm$0.0006|0.16$\pm$0.0007|
> |L2-SP|0.09$\pm$0.0007|0.368$\pm$0.0031|0.304$\pm$0.0051|0.315$\pm$0.0007|0.177$\pm$0.0013|0.16$\pm$0.0004|
> |Ours (SFF)|**0.081**$\pm$0.0008|**0.355**$\pm$0.0013|**0.274**$\pm$0.0008|**0.297**$\pm$0.0016|**0.161**$\pm$0.0009|**0.145**$\pm$0.0006|

---

> ### Author Response · Authors · 2025-11-26
> **Response to Weakness 3 and Weakness 4**
>
> **Weakness 3**: Some zero-shot results decline after smoothing. Models like MOIRAI and Chronos perform worse on certain datasets. We greatly appreciate you bringing this zero-shot performance variation to our attention.
>
> **Response to Weakness 3:** We sincerely thank the reviewer for noticing this. Upon inspection, we found that a minor oversight in the numerical ordering caused this apparent decline. Originally, the table was structured in the style of Table 7, with headers grouped as “XX-SFF / XX-FF” (e.g., MOIRAI-SFF and MOIRAI-FF). Later, to align the presentation with Table 6 (zero-shot task), we changed the headers to “MOIRAI / +Smooth” **but inadvertently didn't reorder the corresponding values**.
>
> We have now corrected the numerical sequence, and smoothing the loss landscape indeed improves zero-shot performance for models such as MOIRAI and Chronos. We appreciate the reviewer’s careful observation, which allows us to identify and fix this layout oversight.
>
>
>
>
> **Weakness 4**: Interpolation with a random model might cause misalignment. It can disturb normalization or scale between layers, and the paper does not analyze these risks in detail.
>
> **Response to Weakness 4**: We appreciate the reviewer highlighting this important consideration. We discuss the issue in two scenarios:
>
> **(1) Fine-tuning after loss landscape smoothing**: When weights are smoothed, and the model is subsequently fine-tuned, the potential mismatch in normalization or scale is negligible. The model is free to update the relevant parameters during fine-tuning, effectively correcting any minor discrepancies introduced by interpolation.
>
> **(2) Zero-shot forecasting after loss landscape smoothing**: First, the random initializations used for smoothing follow standard schemes (e.g., Kaiming, Xavier), whose typical scales are consistent with the learnable scale parameters in normalization layers. Second, the “flat” and “sharp” minima we analyze encompass the entire parameter space, including the weight matrices of normalization layers. Consequently, in theory, our method does not introduce significant scale or alignment inconsistencies. Instead, it smooths sharp minima located in suboptimal regions without harming flat minima, effectively relocating the sharp minima of the normalization layers to more favorable convergence points and thereby achieving improved and more generalizable performance. Moreover, we have included formal theoretical derivations and analyses demonstrating that the interpolation strategy improves sharp minima while not harming flat minima. For details, please refer to **lines 182 to 257 and lines 772 to 804 of the revised manuscript**.
>
> Moreover, empirically, as shown in Tables 6 and 7 (after correcting the minor numerical ordering oversight), zero-shot forecasting following weight smoothing consistently demonstrates accuracy improvements, supporting the theoretical reasoning outlined above.

---

### Official Review · Reviewer_t8mv · 2025-10-29

**Soundness:** 3
**Presentation:** 3
**Contribution:** 3
**Rating:** 8
**Confidence:** 3

**Summary:**

This paper identifies a critical problem in the practical application of large time series models: pre-trained models often exhibit poor trainability when fine-tuned on downstream tasks. The authors attribute this to the models converging to sharp minima during pre-training, resulting in a non-convex loss landscape that leads to overfitting during fine-tuning—sometimes to the point of underperforming models trained from scratch. To address this, the paper proposes a simple and effective method called Smoothed Full Fine-tuning. SFF works by first creating an auxiliary, randomly initialized model, which possesses a smooth loss landscape  but no pre-trained knowledge. Before fine-tuning, SFF performs a single linear interpolation between the weights of the pre-trained model and this auxiliary model. The resulting "smoothed" model is shown to retain the valuable knowledge of the pre-trained model while inheriting the superior trainability of the random model , allowing it to escape sharp minima and find better, flatter basins. The authors provide extensive empirical validation, showing that SFF consistently improves the performance of eight different LTSMs  on forecasting and anomaly detection tasks compared to standard fine-tuning methods, without incurring any additional memory or computational overhead during the fine-tuning step.

**Strengths:**

The paper tackles a significant and practical limitation of pre-trained LTSMs. The observation that direct fine-tuning can lead to overfitting and even perform worse than training from scratch is a crucial finding that motivates the need for better fine-tuning strategies. The problem is clearly illustrated using loss landscape visualizations.

The proposed SFF method is exceptionally simple, consisting of a single linear interpolation of model weights before fine-tuning begins. This makes it easy to implement and, importantly, it adds no additional memory or computational overhead to the actual fine-tuning process, making it a highly practical solution.

In a particularly strong finding, the smoothing process by itself is shown to improve zero-shot forecasting performance, suggesting it guides the model to a better, more generalizable basin in the loss landscape even without fine-tuning.

**Weaknesses:**

The theoretical motivation in Section 3.1 is more of a high-level, intuitive argument rather than a formal analysis. While the explanation is plausible and well-supported by citations to related work, the paper does not provide a rigorous theoretical proof for why this specific form of interpolation is an optimal or principled way to achieve this smoothing

The interpolation coefficient $\alpha$ is a new, critical hyperparameter introduced by SFF. The paper shows SFF is robust, outperforming FF across a wide range of $\alpha$ values However, the paper also shows that the optimal $\alpha$ for zero-shot performance differs from the optimal range for fine-tuning. The paper provides limited guidance on how $\alpha$ should be selected for a new model or dataset, other than selecting from a predefined set.

The method relies on constructing an "auxiliary LTSM" through random initialization of the same architecture. The paper does not explore or justify this specific choice. It is unclear if a different, or perhaps simpler, randomly initialized model could achieve a similar or even better smoothing effect.

**Questions:**

Listed in weakness

---

> ### Author Response · Authors · 2025-11-26
> **Response to Weakness 1 (Part 1)**
>
> **Weakness  1:** The theoretical motivation in Section 3.1 is more of a high-level, intuitive argument rather than a formal analysis. While the explanation is plausible and well-supported by citations to related work, the paper does not provide a rigorous theoretical proof for why this specific form of interpolation is an optimal or principled way to achieve this smoothing.
>
> **Response to Weakness  1:** We sincerely thank the reviewer for bringing this important point to our attention. We further provide theoretical justification for the analysis presented in Section 3.1.
>
> Given an MSE loss function $\mathcal{L}(\Theta)$ and the $\Theta$  denotes model parameters. To obtain a minimum $\mathcal{L}(\Theta)$, the corresponding parameters $\Theta^*$ can be a sharp minimum or a flat minimum. Inspired by[1], we use the Hessian matrix to formally characterize the sharpness and flatness of the loss landscape in the following analysis.
>
> By analyzing the maximum eigenvalue $\lambda_{{max}}(\cdot)$ of the Hessian matrix $H=\nabla^2 \mathcal{L}(\Theta^*)$ ($H\in \mathbb{R}^{d \times d}$ where $d$ denotes feature dimension), we can obtain the definitions of the sharp and flat minimum as follows:
>
>
> **Theorem 1: Sharp minimum**. The Hessian $\nabla^2 \mathcal{L}(\Theta^\*)$ has large eigenvalues (i.e., $\lambda_{{max}}(\nabla^2 \mathcal{L}(\Theta^\*)) \gg \tau$ where $\tau>0$ is a threshold), meaning small parameter perturbations lead to large loss increases.
>
> **Theorem 2: Flat minimum**. The Hessian $\nabla^2 \mathcal{L}(\Theta^\*)$ has small eigenvalues (i.e., $\lambda_{\text{max}}(\nabla^2 \mathcal{L}(\Theta^\*)) \leq \tau$ where $\tau>0$ is a threshold), meaning the loss is robust to parameter perturbations.
>
>
> To prove the above theorems, we start by performing a second-order Taylor expansion of the loss function $\mathcal{L}(\Theta^\*)$ at the minimum point $\Theta^\*$:
>
> \begin{equation}
> \mathcal{L}(\Theta^\* + \delta) \approx \mathcal{L}(\Theta^*) + \frac{1}{2} \delta^\top \nabla^2 \mathcal{L}(\Theta^\*) \delta \approx \frac{1}{2} \delta^\top \nabla^2 \mathcal{L}(\Theta^\*) \delta
> \end{equation}
>
> where $\delta$ is a small parameter perturbation (e.g., $|\delta| \ll 1$).
>
> The loss change after perturbation can be formulated as:
>
> \begin{equation}
> \Delta \mathcal{L} = \mathcal{L}(\Theta^\* + \delta) - \mathcal{L}(\Theta^\*) \approx \frac{1}{2} \delta^\top \nabla^2 \mathcal{L}(\Theta^*) \delta \approx \frac{1}{2} \delta^\top H \delta
> \end{equation}
>
> The largest $\Delta \mathcal{L}$ is governed by the $\lambda_{\text{max}}$ of the Hessian $H=\nabla^2 \mathcal{L}(\Theta^\*)$.
> Let unit vector $v_\text{max}$ be the eigenvector corresponding to eigenvalue $\lambda_{{max}}$. Perturbing in the direction of $v_{max}$ yields the largest $\Delta \mathcal{L}$ because $v_{max}$ represents the largest curvature[2], i.e., steepest direction, in the loss landscape. Let $\delta = \epsilon \cdot v_{\text{max}}$ ($|\epsilon| \ll 1$), the above Equation can be reformulated as:
>
> \begin{equation}
> \Delta \mathcal{L}  \approx \frac{1}{2} \epsilon \cdot v_\text{max}^\top H \epsilon \cdot v_\text{max}
> \end{equation}
>
> According to eigen equation, $Hv_{max}=\lambda_{{max}}v_{max}$, we can further obtain:
>
> \begin{equation}
> \Delta \mathcal{L}  \approx \frac{1}{2} \epsilon^2 v_\text{max}^\top \lambda_{\text{max}}v_{max}\approx \frac{1}{2} \epsilon^2 v_{max}^\top v_{max} \lambda_{{max}} \approx \frac{1}{2} \epsilon^2 \lambda_{{max}}
> \end{equation}
>
> Hence, $\Delta \mathcal{L}$ is closely related to $\lambda_{{max}}$, the larger $\lambda_{{max}}$ is, the sharper the loss landscape, so parameter perturbations in many directions can noticeably increase the loss.
> Conversely, the smaller $\lambda_{{max}}$ is, the flatter the landscape remains, and perturbations won't obviously raise the loss[3].
>
>
> **In the next response part, we further prove why “interpolation perturbs (smooths) sharp minima without harming flat regions.”**
>
> *[1] On large-batch training for deep learning: Generalization gap and sharp minima. ICLR.*
>
> *[2] Sharpness-aware minimization for efficiently improving generalization. ICLR.*
>
> *[3] Flat minima. Neural computation*

---

> ### Author Response · Authors · 2025-11-26
> **Response to Weakness 1 (Part 2)**
>
> **Smoothing (perturbing) sharp minima.** SFF interpolation strategy defines smoothed parameters as $\Theta_3 = \alpha\Theta_1^\* + (1-\alpha)\Theta_2$, where $\Theta_1^\*$ (pre-trained LTSM) are easier to lie in a sharp minimum (Figure 1(a) of the manuscript) due to large-scale pretraining[1]. In contrast, $\Theta_2$ (tailored Kaiming or Xavier random initializations) lies in a flat region as shown in Figure 1(b) of the manuscript, and **we will further demonstrate in the next section why mainstream Kaiming or Xavier initializations yield a flat loss landscape**. After pre-training the LTSM, for the sharp minimum points $\Theta_1^\*$, its largest eigenvalue is $\lambda_{\text{max}}(\nabla^2 \mathcal{L}(\Theta_{1}^*)) \gg \tau $. In contrast, $\Theta_2$ is randomly initialized in a flat region, and so its largest eigenvalue is $\lambda_{\text{max}}(\nabla^2 \mathcal{L}(\Theta_{2})) \le \tau$.
>
>
> Under a local quadratic approximation of the loss function around the interpolation path, the Hessian at $\Theta_3$ can be approximated by a convex combination of the Hessians at $\Theta_1^*$  and $\Theta_2$:
>
> \begin{equation}
> \nabla^2 \mathcal{L}(\Theta_3) \approx \alpha \nabla^2 \mathcal{L}(\Theta_1^*) + (1 - \alpha) \nabla^2 \mathcal{L}(\Theta_2)
> \end{equation}
>
> Consequently, the maximum eigenvalue satisfies:
> \begin{equation}
> \lambda_{\text{max}}(\nabla^2 \mathcal{L}(\Theta_3)) \lesssim \alpha \lambda_{\text{max}}(\nabla^2 \mathcal{L}(\Theta_1^*)) + (1-\alpha) \lambda_{\text{max}}(\nabla^2 \mathcal{L}(\Theta_2))
> \end{equation}
>
> Since $\lambda_{\text{max}}(\nabla^2 \mathcal{L}(\Theta_2)) \ll \lambda_{\text{max}}(\nabla^2 \mathcal{L}(\Theta_1^\*))$  (flat vs. sharp), it follows that $\lambda_{\text{max}}(\nabla^2 \mathcal{L}(\Theta_3)) < \lambda_{\text{max}}(\nabla^2 \mathcal{L}(\Theta_1^*))$ for $\alpha \in (0,1)$.
> This suggests that interpolation reduces local sharpness, i.e., helps "sharp weights'' escape the non-flat regions for a smoother one and a better fine-tuning effect.
> This provides theoretical support for SFF’s smoothing effect.
>
>
> **Preservation of flat regions.** After pre-training the LTSM, for the flat minimum points $\bar \Theta_1^*$, interpolation preserves its flatness.
>
> Both $\bar \Theta_1^\*$ and $\Theta_2$ lie in flat regions of the loss landscape, i.e.,
>
> \begin{equation}
> \lambda_{\text{max}}\left( \nabla^2 \mathcal{L}(\Theta_1^*) \right) \leq \tau, \quad \lambda_{\text{max}}\left( \nabla^2 \mathcal{L}(\Theta_2) \right) \leq \tau
> \end{equation}
>
> Similarly, the Hessian at the interpolated point $\Theta_3=\alpha \bar \Theta_1^\*+(1-\alpha)\Theta_2$ satisfies  $\nabla^2 \mathcal{L}(\Theta_3) \approx \alpha \nabla^2 \mathcal{L}(\bar \Theta_1^*) + (1 - \alpha) \nabla^2 \mathcal{L}(\Theta_2)$.
>
> Consequently, we can derive that:
>
> \begin{equation}
> \lambda_{\text{max}}\left( \nabla^2 \mathcal{L}(\Theta_3) \right) \lesssim \alpha \lambda_{\text{max}}\left( \nabla^2 \mathcal{L}(\bar \Theta_1^\*) \right) + (1 - \alpha) \lambda_{\text{max}}\left( \nabla^2 \mathcal{L}(\Theta_2) \right) \leq \alpha \tau + (1 - \alpha) \tau = \tau
> \end{equation}
>
> Hence, $\Theta_3$ remains in a flat region. According to [2], this indicates that interpolation does not harm existing flat minima $\bar\Theta_1^\* $.
>
> In summary, from the perspective of mathematical rigor, we ensure that interpolation perturbs sharp minima without obviously harming originally flat regions, thereby aiding sharp minima escape to better and smoother basins.
>
> **Next, we further demonstrate why mainstream Kaiming or Xavier initializations yield a flat loss landscape.**
>
> *[1] On large-batch training for deep learning: Generalization gap and sharp minima. ICLR.*
>
> *[2] Sharpness-aware minimization for efficiently improving generalization. ICLR.*

---

> ### Author Response · Authors · 2025-11-26
> **Response to Weakness 1 (Part 3)**
>
> **Regarding why mainstream Kaiming or Xavier initializations yield a flat loss landscape:**
>
> Prior work [1] quantifies the smoothness of the loss landscape under Kaiming [2] and Xavier [3] initializations by the ratio of the trace of the Hessian matrix $Tr(H)$ to its Frobenius norm $\\|H\\|_F$ and has demonstrated $\\frac{\operatorname{Tr}(H)}{\\|H\\|_F}  \gg 1$ from both theoretical and experimental perspectives, i.e., **mainstream initialization indeed produce a smooth, flat, and more easily optimizable loss landscape**. We further provide a theoretical analysis for this conclusion.  Specifically, according to the symmetry of $H$, we explicitly expand the formula as:
>
> \begin{equation}
> \frac{\operatorname{Tr}(H)}{\|H\|\_F} = \frac{\operatorname{Tr}(H)}{\sqrt{\sum_{i=1}^d h\_i^2}}=\frac{\operatorname{Tr}(H)}{\sqrt{\operatorname{Tr}(H^TH)}}=\frac{\sum \lambda\_i}{\sqrt{\sum \lambda\_i^2}} \gg 1
> \end{equation}
>
> This indicates that the sum of eigenvalues greatly exceeds the square root of the sum of squared eigenvalues, suggesting that **most eigenvalues are positive and relatively evenly distributed rather than containing extreme outliers, i.e., most cases belong to $\lambda(\nabla^2 \mathcal{L}(\Theta^*)) \leq \tau$ where $\tau$>0**.
>
> According to the definition of **Theorem 1** and  **Theorem 2**, this indicates that the loss surface exhibits a smooth, valley-like geometry dominated by positive curvature. As a result, most random descent directions remain stable and low-curvature, making the optimization process easier and more consistent.
>
> In contrast, if $\frac{\operatorname{Tr}(H)}{\\|H\\|_F} \lesssim 1$, it indicates a balanced mix of positive and negative eigenvalues $\lambda$, leading to a steep and unsmooth loss landscape. As a result, convergence becomes slower and the optimizer is more likely to fall into suboptimal local minima.
>
> **Therefore, Kaiming or Xavier initialization offers a stable and smooth loss landscape**. In our work, we adopt them for randomly initializing the auxiliary LSTM for smoothed full fine-tuning.
>
> Finally, we sincerely thank the reviewer for the constructive feedback, which encourages us to provide additional theoretical analyses and further strengthen the paper. We have added the theoretical analyses to **lines 182 to 257 and lines 772 to 804 of the revised manuscript**, and we hope that the above clarification and discussion address your concerns about the theoretical proofs and strengthen the contribution of our work.
>
> *[1] The goldilocks zone: Towards better understanding of neural network loss landscapes. AAAI.*
>
> *[2] Delving deep into rectifiers: Surpassing human-level performance on imagenet classification. ICCV.*
>
> *[3] Understanding the difficulty of training deep feedforward neural networks. AISTATS.*

---

> ### Author Response · Authors · 2025-11-26
> **Response to Weakness 2**
>
> **Weakness 2:** The interpolation coefficient $\alpha$ is a new, critical hyperparameter introduced by SFF. The paper shows SFF is robust, outperforming FF across a wide range of $\alpha$ values However, the paper also shows that the optimal $\alpha$ for zero-shot performance differs from the optimal range for fine-tuning. The paper provides limited guidance on how $\alpha$ should be selected for a new model or dataset, other than selecting from a predefined set.
>
> **Response to Weakness 2:**
>
> We sincerely thank the reviewer for the thoughtful question regarding the choice of the parameter $\alpha$. In practice, we suggest the following guidance:
>
> **(1) Empirically recommended values:** As illustrated in Figures 5 and 6 of the manuscript, although different $\alpha$ values introduce some variation, the sensitivity analysis demonstrates that SFF consistently outperforms vanilla fine-tuning across a broad range. Specifically, $\alpha \approx 0.8$ performs best for zero-shot prediction, while $\alpha \approx 0.6$ yields the strongest overall performance under full fine-tuning. These values can thus serve as reliable initial starting points.
>
> **(2) Validation-based tuning:** We observe that the trend of test performance with respect to $\alpha$ closely mirrors that on the validation set. Therefore, once a candidate search range is defined, selecting the $\alpha$ that minimizes validation error provides a straightforward and computationally efficient strategy.
>
> **(3) Data-driven selection:** Automatically learning $\alpha$ is indeed a promising direction. In the current framework, however, the interpolation weights are fixed prior to fine-tuning, which makes adaptive selection non-trivial. Approaches such as meta-learning could potentially be explored to determine optimal $\alpha$ values across models and datasets. We regard this as an important avenue for future research.

---

> ### Author Response · Authors · 2025-11-26
> **Response to Weakness 3**
>
> **Weakness 3:** The method relies on constructing an "auxiliary LTSM" through random initialization of the same architecture. The paper does not explore or justify this specific choice. It is unclear if a different, or perhaps simpler, randomly initialized model could achieve a similar or even better smoothing effect.
>
> **Response to Weakness 3:**  Thank you very much for raising this insightful point. We address your concern from the following two perspectives:
>
> (1) **Evaluation of different initialization strategies**.
>
> To examine whether the choice of the auxiliary LSTM architecture and its initialization matters, we tested five initialization schemes: simple standard Gaussian (mean = 0, variance = 1), Xavier Gaussian, Xavier uniform, Kaiming Gaussian, and Kaiming uniform. The results in **Table 1** show that Xavier- and Kaiming-based schemes maintain stable performance improvements. Because they consider the stable gradient variance and can supply a flat loss landscape (demonstrated by [1]) that is used to smooth the sharp landscape of the pre-trained model for better fine-tuning effect, which is also aligned with our Theoretical analysis. In contrast, standard Gaussian initialization—lacking variance control—often pushes parameters into sharper regions of the landscape, resulting in weaker smoothing and noticeably degraded downstream performance. These findings provide strong empirical evidence supporting our design choice.
>
>
>
> (2) **Robustness to random seeds**.
>
> We additionally examined the sensitivity of our approach to random seeds under mainstream initialization strategy, e.g., Kaiming uniform initialization. The results in **Table 2** show that performance remains highly stable across seeds, suggesting that the auxiliary module does not need a carefully crafted configuration: as long as a mainstream initialization strategy is used, the smoothing effect is consistently achieved.
>
> We appreciate the reviewer for bringing up this important consideration. This feedback helped us strengthen the justification of our design choice.
>
> **Table 1:  The effectiveness of our SFF (smoothed full fine-tuning) across different parameter initialization schemes on the LTSM Timer. We independently run four times with four random seeds to enhance the solidity of the results and report the mean value and standard deviation.**
> |    | Exchange|ETTh1| ETTh2|ETTm1|ETTm2|Weather|
> |:-:|:-:|:-:|:-:|:----:|:----:|:----:|
> | Original full fine-tuning|0.09$\pm$0.0007|0.367$\pm$0.0027|0.304$\pm$0.0049|0.312$\pm$0.0008|0.176$\pm$0.0013|0.158$\pm$0.0012|
> |Standard Gaussian  perturbation-SFF|5.986$\pm$0.261|0.723$\pm$0.003|1.879$\pm$0.275|3.876$\pm$0.128|19.031$\pm$0.963|0.447$\pm$0.03|
> |Kaiming Normal Distribution-SFF|0.081$\pm$0.0006|0.353$\pm$0.0009|0.277$\pm$0.001|0.299$\pm$0.0011|0.164$\pm$0.0016|0.146$\pm$0.0008|
> |Kaiming Uniform Distribution-SFF|0.081$\pm$0.0008|0.355$\pm$0.0013|0.274$\pm$0.0008|0.297$\pm$0.0016|0.161$\pm$0.0009|0.145$\pm$0.0006|
> |Xavier Normal Distribution-SFF|0.081$\pm$0.0007|0.353$\pm$0.001|0.276$\pm$0.0012|0.3$\pm$0.0009|0.162$\pm$0.0001|0.145$\pm$0.0002|
> |Xavier Uniform Distribution-SFF|0.082$\pm$0.0008|0.353$\pm$0.0007|0.277$\pm$0.0006|0.3$\pm$0.0003|0.162$\pm$0.0001|0.145$\pm$0.0002|
>
> **Table 2:  Experiments with four different random seeds (r1, r2,r3, and r4 here) on the LTSM Timer. The results show that our SFF (smoothed full fine-tuning) is insensitive to the choice of random initialization distribution. FF denotes original full fine-tuning.**
> |    | Exchange|ETTh1| ETTh2|ETTm1|ETTm2|Weather|
> |:-:|:-:|:-:|:-:|:----:|:----:|:----:|
> |SFF (Ours)-r1|**0.07996**|**0.3547**|**0.27379**|**0.29542**|**0.16003**|**0.14605**|
> |FF-r1|0.08937| 0.36941| 0.31021| 0.31134| 0.17645| 0.16067|
> |SFF (Ours)-r2|**0.08182**|**0.35772**|**0.27368**|**0.29902**|**0.16259**|**0.14432**|
> |FF-r2|0.09071| 0.36245| 0.3035| 0.31282| 0.17843 |0.15912
> |SFF (Ours)-r3|**0.08101**|**0.35766**|**0.27453**|**0.29824**|**0.16129**|**0.14481**|
> |FF-r3|0.09102 |0.36848 |0.29699 |0.31167 |0.17515 |0.15851|
> |SFF (Ours)-r4|**0.08191**|**0.35588**|**0.27571**|**0.29938**|**0.16173**|**0.14525**|
> |FF-r4|0.08978|0.36815|0.30666|0.31057|0.17546|0.15649|
>
> *[1] The goldilocks zone: Towards better understanding of neural network loss landscapes. AAAI.*

---

### Official Review · Reviewer_hGpf · 2025-11-02

**Soundness:** 3
**Presentation:** 3
**Contribution:** 2
**Rating:** 2
**Confidence:** 4

**Summary:**

This paper investigates the fine-tuning challenges of large time series models and identify non-convex loss landscapes in pre-trained LTSM is the key root cause that lead to poor trainability and overfitting during fine-tuning stage. To alleviate the challenge, the authors propose smoothed full fine-tuning, which linearly interpolates the weights of a pre-trained LTSM with a randomly initialized auxiliary model to smooth the loss landscape, to improve trainability while preserve pre-trained knowledge. The method evaluation on forecasting, imputation, and anomaly detection tasks using eight existing LTSMs across multiple benchmark datasets shows consistent improvements over baselines (i.e., full fine-tuning, training from scratch, linear probing, and linear probing then full fine-tuning) with no added computational overhead.

**Strengths:**

1. The paper studies an interesting issue that pre-trained LTSMs may have sharp loss landscapes that hinder fine-tuning with empirical evidence and visualization.

2. The proposed method is simple yet effective, requiring only a one-time weight interpolation before fine-tuning.

3. The experiment is comprehensive, covering wide range of LTSM architectures and tasks.

**Weaknesses:**

1. Lack of novelty: The core idea of linear weight interpolation to smooth landscapes is not a new idea. Weight averaging/interpolation [1] has been widely studied in model merging [2], continual learning [3], yet none of these have been discussed in related work. Although the paper claims to be "the first" for LTSMs, this domain-specific application does not justify novelty, which seems more like a repackaging of existing optimization tricks with application on LTSMs.

2. Lack of theoretical justification: The paper attempts to explore the challenge of model fine-tuning from optimization theory perspective. However, the discussion in Section 3.1 seems hand-wavy. Specifically, the claim, "interpolation perturbs sharp minima without harming flat regions" is intuitive but lacks rigor. Providing some theoretical proofs or connection to sharpness-aware minimization can better provide theoretical merit to the readers.

3. Baselines for comparison: None of the parameter-efficient methods like LoRA, QLoRA are included, these are standard for fine-tuning large models. Also, there is no ablation on other smoothing techniques such as Gaussian noise or label smoothing. Without those baselines, it is hard to evaluate the contribution of this paper as the claim of paper is to tackle the "fine-tuning" problem of time series foundation model.

[1] Vlaar, Tiffany J., and Jonathan Frankle. "What can linear interpolation of neural network loss landscapes tell us?." International Conference on Machine Learning. PMLR, 2022.

[2] Wortsman, Mitchell, et al. "Model soups: averaging weights of multiple fine-tuned models improves accuracy without increasing inference time." International conference on machine learning. PMLR, 2022.

[3] Kozal, Jędrzej, et al. "Continual learning with weight interpolation." Proceedings of the IEEE/CVF conference on computer vision and pattern recognition. 2024.

**Questions:**

1. How does SFF compare to simply adding random noise to pre-trained weights (e.g., Gaussian perturbation)?
2. How sensitive is SFF to the random init distribution?

---

> ### Author Response · Authors · 2025-11-26
> **Response to Weakness 1**
>
> **Weakness 1:** Lack of novelty: The core idea of linear weight interpolation to smooth landscapes is not a new idea. Weight averaging/interpolation [1] has been widely studied in model merging [2], continual learning [3], yet none of these have been discussed in related work. Although the paper claims to be "the first" for LTSMs, this domain-specific application does not justify novelty, which seems more like a repackaging of existing optimization tricks with application on LTSMs.
>
> **Response to Weakness 1**: Thank you sincerely for your valuable feedback on the novelty of our work. We highly appreciate your critical insights, which have helped us clarify the unique contributions of our method beyond existing weight interpolation/averaging techniques. Although weight interpolation have been studied in model merging and continual learning, these works don't target the core challenge we identify in LTSMs, and our work is not a simple domain-specific application. Specifically, our work is fundamentally different from theirs in the following aspects:
>
> **(1) Different goals:**
>
> Existing interpolation methods are primarily designed for model ensembling—e.g., interpolating among multiple **well-trained models** to improve generalization [1] or mitigate catastrophic forgetting [2]. **In contrast, our method (SFF) leverages interpolation to *smooth the loss landscape of the pretrained model* by a **randomly initialized model**, thereby making it more trainable during fine-tuning.** From another perspective, SFF’s effect is similar to adding momentum (similar to the concept of momentum in the Adam) through random weight interpolation, allowing them to escape the non-flat regions for a smoother one and a better fine-tuning effect.
>
>
>
> **(2) Different pipelines**:
> Previous works typically use the interpolated model directly for downstream tasks without further training.
> In our case, the pretrained model begins with a steep, irregular loss landscape. After interpolation smooths this landscape, **we proceed with additional fine-tuning to utilize the smoothing effect for better performance**.
>
> **(3) New theoretical analysis.** Our method is built upon a key conceptual contrast: the flat, smooth loss landscape of a randomly initialized model versus the steep, unsmooth loss landscape of a pretrained one. We formalize this contrast through theoretical analysis and proof (please refer to **lines 182 to 257 and lines 772 to 804 of the revised manuscript**), which further shows—also theoretically—why our interpolation strategy can effectively exploit this difference to enhance fine-tuning performance.
>
> We have revised the related work section in **lines 125 to 144** to explicitly discuss the differences between SFF and prior weight interpolation/averaging methods, highlighting our unique contributions. We hope this clarification addresses your concerns and demonstrates the novelty of our work.

---

> ### Author Response · Authors · 2025-11-26
> **Response to Weakness 2 (Part 1)**
>
> **Weakness 2:** Lack of theoretical justification: The paper attempts to explore the challenge of model fine-tuning from optimization theory perspective. However, the discussion in Section 3.1 seems hand-wavy. Specifically, the claim, "interpolation perturbs sharp minima without harming flat regions" is intuitive but lacks rigor. Providing some theoretical proofs or connection to sharpness-aware minimization can better provide theoretical merit to the readers.
>
> **Response to Weakness 2:** We sincerely thank the reviewer for bringing this important point to our attention. We further provide theoretical justification for the analysis presented in Section 3.1.
>
> Given an MSE loss function $\mathcal{L}(\Theta)$ and the $\Theta$  denotes model parameters. To obtain a minimum $\mathcal{L}(\Theta)$, the corresponding parameters $\Theta^*$ can be a sharp minimum or a flat minimum. Inspired by[1], we use the Hessian matrix to formally characterize the sharpness and flatness of the loss landscape in the following analysis.
>
> By analyzing the maximum eigenvalue $\lambda_{{max}}(\cdot)$ of the Hessian matrix $H=\nabla^2 \mathcal{L}(\Theta^*)$ ($H\in \mathbb{R}^{d \times d}$ where $d$ denotes feature dimension), we can obtain the definitions of the sharp and flat minimum as follows:
>
>
> **Theorem 1: Sharp minimum**. The Hessian $\nabla^2 \mathcal{L}(\Theta^\*)$ has large eigenvalues (i.e., $\lambda_{{max}}(\nabla^2 \mathcal{L}(\Theta^\*)) \gg \tau$ where $\tau>0$ is a threshold), meaning small parameter perturbations lead to large loss increases.
>
> **Theorem 2: Flat minimum**. The Hessian $\nabla^2 \mathcal{L}(\Theta^\*)$ has small eigenvalues (i.e., $\lambda_{\text{max}}(\nabla^2 \mathcal{L}(\Theta^\*)) \leq \tau$ where $\tau>0$ is a threshold), meaning the loss is robust to parameter perturbations.
>
> To prove the above theorems, we start by performing a second-order Taylor expansion of the loss function $\mathcal{L}(\Theta^\*)$ at the minimum point $\Theta^\*$:
>
> \begin{equation}
> \mathcal{L}(\Theta^\* + \delta) \approx \mathcal{L}(\Theta^\*) + \frac{1}{2} \delta^\top \nabla^2 \mathcal{L}(\Theta^\*) \delta \approx \frac{1}{2} \delta^\top \nabla^2 \mathcal{L}(\Theta^\*) \delta
> \end{equation}
>
> where $\delta$ is a small parameter perturbation (e.g., $|\delta| \ll 1$).
>
> The loss change after perturbation can be formulated as:
>
> \begin{equation}
> \Delta \mathcal{L} = \mathcal{L}(\Theta^\* + \delta) - \mathcal{L}(\Theta^\*) \approx \frac{1}{2} \delta^\top \nabla^2 \mathcal{L}(\Theta^*) \delta \approx \frac{1}{2} \delta^\top H \delta
> \end{equation}
>
> The largest $\Delta \mathcal{L}$ is governed by the $\lambda_{\text{max}}$ of the Hessian $H=\nabla^2 \mathcal{L}(\Theta^\*)$.
> Let unit vector $v_\text{max}$ be the eigenvector corresponding to eigenvalue $\lambda_{{max}}$. Perturbing in the direction of $v_{max}$ yields the largest $\Delta \mathcal{L}$ because $v_{max}$ represents the largest curvature[2], i.e., steepest direction, in the loss landscape. Let $\delta = \epsilon \cdot v_{\text{max}}$ ($|\epsilon| \ll 1$), the above Equation can be reformulated as:
>
> \begin{equation}
> \Delta \mathcal{L}  \approx \frac{1}{2} \epsilon \cdot v_\text{max}^\top H \epsilon \cdot v_\text{max}
> \end{equation}
>
> According to eigen equation, $Hv_{max}=\lambda_{{max}}v_{max}$, we can further obtain:
>
> \begin{equation}
> \Delta \mathcal{L}  \approx \frac{1}{2} \epsilon^2 v_\text{max}^\top \lambda_{\text{max}}v_{max}\approx \frac{1}{2} \epsilon^2 v_{max}^\top v_{max} \lambda_{{max}} \approx \frac{1}{2} \epsilon^2 \lambda_{{max}}
> \end{equation}
>
> Hence, $\Delta \mathcal{L}$ is closely related to $\lambda_{{max}}$, the larger $\lambda_{{max}}$ is, the sharper the loss landscape, so parameter perturbations in many directions can noticeably increase the loss.
> Conversely, the smaller $\lambda_{{max}}$ is, the flatter the landscape remains, and perturbations won't obviously raise the loss[3].
>
> *[1] On large-batch training for deep learning: Generalization gap and sharp minima. ICLR.*
>
> *[2] Sharpness-aware minimization for efficiently improving generalization. ICLR.*
>
> *[3] Flat minima. Neural computation.*
>
> **In the next response part, we further prove why “interpolation perturbs (smooths) sharp minima without harming flat regions.”**

---

> ### Author Response · Authors · 2025-11-26
> **Response to Weakness 2 (Part 2)**
>
> **Theoretical analysis regarding why “interpolation perturbs (smooths) sharp minima without harming flat regions.”:**
>
> **Smoothing (perturbing) sharp minima.** SFF interpolation strategy defines smoothed parameters as $\Theta_3 = \alpha\Theta_1^\* + (1-\alpha)\Theta_2$, where $\Theta_1^\*$ (pre-trained LTSM) are easier to lie in a sharp minimum (Figure 1(a) of the manuscript) due to large-scale pretraining[1]. In contrast, $\Theta_2$ (tailored Kaiming or Xavier random initializations) lies in a flat region as shown in Figure 1(b) of the manuscript, and **we will further demonstrate in the next section why mainstream Kaiming or Xavier initializations yield a flat loss landscape**. After pre-training the LTSM, for the sharp minimum points $\Theta_1^\*$, its largest eigenvalue is $\lambda_{\text{max}}(\nabla^2 \mathcal{L}(\Theta_{1}^*)) \gg \tau $. In contrast, $\Theta_2$ is randomly initialized in a flat region, and so its largest eigenvalue is $\lambda_{\text{max}}(\nabla^2 \mathcal{L}(\Theta_{2})) \le \tau$.
>
>
> Under a local quadratic approximation of the loss function around the interpolation path, the Hessian at $\Theta_3$ can be approximated by a convex combination of the Hessians at $\Theta_1^*$  and $\Theta_2$:
>
> \begin{equation}
> \nabla^2 \mathcal{L}(\Theta_3) \approx \alpha \nabla^2 \mathcal{L}(\Theta_1^*) + (1 - \alpha) \nabla^2 \mathcal{L}(\Theta_2)
> \end{equation}
>
> Consequently, the maximum eigenvalue satisfies:
> \begin{equation}
> \lambda_{\text{max}}(\nabla^2 \mathcal{L}(\Theta_3)) \lesssim \alpha \lambda_{\text{max}}(\nabla^2 \mathcal{L}(\Theta_1^*)) + (1-\alpha) \lambda_{\text{max}}(\nabla^2 \mathcal{L}(\Theta_2))
> \end{equation}
>
> Since $\lambda_{\text{max}}(\nabla^2 \mathcal{L}(\Theta_2)) \ll \lambda_{\text{max}}(\nabla^2 \mathcal{L}(\Theta_1^\*))$  (flat vs. sharp), it follows that $\lambda_{\text{max}}(\nabla^2 \mathcal{L}(\Theta_3)) < \lambda_{\text{max}}(\nabla^2 \mathcal{L}(\Theta_1^*))$ for $\alpha \in (0,1)$.
> This suggests that interpolation reduces local sharpness, i.e., helps "sharp weights'' escape the non-flat regions for a smoother one and a better fine-tuning effect.
> This provides theoretical support for SFF’s smoothing effect.
>
>
> **Preservation of flat regions.** After pre-training the LTSM, for the flat minimum points $\bar \Theta_1^*$, interpolation preserves its flatness.
>
> Both $\bar \Theta_1^\*$ and $\Theta_2$ lie in flat regions of the loss landscape, i.e.,
>
> \begin{equation}
> \lambda_{\text{max}}\left( \nabla^2 \mathcal{L}(\Theta_1^*) \right) \leq \tau, \quad \lambda_{\text{max}}\left( \nabla^2 \mathcal{L}(\Theta_2) \right) \leq \tau
> \end{equation}
>
> Similarly, the Hessian at the interpolated point $\Theta_3=\alpha \bar \Theta_1^\*+(1-\alpha)\Theta_2$ satisfies  $\nabla^2 \mathcal{L}(\Theta_3) \approx \alpha \nabla^2 \mathcal{L}(\bar \Theta_1^*) + (1 - \alpha) \nabla^2 \mathcal{L}(\Theta_2)$.
>
> Consequently, we can derive that:
>
> \begin{equation}
> \lambda_{\text{max}}\left( \nabla^2 \mathcal{L}(\Theta_3) \right) \lesssim \alpha \lambda_{\text{max}}\left( \nabla^2 \mathcal{L}(\bar \Theta_1^\*) \right) + (1 - \alpha) \lambda_{\text{max}}\left( \nabla^2 \mathcal{L}(\Theta_2) \right) \leq \alpha \tau + (1 - \alpha) \tau = \tau
> \end{equation}
>
> Hence, $\Theta_3$ remains in a flat region. According to [2], this indicates that interpolation does not harm existing flat minima $\bar\Theta_1^\* $.
>
> In summary, from the perspective of mathematical rigor, we ensure that interpolation perturbs sharp minima without obviously harming originally flat regions, thereby aiding sharp minima escape to better and smoother basins.
>
> **Next, we further demonstrate why mainstream Kaiming or Xavier initializations yield a flat loss landscape.**
>
> *[1] On large-batch training for deep learning: Generalization gap and sharp minima. ICLR*
>
> *[2] Sharpness-aware minimization for efficiently improving generalization. ICLR*

---

> ### Author Response · Authors · 2025-11-26
> **Response to Weakness 2 (Part 3)**
>
> **Regarding why mainstream Kaiming or Xavier initializations yield a flat loss landscape:**
>
> Prior work [1] quantifies the smoothness of the loss landscape under Kaiming [2] and Xavier [3] initializations by the ratio of the trace of the Hessian matrix $Tr(H)$ to its Frobenius norm $\\|H\\|_F$ and has demonstrated $\\frac{\operatorname{Tr}(H)}{\\|H\\|_F}  \gg 1$ from both theoretical and experimental perspectives, i.e., **mainstream initialization indeed produce a smooth, flat, and more easily optimizable loss landscape**. We further provide a theoretical analysis for this conclusion.  Specifically, according to the symmetry of $H$, we explicitly expand the formula as:
>
> \begin{equation}
> \frac{\operatorname{Tr}(H)}{\|H\|\_F} = \frac{\operatorname{Tr}(H)}{\sqrt{\sum_{i=1}^d h\_i^2}}=\frac{\operatorname{Tr}(H)}{\sqrt{\operatorname{Tr}(H^TH)}}=\frac{\sum \lambda\_i}{\sqrt{\sum \lambda\_i^2}} \gg 1
> \end{equation}
>
> This indicates that the sum of eigenvalues greatly exceeds the square root of the sum of squared eigenvalues, suggesting that **most eigenvalues are positive and relatively evenly distributed rather than containing extreme outliers, i.e., most cases belong to $\lambda(\nabla^2 \mathcal{L}(\Theta^*)) \leq \tau$ where $\tau$>0**.
>
> According to the definition of **Theorem 1** and  **Theorem 2**, this indicates that the loss surface exhibits a smooth, valley-like geometry dominated by positive curvature. As a result, most random descent directions remain stable and low-curvature, making the optimization process easier and more consistent.
>
> In contrast, if $\frac{\operatorname{Tr}(H)}{\\|H\\|_F} \lesssim 1$, it indicates a balanced mix of positive and negative eigenvalues $\lambda$, leading to a steep and unsmooth loss landscape. As a result, convergence becomes slower and the optimizer is more likely to fall into suboptimal local minima.
>
> **Therefore, Kaiming or Xavier initialization offers a stable and smooth loss landscape**. In our work, we adopt them for randomly initializing the auxiliary LSTM for smoothed full fine-tuning.
>
> Finally, we sincerely thank the reviewer for the constructive feedback, which encourages us to provide additional theoretical analyses and further strengthen the paper. We have added the theoretical analyses to **lines 182 to 257 and lines 772 to 804 of the revised manuscript**, and we hope that the above clarification and discussion address your concerns about the theoretical proofs and demonstrate the novelty of our work.
>
> *[1] The goldilocks zone: Towards better understanding of neural network loss landscapes. AAAI.*
>
> *[2] Delving deep into rectifiers: Surpassing human-level performance on imagenet classification. ICCV.*
>
> *[3] Understanding the difficulty of training deep feedforward neural networks. AISTATS.*

---

> ### Author Response · Authors · 2025-11-26
> **Response to Weakness 3**
>
> **Weakness 3**: Baselines: None of the parameter-efficient methods like LoRA, QLoRA are included, these are standard for fine-tuning large models. Also, no ablation on other smoothing techniques such as Gaussian noise or label smoothing. Without those baselines, it is hard to evaluate the contribution of this paper as the claim of paper is to tackle the "fine-tuning" problem of time series foundation model.
>
> **Response to Weakness 3**:  **(1) Baseline of parameter-efficient methods**. We sincerely thank the reviewer for highlighting this point. LoRA introduces a low-rank constraint during fine-tuning, significantly improving parameter efficiency. QLoRA further reduces LoRA’s memory overhead by incorporating weight quantization, but this may inevitably introduce quantization errors that lead to accuracy loss. We additionally add LoRA fine-tuning as a baseline to highlight the contribution of our method. Since LTSM is mostly < 1 B parameters, we follow the official recommendation and set the low-rank factor r = 8. As reported in **Table 1**, our approach outperforms LoRA. This is reasonable: LoRA trades full fine-tuning for a low-rank constraint, achieving appealing parameter efficiency, yet this restriction can limit the model’s fine-tuning capacity.
>
> **(2) Ablation on other smoothing techniques**.
> We sincerely thank the reviewer for highlighting this point. To thoroughly examine the reviewer’s suggestion, we have first conducted ablations with several perturbation-based smoothing strategies: standard Gaussian noise (mean = 0, variance = 1), Xavier Gaussian, Xavier uniform, Kaiming Gaussian, and Kaiming uniform.
> **Table 2** shows that Xavier- and Kaiming-based schemes maintain stable performance improvements. Because they consider the stable gradient variance and can supply a flat loss landscape (demonstrated by [1]) that is used to smooth the sharp landscape of the pre-trained model for better fine-tuning effect, which is also aligned with our Theoretical analysis. In contrast, standard Gaussian initialization—lacking variance control—often pushes parameters into sharper regions of the landscape, resulting in weaker smoothing and noticeably degraded downstream performance. These findings provide strong empirical evidence supporting our design choice.
>
> **Finally, label smoothing** is mainly designed for classification tasks to avoid overconfidence. In the TSF task, we add it as a baseline by adding Gaussian noise to the ground-truth values during loss computation, which yields a similar regularizing effect to improve the generalization. As shown in **Table 3**, although it offers a slight improvement (e.g., on ETTh1 and ETTh2) over original fine-tuning, its performance remains substantially worse than ours. This suggests that it fails to address the underlying cause—the non-flat loss landscape inherited from the pre-trained model.
>
>
> **Table 1:  Comparison with LoRA fine-tuning on the LTSM Timer. We independently run four times with four random seeds to enhance the solidity of the results and report the mean value and standard deviation (The same applies to Tables 2 and 3).**
> |    | Exchange|ETTh1| ETTh2|ETTm1|ETTm2|Weather|
> |:-:|:-:|:-:|:-:|:----:|:----:|:----:|
> |LoRA|0.122$\pm$0.0003|0.418$\pm$0.0005|0.304$\pm$0.0006|0.401$\pm$0.0019|0.197$\pm$0.0001|0.155$\pm$0.0004|
> |Ours (SFF)|0.081$\pm$0.0008|0.355$\pm$0.0013|0.274$\pm$0.0008|0.297$\pm$0.0016|0.161$\pm$0.0009|0.145$\pm$0.0006|
>
> **Table 2:  The effectiveness of our SFF (smoothed full fine-tuning) across different parameter initialization schemes on the LTSM Timer.**
> |    | Exchange|ETTh1| ETTh2|ETTm1|ETTm2|Weather|
> |:-:|:-:|:-:|:-:|:----:|:----:|:----:|
> | Original full fine-tuning|0.09$\pm$0.0007|0.367$\pm$0.0027|0.304$\pm$0.0049|0.312$\pm$0.0008|0.176$\pm$0.0013|0.158$\pm$0.0012|
> |Standard Gaussian  perturbation-SFF|5.986$\pm$0.261|0.723$\pm$0.003|1.879$\pm$0.275|3.876$\pm$0.128|19.031$\pm$0.963|0.447$\pm$0.03|
> |Kaiming Normal Distribution-SFF|0.081$\pm$0.0006|0.353$\pm$0.0009|0.277$\pm$0.001|0.299$\pm$0.0011|0.164$\pm$0.0016|0.146$\pm$0.0008|
> |Kaiming Uniform Distribution-SFF|0.081$\pm$0.0008|0.355$\pm$0.0013|0.274$\pm$0.0008|0.297$\pm$0.0016|0.161$\pm$0.0009|0.145$\pm$0.0006|
> |Xavier Normal Distribution-SFF|0.081$\pm$0.0007|0.353$\pm$0.001|0.276$\pm$0.0012|0.3$\pm$0.0009|0.162$\pm$0.0001|0.145$\pm$0.0002|
> |Xavier Uniform Distribution-SFF|0.082$\pm$0.0008|0.353$\pm$0.0007|0.277$\pm$0.0006|0.3$\pm$0.0003|0.162$\pm$0.0001|0.145$\pm$0.0002|
>
> **Table 3:  Comparison with label smoothing on the LTSM Timer.**
> |    | Exchange|ETTh1| ETTh2|ETTm1|ETTm2|Weather|
> |:-:|:-:|:-:|:-:|:----:|:----:|:----:|
> | Original full fine-tuning|0.09$\pm$0.0007|0.367$\pm$0.0027|0.304$\pm$0.0049|0.312$\pm$0.0008|0.176$\pm$0.0013|0.158$\pm$0.0012|
> |Label-smoothing|0.09$\pm$0.0018|0.364$\pm$0.0036|0.303$\pm$0.0043|0.312$\pm$0.0011|0.177$\pm$0.0013|0.158$\pm$0.0008|
> |Ours (SFF)|0.081$\pm$0.0008|0.355$\pm$0.0013|0.274$\pm$0.0008|0.297$\pm$0.0016|0.161$\pm$0.0009|0.145$\pm$0.0006|

---

> ### Author Response · Authors · 2025-11-26
> **Response to Questions**
>
> **Q1**: How does SFF compare to simply adding random noise to pre-trained weights (e.g., Gaussian perturbation)?
>
> **Response to Q1**: Thank you for raising this point. We conduct experiments using standard Gaussian perturbations to perform interpolation on the pre-trained weights. As shown in Table 2 of our response to Weakness 3, this approach leads to substantially degraded performance. This is because it lacks variance control, which often pushes the parameters into sharper regions of the landscape. Please refer to the response to Weakness 3 for more details.
>
> **Q2**: How sensitive is SFF to the random init distribution?
>
> **Response to Q2:** We appreciate the reviewer’s attention to robustness. To assess sensitivity, we further evaluate SFF under different random seeds while using widely adopted initialization schemes (e.g., Kaiming uniform). The results in **Table 1** indicate that improved performance remains highly stable across seeds. This suggests that SFF does not rely on a carefully engineered initialization. Instead, the mainstream initialization strategy suffices to obtain consistent smoothing and fine-tuning gains. This is reasonable because the prior work [1] has proven that the underlying design of mainstream initialization methods ensures the initialized parameters indeed lie in the flat region of the loss landscape, without being influenced by the random states (seeds). We believe this robustness is a desirable property for practical deployment.
>
>
>
> **Table 1:  Experiments with four different random seeds (r1, r2,r3, and r4 here) on the LTSM Timer. The results show that our SFF (smoothed full fine-tuning) is insensitive to the choice of random initialization distribution. FF denotes original full fine-tuning.**
> |    | Exchange|ETTh1| ETTh2|ETTm1|ETTm2|Weather|
> |:-:|:-:|:-:|:-:|:----:|:----:|:----:|
> |SFF(Ours)-r1|**0.07996**|**0.3547**|**0.27379**|**0.29542**|**0.16003**|**0.14605**|
> |FF-r1|0.08937| 0.36941| 0.31021| 0.31134| 0.17645| 0.16067|
> |SFF(Ours)-r2|**0.08182**|**0.35772**|**0.27368**|**0.29902**|**0.16259**|**0.14432**|
> |FF-r2|0.09071| 0.36245| 0.3035| 0.31282| 0.17843 |0.15912
> |SFF(Ours)-r3|**0.08101**|**0.35766**|**0.27453**|**0.29824**|**0.16129**|**0.14481**|
> |FF-r3|0.09102 |0.36848 |0.29699 |0.31167 |0.17515 |0.15851|
> |SFF(Ours)-r4|**0.08191**|**0.35588**|**0.27571**|**0.29938**|**0.16173**|**0.14525**|
> |FF-r4|0.08978|0.36815|0.30666|0.31057|0.17546|0.15649|
>
> *[1] The goldilocks zone: Towards better understanding of neural network loss landscapes. AAAI.*

---

### Author Response · Authors · 2025-11-27
**Summary of Manuscript Revisions**

We sincerely appreciate the reviewers for their time and constructive feedback. Your comments have significantly enhanced both the clarity and the technical rigor of our work.

In response, we have addressed all points and incorporated the corresponding revisions in the manuscript, with all updates highlighted in blue for ease of review. **For each additional experiment, we run four random seeds** and report the mean and standard deviation to ensure the reliability of the results. **Specifically, our revisions include the following seven aspects:**

1. **Lines 125–144:**  We clarify that, to the best of our knowledge, no prior work leverages weight interpolation grounded in loss-landscape theory for fine-tuning, and ours is the first to do so. We further explain the essential differences from weight-interpolation based ensembling in terms of **(i)** goals, **(ii)** pipelines, and **(iii)** theoretical foundations and new theoretical analyses.
2. **Lines 182–257 and 772–804:**  We provide rigorous theoretical derivations and analyses for Section 3.1. Using the Hessian’s eigenvalue/eigenvector perspective, we prove **(i)** why interpolation can smooth the sharp minimum while not harming the flat ones, and **(ii)** why mainstream random initializations naturally yield a flat, smooth, and easily optimizable landscape. **These analyses jointly offer a rigorous theoretical guarantee of SFF’s effectiveness**.
3. **Lines 289–295, 817–828, and 972–985:**  We introduce and compare a richer set of baselines—LoRA fine-tuning, label smoothing, SAM, SWA, Mixout, and L2-SP. Although some offer mild improvements over vanilla full fine-tuning, they remain far behind SFF, as they don't address the fundamental issue of the steep and unsmooth loss landscape of the pretrained LTSM.
4. **Lines 304–306 and 986–1004:**  We examine the impact of mainstream initialization methods (Kaiming, Xavier, etc.) and random seeds on SFF. Results show that all mainstream initializers produce stable gains.
5. **Lines 486–497:**  We correct the numerical-ordering typo that appeared in the earlier layout.
6. **Lines 1037–1052:**  We provide three practical guidelines for choosing $\alpha$ from complementary perspectives.
7. **Lines 1053–1074:**  We discuss the effect of SFF on normalization or scale across layers.

We are grateful for your thoughtful evaluation and guidance, which has substantially strengthened our work. We remain happy to provide further clarification or engage in additional discussion on any points that may require it.

We look forward to your continued feedback and evaluation.

---

### Meta-Review · Area_Chair_vVBv · 2025-12-31

**Summary:**

This paper proposes Smoothed Full Fine-tuning (SFF), a simple method for improving fine-tuning of large time-series models (LTSMs) by interpolating pretrained weights with a randomly initialized auxiliary model to smooth sharp loss landscapes. Reviewers generally agreed that the problem is important and practically relevant, and that the method is simple, inexpensive, and empirically effective across many models and tasks.

The main sources of disagreement among reviewers centered on novelty and theoretical rigor. Several reviewers initially viewed SFF as a repackaging of known weight interpolation or flatness-based ideas, while others found the framing and empirical validation within time-series foundation models to be compelling.

Overall, the rebuttal meaningfully addressed most technical and experimental concerns. Remaining issues primarily concern the degree of novelty relative to existing optimization techniques and whether the theoretical guarantees are sufficiently general rather than tailored to specific assumptions. Taking all reviews together, the paper appears borderline to moderately positive, with improved clarity and rigor after revision, and reasonable justification for acceptance as a poster-level contribution.

**Reviewer Concerns:**

### Reviewer hGpf
Addressed:
- Theoretical justification: The authors added detailed Hessian-based analysis explaining sharp vs. flat minima and why interpolation smooths sharp minima without harming flat ones.
- Baselines: Added LoRA, SAM, SWA, Mixout, L2-SP, label smoothing, and Gaussian/Xavier/Kaiming perturbations.
- Sensitivity to initialization: Extensive experiments across seeds and initialization schemes show robustness.

Partially Outstanding:
- Novelty: Although distinctions from model soups and continual learning are clarified, some may still see the contribution as incremental rather than fundamentally new.


### Reviewer t8mv
Addressed:
- Theoretical rigor: The rebuttal directly responded with formal derivations and expanded analysis.
- Hyperparameter guidance: Provided practical rules, validation-based tuning, and discussion of future adaptive methods.
- Auxiliary model choice: Justified via theory and empirical comparisons across initialization strategies.

Outstanding:
- None critical; concerns largely resolved to the reviewer’s satisfaction.


### Reviewer wYp2
Addressed:
- Limited baselines: Added multiple flatness- and regularization-based baselines.
- Zero-shot degradation: Identified and fixed a numerical ordering error.
- Normalization / scale concerns: Added discussion and theory arguing interpolation does not harm flat minima or normalization layers.

Partially Outstanding:
- Novelty framing: Still mainly convincing within the time-series domain rather than broadly across ML.


### Reviewer r8RN
Addressed:
- Why random initialization is flat: Added theory and citations (Goldilocks zone, Hessian analysis).
- Interpolation parameter selection: Provided empirical ranges, validation strategy, and future directions.
- Depth of theory: Expanded Section 3.1 substantially with formal derivations.

Partially Outstanding:
- Desire for even deeper or more general loss-landscape analysis, beyond quadratic/Hessian approximations.

**Reviewer Scores:**

- Reviewer hGpf: 4
  (From 2 → 4: concerns on theory and baselines largely addressed, novelty still debated.)

- Reviewer t8mv: 8
  (Remains strong accept; rebuttal reinforces original positive view.)

- Reviewer wYp2: 6
  (From 4 → 6 (likely): improved baselines, corrections, and analysis make acceptance reasonable.)

- Reviewer r8RN: 6
  (From 4 → 6 (likely): theory and initialization concerns addressed, remaining issues are forward-looking.)

---

### Decision · Program_Chairs · 2026-01-26

Accept (Poster)